# KAIROS: TOWARDS ADAPTIVE AND GENERALIZABLE TIME SERIES FOUNDATION MODELS

## ABSTRACT

Time series foundation models (TSFMs) have emerged as a powerful paradigm for time series analysis, driven by large-scale pretraining on diverse data corpora. However, time series inherently exhibit heterogeneous information density over time, influenced by system states and signal complexity, presenting significant modeling challenges especially in a zero-shot scenario. Current TSFMs rely on non-adaptive processing pipelines that fail to capture this dynamic nature. For example, common tokenization strategies such as fixed-size patching enforce rigid observational granularity, limiting their ability to adapt to varying information densities. Similarly, conventional positional encodings impose a uniform temporal scale, making it difficult to model diverse periodicities and trends across series. To overcome these limitations, we propose KAIROS, a flexible TSFM framework that integrates a dynamic patching tokenizer and an instance-adaptive positional embedding. KAIROS adaptively selects tokenization granularity and tailors positional encodings to the unique characteristics of each time series instance. Trained on a large-scale Predictability-Stratified Time Series (PreSTS) corpus comprising over 300 billion time points and adopting a multi-patch prediction strategy in the inference stage, KAIROS achieves superior performance with much fewer parameters on two common zero-shot benchmarks, GIFT-Eval and the Time-Series-Library benchmark, consistently outperforming established methods across diverse tasks.

## 1 INTRODUCTION

Time series forecasting is a core component of many real-world applications, including cloud services, traffic management, retail, finance, and energy (Box et al., 2015; Hyndman & Athanasopoulos, 2018). Although time series data are ubiquitous, many individual datasets suffer from scarcity, making it difficult to train models independently, particularly for deep neural networks that require large amounts of data. This challenge has long motivated the development of few-shot and zero-shot forecasting methods (Oreshkin et al., 2020). More recently, inspired by the success of large-scale foundation models (Achiam et al., 2023; Kirillov et al., 2023), researchers have turned to building general-purpose time series foundation models (TSFMs) for forecasting. These models leverage vast and diverse time series corpora to acquire strong generalization capabilities, showing promising results across a wide range of downstream tasks (Ansari et al., 2024; Woo et al., 2024; Shi et al., 2024).

Despite this progress, a key limitation persists: current TSFMs rely on non-adaptive processing pipelines that fail to reflect the dynamic and heterogeneous nature of time series. As shown in Figure 1(b) and exemplified in Figure 1(c), the statistical information density measured via spectral entropy on sliding windows (see Appendix G) varies not only across datasets but also across samples within the same series. However, existing models typically impose a *uniform and invariant scale* across all series, thereby overlooking internal variability. This holds true whether they tokenize each time point individually (Ansari et al., 2024; Shi et al., 2024), tokenize using fixed-size patches (Das et al., 2024), or model temporal dependencies using standard positional encodings (Liu et al., 2025).

This non-adaptive paradigm gives rise to two major challenges. (i) Local information extraction: Current tokenization strategies enforce fixed granularity, requiring the model to observe whole series with the same temporal resolution. Such rigidity prevents adaptation to heterogeneous or fluctuating information density, both across different series and within a single sequence. For instance, models

cannot zoom in on fine-grained details during sudden market shocks, nor can they abstract more efficiently when the data remain stable. (ii) Temporal relationship modeling: Conventional positional encodings impose a unified temporal scale across sequences, neglecting differences in periodicity, trend and seasonality. As a result, TSFMs struggle to adapt to heterogeneous domains, for example, power consumption data measured hourly versus retail sales data measured daily, each exhibiting distinct temporal dynamics. Figure 1(b) and 1(d) illustrate these challenges, highlighting both the variability of information density within time series and the rigidity of current methods.

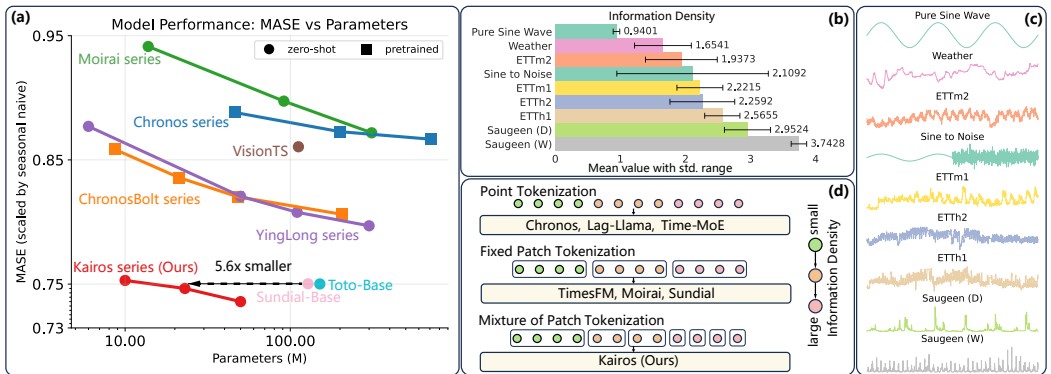

Figure 1: **(a)** The trade-off between performance (normalized MASE) and the number of parameters on GIFT-Eval benchmark (Aksu et al., 2024) for existing TSFMs. Our KAIROS achieves a superior performance at a comparable parameter scale. **(b) (c)** Significant variation exists in information density (i.e., signal complexity) across and within different time series datasets. **(d)** Existing TSFMs primarily use tokenization methods like point-wise or fixed-size patching, while our KAIROS utilizes a Mixture of Patch Tokenization to address dynamic changes in information density.

To address these challenges, we propose KAIROS, a foundation model designed to handle the complexities of heterogeneous time series data. The name KAIROS is derived from its mythological significance as the Greek god of the "opportune moment", reflecting its adaptability. For local information extraction, we introduce a Mixture-of-Size Dynamic Patching (MoS-DP) mechanism that adaptively models time series at multiple granularities by selecting patch sizes based on local information density. Inspired by the computationally free null experts in Mixture-of-Experts architectures (Zeng et al., 2024; Jin et al., 2024), we further incorporate null patch sizes, which allow the model to flexibly vary the number of active patches. This design moves beyond static patching strategies, enabling more expressive modeling of nuanced, time-varying dynamics. For temporal relationship modeling, we propose Instance-adaptive Rotary Position Embedding (IARoPE), an extension of RoPE (Su et al., 2024), tailored for diverse temporal dynamics. Unlike standard RoPE, which represents positional information with a fixed temporal scale, our IARoPE modulates these scales using spectral features extracted from each series, thereby generating positional encodings uniquely adapted to individual instances.

In addition to our core architectural innovations, we further introduce two key optimizations in the inference and training phases, respectively. For inference, we propose a multi-patch prediction strategy that simultaneously forecasts multiple future patches. This approach enhances forecasting robustness by mitigating cumulative errors and offers greater flexibility for variable-length prediction horizons. To further enhance training, we construct the Predictability-Stratified Time Series (PreSTS) corpus, a large-scale and diverse pretraining dataset curated through a targeted sampling strategy. By prioritizing more predictable sequences while maintaining broad coverage, PreSTS provides high-quality supervision that supports efficient model scaling. As demonstrated by the results on the GIFT-Eval benchmark (Aksu et al., 2024) in Figure 1(a), these combined innovations allow our KAIROS to achieve superior performance while using fewer parameters. This strong performance is consistently observed on Time-Series-Library (TSLib) (Wang et al., 2024b) benchmarks as well.

In summary, our main contributions are as follows: 1) We propose KAIROS, a novel foundation model specifically designed to handle dynamically changing time series data. KAIROS demonstrates strong resilience to commonly observed data variability in real-world time series. 2) The two key architectural innovations introduced, i.e., MoS-DP and IARoPE, address the challenges of varying

information density and instance-specific temporal dynamics, thereby enhancing the generalization capability of foundation models for time series. 3) Through extensive empirical evaluation, we demonstrate the superior performance of KAIROS with significantly fewer parameters on zero-shot forecasting benchmarks, highlighting its flexibility and adaptability across a wide range of diverse and non-stationary time series datasets.

## 2 RELATED WORK

**Time Series Foundation Models (TSFMs).** The scarcity of domain-specific time series data has long motivated research into models with strong few-shot or zero-shot learning capabilities (Oreshkin et al., 2020; 2021). Recently, inspired by the strong generalization of large language models, researchers are devoting more effort to TSFMs leveraging vast data corpora. Some works attempt to directly transfer the sequence modeling capabilities of large language models to the time series domain (Gruver et al., 2023). Others leverage the scalability of Transformer (Vaswani et al., 2017) and directly train unified foundation models on extensive time series corpora (Ansari et al., 2024; Das et al., 2024; Woo et al., 2024; Cohen et al., 2025; Liu et al., 2025). A few studies have also explored lightweight architectures, including Multi-Layer Perceptrons (MLPs) such as TTM (Ekambaram et al., 2024). However, these models often apply fixed and unified tokenization strategies and position embedding to time series from diverse domains with distinct characteristics, limiting their cross-domain generalization. KAIROS, in contrast, employs a dynamic patching tokenizer and instance-adaptive position embedding to enhance its generalizability for diverse time series data.

**Time Series Tokenization.** Leveraging Transformer model backbone requires an effective mechanism for local information extraction, i.e., to transform raw data into sequential elements (tokens), discretely or continuously (Sennrich et al., 2016; Agarwal et al., 2025). (1) Point-wise: The intuitive approach for time series tokenization is to map each time point to a token. However, models employ this approach (Ansari et al., 2024; Rasul et al., 2023; Shi et al., 2024) suffer from sensitivity to noise and computational inefficiency. (2) Uni-size: To address this, PatchTST (Nie et al., 2023) introduces a fixed-size patching method that became a widely adopted paradigm in many subsequent TSFMs (Woo et al., 2024; Liu et al., 2025). However, the fixed-size patches failed to flexibly adapt to time series with varying information densities. (3) Multi-size: While models like Pathformer (Chen et al., 2024) and ElasTST (Zhang et al., 2024) attempt to address these limitations by improving feature fusion strategies, they still simply partition the entire sequence into a few predefined patch sizes, failing to dynamically adjust to the varying information densities in single time series data. Furthermore, as they are designed for dataset-specific training rather than as foundation models, they lack cross-domain generalization capabilities. Other foundation models implement different yet similarly rigid strategies. For instance, OTIS (Turgut et al., 2024) adapts at the domain level with learnable variate embeddings but maintains a fixed patch granularity; WaveToken (Masserano et al., 2024) uses a non-adaptive pipeline, transforming signals via a fixed wavelet decomposition and quantization process. Similarly, while TTM (Ekambaram et al., 2024) incorporates multi-scale feature extraction, this process also remains static for heterogeneous time series. The core limitation unifying these methods is their inability to adapt tokenization granularity to the varying, local information densities within a single time series. (4) Mixture-size: In contrast, KAIROS improves upon the existing methods by dynamically selecting appropriate patch sizes for each segment of a time series, rather than relying on one or more fixed patch sizes for the entire sequence.

**Position Embedding.** Due to the position-unaware characteristic of self-attention mechanism, Transformers relies on position embeddings (PE) to model temporal information. This contrasts with recent non-Transformer architectures, such as TiRex (Auer et al., 2025) and FlowState (Graf et al., 2025), which inherently preserve temporal order through sequential states and thus obviate the need for PE. However, as Transformers remain the dominant backbone, most adopt existing designs for PE in Natural Language Processing (NLP) domains (Das et al., 2024; Shi et al., 2024), which typically emphasize long-term decay through methods like sinusoidal calculations or by suppressing long-range positional information (Vaswani et al., 2017; Su et al., 2024). By imposing a uniform temporal scale, these PEs struggle to model the heterogeneous temporal relationships of time series. Existing adaptive PE methods in NLP (Zheng et al., 2024; Lin et al., 2024) are also insufficient, as they target context extrapolation rather than the complex temporal structures of time series. In the time series domain, ElasTST (Zhang et al., 2024) proposed a tunable RoPE to better adapt to time series data. However, its adaptation uses per-dataset training rather than dynamic adjustment. To

address this limitation, we propose to dynamically modulate PE tailored to the intrinsic characteristics of each input time series, enhancing the effectiveness of time series foundation models.

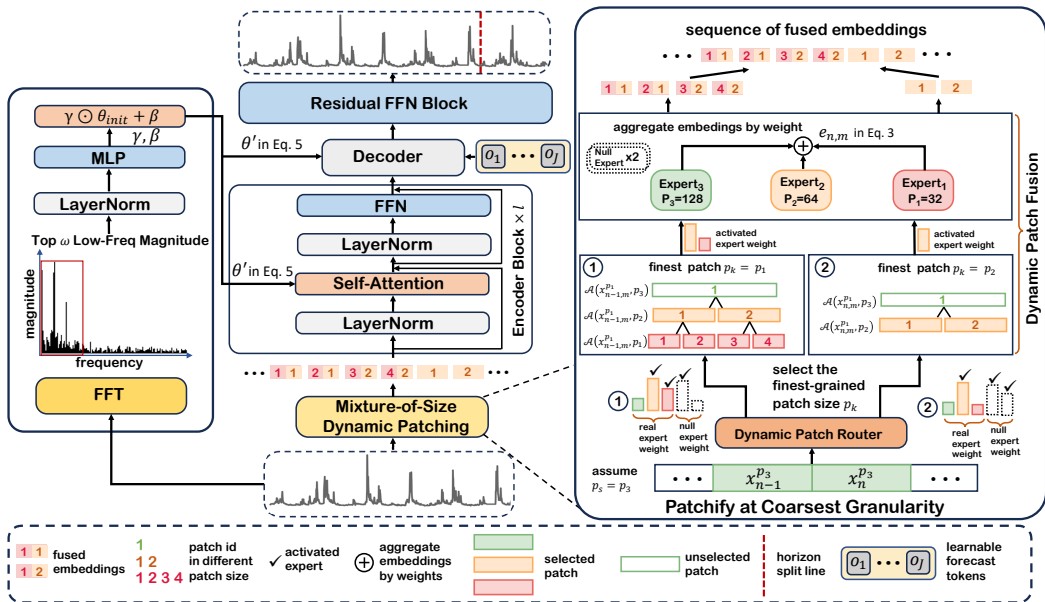

Figure 2: The architecture of KAIROS, which including (1) Mixture-of-Size Dynamic Patching (MoS-DP): This module adaptively tokenizes the time series by fusing features from multiple granularities. As detailed in the expanded view, (i) a Dynamic Patch Router first selects active experts (each corresponding to a patch size using same color) for a coarsest patch and determines the finest patch size $p_k$ for tokenization. Activation is indicated by routing weights and check marks. (ii) The final embedding for each resulting finest patch is then created via a hierarchical fusion process. For instance, the case ① illustrates how the embedding for a finest patch of size $p_1$ aggregates information from its ancestors of size $p_2$, but not from its ancestors of size $p_3$, with filled patches denoting selected experts and outlined patches denoting inactive ones, as summarized in the legend. (iii) This results in a sequence of fused embeddings rich with multi-scale information that is fed into the encoder. (2) Instance-Adaptive RoPE (IARoPE): This module (left) adjusts positional encodings for the Transformer by modulating them based on the unique frequency profile of each input series.

## 3 METHODOLOGY

### 3.1 OVERALL FRAMEWORK

We present the forecasting problem formulation: given the historical observations of a time series $\boldsymbol{x}_{1:T} = (x_1, \ldots, x_T) \in \mathbb{R}^T$, the objective is to forecast the future values $\boldsymbol{x}_{T+1:T+H} = (x_{T+1}, \ldots, x_{T+H}) \in \mathbb{R}^H$, where $T$ is the lookback window and $H$ denotes the forecast horizon.

The architecture of KAIROS, shown in Figure 2, consists of three main components. First, the input time series is tokenized into a sequence of embeddings by our Mixture-of-Size Dynamic Patching (MoS-DP) module to extract multi-granularity local information. These embeddings are then processed by a Transformer encoder, which uses our proposed Instance-Adaptive Rotary Position Embedding (IARoPE) to model complex temporal relationships. Finally, a Transformer decoder utilizes a multi-patch prediction strategy for forecasting. The details of each key component, including MoS-DP, IARoPE, and the multi-patch prediction strategy, are presented in the following sections.

### 3.2 MIXTURE-OF-SIZE DYNAMIC PATCHING (MOS-DP)

In contrast to time-series foundation models such as Time-MoE (Shi et al., 2024) and Moirai-MoE (Liu et al., 2024b), which directly replace the original feed-forward network (FFN) layers in Transformer blocks with Mixture-of-Experts (MoE) modules (Jacobs et al., 1991; Shazeer et al.,

2017), KAIROS takes inspiration from MoE to tackle the intrinsic heterogeneity of time series. To improve local information extraction, we introduce the MoS-DP module, which performs routing at the level of each time series patch and thereby effectively addresses the challenge of adapting foundation models to diverse datasets with varying information granularities. MoS-DP unfolds in three stages: Patchify at Coarsest Granularity, Dynamic Patch Router, and Dynamic Patch Fusion.

**Patchify at Coarsest Granularity.** In this stage, the entire time series is partitioned based on the maximum patch size. We begin with an initial partition of the sequence. A set of $S$ patch sizes is defined in advance, denoted as $\{p_1, \ldots, p_S\} \in \mathbb{R}^S$, where $p_1 < p_2 < \ldots < p_S$. We first tokenize the input time series $\boldsymbol{x}_{1:T} \in \mathbb{R}^T$ using non-overlapping patching using the coarsest patch size $p_S$. The series is divided into $N = \lceil \frac{T}{p_S} \rceil$ coarsest patches. If the original series length $T$ is not an exact multiple of $p_S$, $\boldsymbol{x}_{1:T}$ is augmented with left-padding of zeros as needed. This process yields a sequence of $N$ coarsest patches, $\{\boldsymbol{x}_n^{p_S}\}_{n=1}^N$, where each patch has a uniform length of $p_S$.

**Dynamic Patch Router.** To perform the final multi-granularity information fusion, we select a set of appropriate granularities for each coarsest patch $\boldsymbol{x}_n^{p_S}$. Inspired by the MoE paradigm, we employ multiple FFNs as experts, with each expert dedicated to extracting features at a unique granularity level from the set $\{p_1, \ldots, p_S\}$. Furthermore, we introduce $Z$ null experts, which perform no computation when selected, allowing the number of activated experts to be dynamic for each input, facilitating more flexible modeling of heterogeneous time series. The routing process is formulated as

$$g_{n,i} = \begin{cases} s'_{n,i}, & s'_{n,i} \in \mathrm{TopK}(\{s'_{n,j} | 1 \leqslant j \leqslant S + Z\}, K) \text{ and } 1 \leqslant i \leqslant S \\ 0, & \text{otherwise.} \end{cases} \quad (1)$$

$$s'_{n,i} = \frac{\exp(s_{n,i} + \boldsymbol{b}_i)}{\sum_{j=1}^S \exp(s_{n,j} + \boldsymbol{b}_j)}, \quad \text{where } s_{n,i} = \mathbf{W}_i \boldsymbol{x}_n^{p_S}. \quad (2)$$

Here $g_{n,i}$ represents the gating value for the $i$-th expert with respect to the $n$-th coarsest patch; $s_{n,i}$ is the affinity score between patch and the expert, and $\mathbf{W}_i$ is the trainable weight matrix for the $i$-th expert. The bias term $\boldsymbol{b}_i$ is utilized for an Auxiliary-Loss-Free Load Balancing method (Liu et al., 2024a) and is discussed in Appendix D.3.1.

The coarsest patch $\boldsymbol{x}_n^{p_S}$ will then be partitioned into smaller finest patches. The size of these finest patches, $p_k$ is determined by the finest granularity selected by the router for this specific coarsest patch, i.e., $p_k = \min\{p_i | g_{n,i} > 0\}$. Note that this finest granularity can differ for each coarsest patch $\boldsymbol{x}_n^{p_S}$. All $N$ coarsest patches are re-segmented in parallel based on their respective routing decisions. Compared to uniformly segmenting the entire sequence with a fixed size (Chen et al., 2024; Zhang et al., 2024) as discussed in Section 2, our method offers greater flexibility to adapt to temporal sequences with varying information densities.

**Dynamic Patch Fusion.** In the final stage, we yield the embedding for each finest patch by fusing information from multiple granularities. This process ensures that the final token representations are aware of the context at various temporal scales. Specifically, for the $n$-th coarsest patch $\boldsymbol{x}_n^{p_S}$, which was determined in the previous stage, it is first partitioned into $M_n = \frac{p_S}{p_k}$ non-overlapping finest patches. Here, $p_k$ is the finest patch size selected by the Dynamic Patch Router for this coarsest patch. We denote the $m$-th finest patch within the $n$-th coarsest patch as $\boldsymbol{x}_{n,m}^{p_k}$. Each such finest patch corresponds to a single token for the subsequent Transformer encoder. The embedding $e_{n,m}$ for the finest patch $\boldsymbol{x}_{n,m}^{p_k}$ is calculated as the weighted summation of semantic information originating from all activated granularities, formulated as:

$$\boldsymbol{e}_{n,m} = \sum_{i=k}^S \alpha_{n,i} \cdot \mathrm{MLP}_i(\mathcal{A}(x_{n,m}^{p_k}, p_i)), \text{ where } \alpha_{n,i} = \frac{g_{n,i}}{\sum_{j=k}^S g_{n,j}}, \quad \boldsymbol{e}_{n,m} \in \mathbb{R}^{D_h}, \quad (3)$$

where $\mathcal{A}(x_{n,m}^{p_k}, p_i)$ identifies the ancestor patch of size $p_i$ that contains the finest patch $x_{n,m}^{p_k}$, as illustrated in Figure 2. We provide a formal definition of $\mathcal{A}(\cdot, \cdot)$ and a detailed example in Appendix D.3.2. $\mathrm{MLP}_i$ denotes the expert that extracts the embedding for patch size $p_i$. The term $\alpha_{n,i}$ represents the normalized gating value from the router for the $n$-th coarsest patch, which dynamically weighs the importance of each granularity. $D_h$ is the hidden dimension. Following Eq. 3, $M$ embeddings are extracted from the coarsest patch $\boldsymbol{x}_n^{p_S}$ via the Dynamic Patch Fusion (DPF) module as:

$$\mathrm{DPF}(\boldsymbol{x}_n^{p_S}, \boldsymbol{g}_n) = [\boldsymbol{e}_{n,1}, \boldsymbol{e}_{n,2}, \ldots, \boldsymbol{e}_{n,M_n}] \in \mathbb{R}^{M_n \times D_h}, \quad \boldsymbol{g}_n = (g_{n,1}, g_{n,2} \ldots, g_{n,S}). \quad (4)$$

Eventually, the original time series $\boldsymbol{x}_{1:T}$ is transformed into a sequence of embeddings by concatenating all DPF module outputs as $\mathbf{E} = \text{DPF}(\boldsymbol{x}_1^{p_S}, \boldsymbol{g}_1)||\text{DPF}(\boldsymbol{x}_2^{p_S}, \boldsymbol{g}_2)|| \ldots ||\text{DPF}(\boldsymbol{x}_N^{p_S}, \boldsymbol{g}_N)$ with sequence length $T' = \sum_{n=1}^N M_n$, where $M_n$ represents the number of finest patches that the $n$-th patch in $\boldsymbol{x}_{1:N}^{p_S}$ is segmented into by the Dynamic Patch Router. This final sequence $\mathbf{E} \in \mathbb{R}^{T' \times D_h}$ would serve as the input to the subsequent Transformer module of KAIROS. This design allows the model to use small patches for in-depth analysis where local details are rich and large patches for efficient modeling where information is stable, as demonstrated in Section 4.4.1.

### 3.3 INSTANCE-ADAPTIVE ROTARY POSITION EMBEDDING (IARoPE)

Rotary Position Embedding (RoPE) (Su et al., 2024) encodes relative positions by rotating hidden vectors within Transformer layer, such as a query ($\boldsymbol{q}$) or a key ($\boldsymbol{k}$), which we denote generically as $\boldsymbol{z}$. The operation is formulated as $f_{\text{RoPE}}(\boldsymbol{z}, t) = (\boldsymbol{z}_{2j} + i\boldsymbol{z}_{2j+1})e^{it\theta_{\text{init},j}}$, where the base RoPE frequencies $\theta_{\text{init},j} = b^{-2j/D_h}$ are determined by a fixed hyperparameter $b$. A detailed derivation of the RoPE calculation is provided in Appendix D.4. Moreover, our theoretical analysis in Theorem 1 shows that, under standard RoPE, the contribution to attention weight from each 2D subspace of query and key vectors varies periodically in the relative distance between positions, with the period determined by the fixed base frequency $\theta_{\text{init},j}$. However, these fixed frequencies impose a single, uniform temporal scale across all data. This rigidity prevents the model from effectively modeling temporal relationships for series with diverse intrinsic structures, such as different seasonalities or trends. To address this limitation, we propose Instance-Adaptive Rotary Position Embedding (IARoPE), which dynamically tailors customized positional encodings for each time series instance.

The core idea of IARoPE is to modulate the base RoPE frequencies $\theta_{\text{init}}$ using spectral features extracted from each instance. Specifically, Given an input time series instance $\mathbf{x}_{1:T}$, we first extract its frequency features by applying the Fast Fourier Transform (FFT) and select the first $\omega$ low-frequency components (their magnitudes) to form a feature vector $\mathbf{x}_{\text{fft}}$, where $\omega$ is a hyperparameter. We focus on low frequencies as they often capture dominant trends and periodic patterns crucial for understanding the temporal structure of instance (Zhou et al., 2022). Then we feed $\mathbf{x}_{\text{fft}}$ into a small MLP to predict instance-specific modulation parameters: a scaling factor $\gamma \in \mathbb{R}^{D_h/2}$ and a bias factor $\beta \in \mathbb{R}^{D_h/2}$. These parameters then adapt the base RoPE frequencies $\theta_{\text{init}} \in \mathbb{R}^{D_h/2}$ via an element-wise affine transformation to produce the adaptive frequencies $\theta' \in \mathbb{R}^{D_h/2}$:

$$\theta'_j = \gamma_j \odot \theta_{\text{init},j} + \beta_j \quad \text{for } j = 0, 1, \ldots, D_h/2 - 1, \tag{5}$$

where $\odot$ denotes the Hadamard product. The resulting $\theta'$ incorporates learned information from the instance's frequency characteristics. Finally, these adaptive angles $\theta'$ replace the fixed $\theta$ in the RoPE calculation for the current instance:

$$f_{\text{IARoPE}}(\boldsymbol{z}, t) = (\boldsymbol{z}_{2j} + i\boldsymbol{z}_{2j+1})e^{it\theta'_j}, \tag{6}$$

This mechanism adapts the positional encoding to the frequency profile of each time series instance, potentially offering a more accurate representation of relative temporal positions. We demonstrate the effectiveness of this adaptation in Section 4.4.2.

### 3.4 MULTI-PATCH PREDICTION

KAIROS employs multi-patch prediction to effectively mitigate the cumulative errors inherent in autoregressive processes. Specifically, after the encoder outputs the hidden representation $\boldsymbol{h}^{\text{en}} \in \mathbb{R}^{T' \times D_h}$, a Transformer decoder is employed to forecast future values. We introduce a set of $J$ learnable forecast tokens, denoted as $Q_{\text{forecast}} = \{o_1, \ldots, o_J\}$. These tokens are passed as input to decoder, which processes them simultaneously to produce a sequence of output embeddings $\boldsymbol{h}^{\text{de}} \in \mathbb{R}^{J \times D_h}$:

$$\boldsymbol{h}^{\text{de}} = \text{Decoder}(Q_{\text{forecast}}, \boldsymbol{h}^{\text{en}}). \tag{7}$$

Each output embedding $\boldsymbol{h}_j^{\text{de}}$ is then passed through a shared prediction head (a residual FFN block (Das et al., 2024)) to forecast a patch of the future of length $H_s = H/J$ as

$$\hat{\boldsymbol{x}}_{T+jH_s+1:T+(j+1)H_s} = \text{ResidualFFN}(\boldsymbol{h}_j^{\text{de}}), \quad j = \{1, \ldots, J\} \tag{8}$$

This multi-token design is motivated by the limitations of two common forecasting paradigms. First, compared to conventional single-step autoregressive models (Ansari et al., 2024; Liu et al., 2024c),

predicting $J$ patches at once enhances computational efficiency and mitigates cumulative error by reducing iterative steps. Second, unlike models that forecast a larger patch (Das et al., 2024), our approach provides greater flexibility for shorter prediction horizons. It can construct a precise forecast by aggregating outputs from the necessary number of tokens, avoiding the potentially imprecise truncation of a larger-than-needed output. We provide a detailed analysis of this design in Appendix B.

## 3.5 TRAINING CORPUS

For pre-training, we curated Predictability-Stratified Time Series (PreSTS) corpus, a corpus of over 300 billion time points. PreSTS comprises real-world time series spanning multiple domains and is augmented by a synthetic dataset we constructed to ensure comprehensive coverage (see Appendix E.1 for analysis). Additionally, we have excluded all test sets found within the GIFT-Eval benchmark (Aksu et al., 2024) to maintain evaluation integrity. While a prevailing consensus regarding TSFMs highlights the necessity of a diverse training dataset, research has also shown that the presence of anomalies in the training set can severely degrade a model's predictive capabilities (Cheng et al., 2024). Therefore, the real-world datasets in PreSTS are stratified into five tiers based on their degree of predictability. Datasets exhibiting higher predictability are assigned a greater sampling probability during training, a strategy that enables the model to undergo a more robust, high-quality training regimen without sacrificing the comprehensiveness of the data. Comprehensive descriptions of our training datasets along with synthetic data generation algorithm, are provided in Appendix E.1.

## 4 EXPERIMENT

In this section, we conduct a comprehensive evaluation to assess the performance of KAIROS and the effect of proposed components. Specifically, we investigate the following three key research questions. **RQ1:** Does KAIROS achieve better generalization and zero-shot prediction than previous methods? (Section 4.2) **RQ2:** How effectively does MoS-DP handle time series heterogeneity by adapting its patch size according to local information densities? (Section 4.3 and 4.4.1) **RQ3:** Does IARoPE effectively generate customized position embedding tailored to the unique temporal structure of each time series instance? (Section 4.3 and 4.4.2)

### 4.1 EXPERIMENT SETTINGS

#### 4.1.1 EVALUATION DATASETS AND METRIC

To comprehensively evaluate the performance of our proposed method, we conducted zero-shot assessments on two well-known public benchmarks: GIFT-Eval (Aksu et al., 2024) and Time-Series-Library (TSLib) (Wang et al., 2024b). We train KAIROS in three sizes: mini (10M), small (23M) and base (50M). The training details are provided in Appendix D.1.

**Benchmarks.** GIFT-Eval benchmark consists of 97 prediction tasks, with 55 tasks dedicated to short-term predictions, 21 for medium-term predictions, and 21 for long-term predictions, providing a comprehensive means of assessing the model's predictive capabilities. For the TSLib benchmark, we followed the common practice in Time-MoE (Shi et al., 2024) and selected the ETT datasets (Zhou et al., 2021) and Weather (Zhou et al., 2021). Additionally, to test the model's adaptability to diverse sampling frequencies, we incorporated the Saugeen datasets (Godahewa et al., 2021), which include data with both daily and weekly sampling frequencies. These were referred to as Saugeen (D) and Saugeen (W), respectively. The detailed description of the evaluation datasets, their splitting configurations, and the selection of the evaluation length are provided in Appendix E.2 and E.3.

**Metric.** In our evaluation of GIFT-Eval, consistent with prior research (Auer et al., 2025; Liu et al., 2025), we employ the official Mean Absolute Scaled Error (MASE) and Continuous Ranked Probability Score (CRPS) metrics to assess point and probabilistic forecasting performance, respectively. These metrics are first normalized by the Seasonal Naïve baseline for each individual task. The final score is then computed as the geometric mean of these normalized values. For the TSLib benchmark, we followed previous works (Liu et al., 2023; 2024c; Nie et al., 2023) by adopting Mean Squared Error (MSE) and Mean Absolute Error (MAE) to evaluate time series forecasting performance. The metric calculations are detailed in Appendix D.2.1.

### 4.1.2 COMPARED METHODS

We compare KAIROS with several state-of-the-art models, including TSFMs such as Sundial (Liu et al., 2025), Toto (Cohen et al., 2025), YingLong (Wang et al., 2025), VisionTS (Chen et al., 2025), Time-MoE (Shi et al., 2024), ChronosBolt (Ansari et al., 2024), TTM (Ekambaram et al., 2024), Moirai (Woo et al., 2024), Timer-XL (Liu et al., 2024c), TimesFM-2.0 (Das et al., 2024), and Chronos (Ansari et al., 2024), as well as advanced full-shot deep learning models, including PatchTST (Nie et al., 2023), DLinear (Zeng et al., 2022), iTransformer (Liu et al., 2023), Pathformer (Chen et al., 2024), and TimesNet (Wu et al., 2022).

## 4.2 ZERO-SHOT EVALUATION

**Finding 1:** KAIROS *demonstrates remarkable zero-shot forecasting capabilities across multiple benchmarks.* On the GIFT-Eval benchmark, as presented in Table 1, KAIROS$_{base}$ (50M) achieves the best MASE and the second-best CRPS when compared to state-of-the-art task-specific deep learning methods and other TSFMs. Notably, KAIROS$_{small}$ (23M) surpasses both Toto and Sundial in MASE, despite having a parameter count that is 6.6 and 5.6 times smaller, respectively. On the TSLib benchmark, as shown in Figure 3, our lightweight KAIROS$_{mini}$ (10M) surpasses the performance of both recent advanced TSFMs and the majority of full-shot deep learning models. Compared with other TSFMs, KAIROS achieves a significant improvement in predictive performance.

Table 1: Performance evaluation on the GIFT-Eval. We evaluated model performance using the normalized MASE and CRPS metrics delineated in Section 4.1.1, where lower values indicate higher forecasting accuracy. The baseline models fall into three categories: statistical methods, deep learning (DL) models, and TSFMs. Following the official classification of GIFT-Eval, TSFMs are further subdivided into TestData Leakage and Zero-Shot. TestData Leakage signifies that the model's training set included test data, whereas Zero-Shot indicates that the model was trained without any data from the test set or its corresponding training split. Baseline results are officially reported by GIFT-Eval.

| Type | Statistical | DL (Full-Shot) | | TSFMs (TestData Leakage) | | | | TSFMs (Zero-Shot) | | | | | | |
|---|---|---|---|---|---|---|---|---|---|---|---|---|---|---|
| Method | Seasonal Naïve | DLinear | PTST. | TTM | Chronos | Chronos Bolt | TimesFM | Moirai | VisionTS | Ying. | Toto | Sundial | KAIROS$_s$ (ours) | KAIROS$_b$ (ours) |
| #Params | - | - | - | 5M | 709M | 205M | 500M | 311M | 112M | 300M | 151M | 128M | 23M | 50M |
| MASE | 1.000 | 1.061 | 0.849 | 1.020 | 0.870 | 0.808 | 0.758 | 0.875 | 0.863 | 0.798 | 0.750 | 0.750 | 0.748 | **0.742** |
| CRPS | 1.000 | 0.846 | 0.587 | 0.873 | 0.574 | 0.574 | 0.550 | 0.599 | 0.755 | 0.548 | **0.517** | 0.559 | 0.554 | 0.548 |

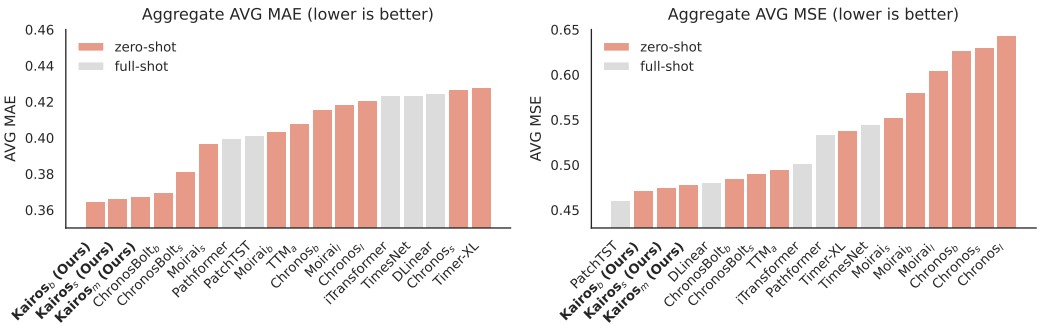

Figure 3: Zero-shot forecasting performance on TSLib. Results are averaged across prediction lengths {96, 192, 336, 720}. The subscripts $l$, $b$, $s$, and $a$ represent model sizes of large, base, small, and advanced, respectively. The complete experimental results are presented in Appendix F.

## 4.3 ABLATION STUDY

**Finding 2:** *MoS-DP and IARoPE collaboratively enable* KAIROS *to adapt to heterogeneous time series, achieving powerful predictive performance.* To investigate the effectiveness of our proposed MoS-DP and IARoPE, we conducted a comprehensive ablation study, with the results summarized in Table 2. For a fair comparison, KAIROS without MoS-DP uses a fixed patch size of 64, which was determined to be optimal via hyperparameter search. For IARoPE, we compare against the original RoPE (Su et al., 2024), which employs a fixed, predefined $\theta$. The integration of MoS-DP and IARoPE into our method leads to the best performance on the 97 tasks of the GIFT-Eval benchmark,

which validates the superiority of our designs. Furthermore, the improvement of KAIROS gets more significant compared to KAIROS without MoS-DP or IARoPE as horizon gets longer, exhibiting better long-term forecasting ability.

## 4.4 MODEL ANALYSIS

### 4.4.1 PATCH DISTRIBUTION ANALYSIS

**Finding 3:** KAIROS *demonstrates adaptive capability to varying information densities, enabling it to model complex temporal dynamics with appropriate granularity.* For each coarsest patch, KAIROS dynamically fuses information by selecting from a set of granularities (patch sizes). To quantify this dynamic routing, we calculate a weighted average patch size by multiplying the weight assigned to each granularity by its corresponding patch size. Consequently, a smaller value for this metric indicates the model's preference for finer-grained processing

Table 2: Ablation study comparing MoS-DP and RoPE variants. We evaluated ablation settings using the normalized MASE from Section 4.1.1, assessing performance on individual prediction horizons and in aggregate across all tasks.

| MoS-DP | RoPE Variant | Short | Medium | Long | **AVG** |
|:---:|:---:|:---:|:---:|:---:|:---:|
| ✓ | IARoPE | **0.719** | **0.759** | **0.789** | **0.742** |
| ✗ | IARoPE | 0.725 | 0.779 | 0.813 | 0.755 |
| ✓ | RoPE | 0.726 | 0.783 | 0.828 | 0.759 |
| ✗ | RoPE | 0.731 | 0.784 | 0.829 | 0.763 |

of a given coarsest patch, while a larger value suggests a preference for coarser granularity. In Figure 4, this weighted average patch size is visualized by the background color, with darker shades corresponding to smaller values. As the figure illustrates, KAIROS strategically employs smaller patch sizes for sequences with abrupt changes and larger patch sizes for stable sequences. This adaptive resolution enhances KAIROS versatility for a wide range of domains and tasks.

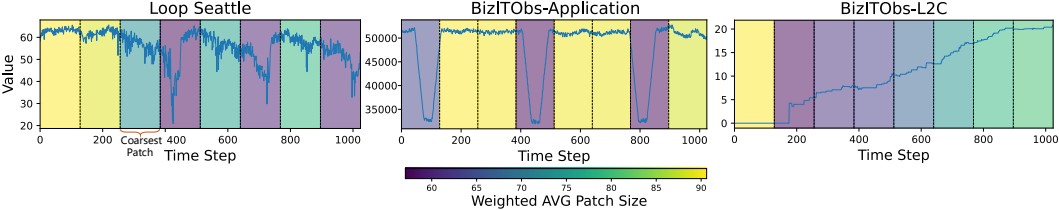

Figure 4: Patch size preferences in GIFT-Eval test datasets. Darker shades indicate a smaller weighted average patch size, signifying the model's preference for finer-grained processing in that region.

### 4.4.2 IARoPE ANALYSIS

**Finding 4:** *Modulating RoPE frequencies $\theta$ instance-wise can indeed better model time series temporal relationships.* To verify that the performance gains of IARoPE are indeed caused by its instance-specific RoPE frequencies $\theta$ modulation, we designed a series of control experiments that impair or ablate this mechanism by shuffling the modulation parameters (details in Figure 5; Appendix E.4). The results confirm IARoPE's superior forecasting accuracy, demonstrating the advantage of tailoring PE to individual instance frequency characteristics. The Intra-Dataset Shuffle ($\theta$ modulations permuted between different instances within the same dataset) performed second best, suggesting that while instance-specific modulation is optimal, leveraging dataset-level frequency similarities still offers utility. This result is attributed to the shared spectral characteristics among samples within the same dataset, which allow shuffled parameters to preserve dataset-level adaptivity and thus outperform the rigid original RoPE. Conversely, both Inter-

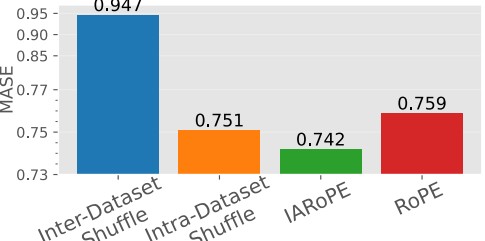

Figure 5: Causal analysis of adaptive modulation by IARoPE. This experiment validates the criticality of matching positional encodings to the unique characteristics of each time series instance by disrupting or removing this adaptation. We test this by manipulating the RoPE frequencies $\theta$ under several conditions: IARoPE (standard), Intra-Dataset Shuffle ($\theta$ modulations permuted between different instances within the same dataset), Inter-Dataset Shuffle ($\theta$ modulations from instances of other datasets), and Fixed RoPE (no modulation).

Dataset Shuffle ($\theta$ modulations from instances of other datasets) and Fixed RoPE (no modulation) yielded poorer performance, highlighting that mismatched or static $\theta$ parameters are suboptimal as they fail to capture the specific temporal nature of each series.

## 5 CONCLUSIONS

We introduce KAIROS, a time series foundation model that advances the design of modules specifically adapted to the unique characteristics of time series data. KAIROS achieves this through key innovations, including Mixture-of-Size Dynamic Patching, which considers varying information densities for more precise and efficient modeling, and Instance-Adaptive RoPE, which provides more accurate positional information tailored to time series nuances. Our comprehensive experiments reveal superior performance achieved KAIROS, highlighting the importance of designing foundational models that are inherently aligned with the structural nuances of time series data. One limitation of our current approach is that KAIROS does not explicitly model inter-variable dependencies. While it supports multivariate time series through channel-independent modeling (Nie et al., 2023), it does not capture the complex interactions among different variables. Incorporating mechanisms to model inter-variable relationships remains an important direction for future research. Additionally, exploring the application of KAIROS to a broader range of time series tasks beyond forecasting is a worthwhile direction for future work.

## 6 REPRODUCIBILITY STATEMENT

To facilitate full reproducibility, our complete implementation is publicly available in the supplementary material, including source code, pretrained model weights, and configuration files. Our proposed method, KAIROS, is detailed in Section 3. All experimental settings, encompassing dataset preprocessing, hyperparameters, and the computational environment, are specified in Appendix D.1. These resources are provided to enable the community to verify our results and build upon this work.

## 7 ETHICS STATEMENT

Our research is grounded in the ethical principles of the ICLR Code of Ethics, focusing on data integrity and responsible model development. Our methodology leverages both public datasets, including the pretraining datasets and evaluation datasets, and a custom synthetic dataset. The public datasets were sourced and utilized in strict compliance with their governance policies and are free of personally identifiable information (PII). Our synthetic dataset, being algorithmically generated, inherently carries no privacy risks and its characteristics are fully controlled. Acknowledging the potential for societal biases within the public data, we have integrated fairness considerations into our research process to prevent discriminatory outcomes. We are committed to advancing the field through both technical innovation and ethically sound practices.

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

## A The Use of Large Language Models

To improve the quality of this manuscript, we employed a Large Language Model (LLM) to assist with refining prose, correcting grammar, and enhancing the overall narrative flow, thereby making our scientific contributions more accessible. The intellectual contributions originated exclusively from the authors. Every LLM-generated suggestion was critically reviewed, edited, and approved by the authors to ensure it accurately reflected our original thoughts and findings. The authors assume full accountability for the originality and accuracy of the entire manuscript.

## B Analysis of Multi-Patch Prediction

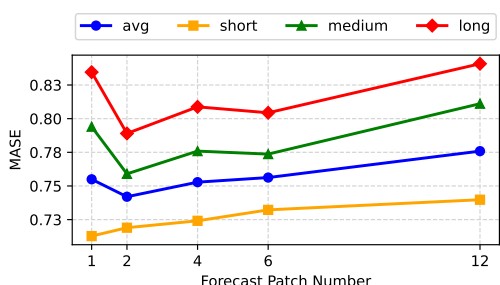

Figure 6: Performance analysis of multi-patch prediction on the GIFT-Eval benchmark across short, medium, and long horizons. We vary the number of forecast patches and find that a forecast patch number of 2 achieves the optimal trade-off, resulting in the best overall performance (lowest normalized MASE).

Unlike task-specific, non-autoregressive models such as TFT (Lim et al., 2021) and Informer (Zhou et al., 2021), which predict all future time points simultaneously, their architecture confines them to forecasting a fixed length identical to the training sequence, precluding variable-length predictions. Conversely, time series foundation models typically employ an iterative process: they predict a segment of the sequence, append it to the historical data, and then forecast the subsequent segment, thereby enabling autoregressive predictions of arbitrary length. To mitigate the cumulative errors inherent in this autoregressive methodology, Kairos introduces forecast tokens designed to predict multiple patches within each autoregressive step.

In this section, we present an analysis of the multi-patch prediction strategy introduced in Section 3.4. During the training phase, we experimented with a range of forecast patch numbers, specifically $J = \{1, 2, 4, 6, 12\}$. The corresponding evaluation results on the GIFT-Eval benchmark are illustrated in Figure 6. Our observations reveal that when $J = 1$, which corresponds to the conventional approach of predicting a single patch (Ansari et al., 2024; Liu et al., 2024c; Das et al., 2024), the model achieves optimal performance in short-term forecasting. However, this method necessitates multiple iterations of autoregressive prediction, leading to a significant degradation in performance for medium- and long-term forecasting.

Conversely, as we increase the forecast patch number $J$, the number of required autoregressive steps is markedly reduced. For instance, when $J = 2$, the autoregression frequency for medium- and long-term predictions is halved compared to the $J = 1$ case. This reduction yields a substantial improvement in forecasting accuracy over these longer horizons, demonstrating that our proposed multi-patch prediction strategy effectively mitigates the cumulative error inherent in autoregressive processes for medium- and long-term forecasting.

Nevertheless, we noted that the forecasting performance does not improve indefinitely with an increasing the forecast patch number $J$. We attribute this phenomenon to the escalated difficulty of the prediction task, which hinders the model's ability to optimize effectively. Ultimately, by evaluating the mean normalized MASE, we identified the optimal trade-off, selecting a forecast patch number of $J = 2$ for KAIROS.

# C  ANALYSIS OF INFERENCE SPEED

To provide a comprehensive view of our model's efficiency, we benchmarked the single-batch inference speed of KAIROS against several state-of-the-art models. The experimental setup involved an input sequence length of 2048 and a prediction horizon of 96 on a single NVIDIA TITAN RTX GPU. For TTM-Advanced, its maximum supported input length of 1536 was used.

As shown in Table 3, the inference time of KAIROS is on the same order of magnitude as other highly efficient models like ChronosBolt and Moirai. This demonstrates that KAIROS maintains competitive computational efficiency while delivering the superior forecasting accuracy detailed in the main paper. Notably, it is significantly faster than models like Time-MoE, TimesFM, and especially Chronos.

Table 3: Comparison of single-batch inference speeds. All models were tested with an input length of 2048 and an output length of 96, with the exception of TTM-Advanced (*), which used its maximum input length of 1536. Times are averaged per batch and reported in seconds.

| Model | Inference Time (s) |
|---|---|
| TTM-Advanced* | 0.012 |
| Timer-XL | 0.021 |
| ChronosBolt-Base | 0.062 |
| **KAIROS (Ours)** | **0.065** |
| Moirai-Large | 0.077 |
| Time-MoE-Large | 0.156 |
| TimesFM | 0.210 |
| Chronos-Large | 4.901 |

# D  IMPLEMENTATION DETAILS

## D.1  TRAINING DETAILS

### D.1.1  MODEL CONFIGURATIONS

We train KAIROS in three sizes: mini (10M), small (23M) and base (50M) with the detailed model configurations are in Table 4. The base model are trained for 300,000 steps with batch sizes of 512. We employ the AdamW optimizer, a linear decay learning rate adjustment strategy for model optimization. The learning rate for parameters related to IARoPE is set to 1e-5, while the learning rate for others is set to 1e-3. Training is conducted on $4 \times$ NVIDIA A100 GPUs using TF32 precision, which takes only 15 hours for base size.

Table 4: Details of KAIROS model configurations.

| | Layers | Heads | $d_{\text{model}}$ | $d_{\text{ff}}$ | $d_{\text{expert}}$ | $P$ | $K$ | $S$ | $Z$ | $\tau$ | $\eta_b$ | $w$ | $\theta_{\text{init}}$ | Params |
|---|---|---|---|---|---|---|---|---|---|---|---|---|---|---|
| KAIROS$_{\text{mini}}$ | 4 | 4 | 256 | 1024 | 1408 | $\{32, 64, 128\}$ | 3 | 3 | 2 | $\{0.55, 0.1, 0.05, 0.15, 0.15\}$ | 0.01 | 128 | $10000^{-2j/64}$ | 10M |
| KAIROS$_{\text{small}}$ | 4 | 8 | 384 | 1536 | 1408 | $\{32, 64, 128\}$ | 3 | 3 | 2 | $\{0.55, 0.1, 0.05, 0.15, 0.15\}$ | 0.01 | 128 | $10000^{-2j/64}$ | 23M |
| KAIROS$_{\text{base}}$ | 6 | 8 | 512 | 2048 | 1408 | $\{32, 64, 128\}$ | 3 | 3 | 2 | $\{0.55, 0.1, 0.05, 0.15, 0.15\}$ | 0.01 | 128 | $10000^{-2j/64}$ | 50M |

### D.1.2  LOSS FUNCTION

To better accommodate varied forecast horizons, and following the methodology of ElasTST (Zhang et al., 2024), we build upon the standard quantile loss by assigning distinct weights to each timestep, such that earlier predictions are given greater importance. The model's parameters are optimized by minimizing the quantile loss with weight decay, formulated as:

$$\mathcal{L} = \frac{1}{B} \sum_{i=1}^{B} \sum_{t=1}^{H} \frac{1}{K} \sum_{k=1}^{K} \omega(t) L_{\alpha_k}(y_{i,t}, q_{i,t}(\alpha_k)), \tag{9}$$

$$\omega(t) = \frac{1}{H}(\ln(H) - \ln(t)), \quad L_\alpha(y, q) = (\alpha - \mathbf{1}_{\{y < q\}})(y - q), \tag{10}$$

where $B, H, K$ are the batch size, forecast horizon, and number of quantiles, respectively; $y_{i,t}$ is the ground truth value, $q_{i,t}(\alpha_k)$ is the $k$-th quantile forecast, $\omega(t)$ is the weight at each time step $t$, and $L_{\alpha_k}$ denotes the pinball loss function. Empirically, we set $K = 9$ and use quantile levels of $\alpha = 0.1, 0.2, \ldots, 0.9$.

## D.2 EVALUATION DETAILS

### D.2.1 METRIC

**GIFT-Eval.** GIFT-Eval employs the Mean Absolute Scaled Error (MASE) and the Continuous Ranked Probability Score (CRPS) to assess the performance of point and probabilistic forecasts, respectively. Following the official evaluation protocol, we normalize the metrics for each task using a seasonal naïve baseline and subsequently aggregate the scores across all tasks via the geometric mean.

**TSLib.** We adopt mean square error (MSE) and mean absolute error (MAE) as evaluation metrics. These metrics are calculated as follows:

$$\text{MSE} = \frac{1}{H}\sum_{i=1}^{H}(x_i - \hat{x})^2, \quad \text{MAE} = \frac{1}{H}\sum_{i=1}^{H}|x_i - \hat{x}| \tag{11}$$

where $x_i$ is the ground truth and $\hat{x}_i$ s the prediction for the $i$-th future time point.

### D.2.2 STRIDE

Due to the significant inference latency of our baseline, Chronos (Ansari et al., 2024), we set the evaluation stride to 96 to enhance evaluation efficiency without compromising fairness. This setting is consistent with the protocols of Moirai (Woo et al., 2024), Moirai-MoE (Liu et al., 2024b), and the GIFT-Eval benchmark (Aksu et al., 2024), all of which set the stride equal to the prediction length. Additionally, the original Chronos paper evaluates the model on only a single window per dataset. Therefore, we wish to emphasize a critical point: while the specific numerical results would likely vary with a different stride, our chosen protocol is applied uniformly to all models under evaluation, ensuring a fair and equitable comparison.

## D.3 MIXTURE-OF-SIZE DYNAMIC PATCHING (MoS-DP)

### D.3.1 AUXILIARY-LOSS-FREE LOAD BALANCING

In this section, we elaborate on the details of bias previously introduced in Section 3.2. In order to ensure that different experts are adequately trained and to control the distribution ratios of various patch sizes, we employ an Auxiliary-Loss-Free Load Balancing method similar to that used in DeepSeek-V3 (Liu et al., 2024a). Specifically, we sum the normalized gating value of the $i$-th expert activated throughout the entire sequence to obtain $L_i$:

$$L_i = \sum_{n=1}^{N} \alpha_{n,i}, \tag{12}$$

where

$$\alpha_{n,i} = \frac{g_{n,i}}{\sum_{j=1}^{S} g_{n,j}}. \tag{13}$$

We define a target load distribution $\tau = (\tau_1, \tau_2, \ldots, \tau_S)$, where $\tau_i$ is the desired proportion of the total load for expert $i$, satisfying $\sum_{j=1}^{S} \tau_j = 1$. Next, we update the bias term $\boldsymbol{b}_i$ using $L_i$ and $\tau$:

$$\boldsymbol{b}_i \leftarrow \boldsymbol{b}_i + \eta_b \cdot \frac{\tau_i \cdot \sum_{j=1}^{S} L_j - L_i}{\sum_{j=1}^{S} L_j}, \tag{14}$$

where $\eta_b$ is a hyper-parameter to govern the magnitude of this adjustment and is referred to as the bias update speed.

In practical implementation, $\boldsymbol{b}_i$ is updated once per batch, i.e., $L_i$ denotes the sum of the normalized gating values for the $i$-th expert across all sequences within the entire batch. At the end of each step, the bias term $\boldsymbol{b}_i$ for each expert is adjusted. This dynamic adjustment of $\boldsymbol{b}_i$ aims to balance the workload across the experts according to the desired distribution by influencing future top-K selections, while also steering the patches-to-expert affinity scores $s'_{n,i}$ in subsequent batches, ensuring that the actual load distribution $L_i$ progressively aligns with the target distribution $\tau_i$, thereby promoting balanced expert utilization over time.

### D.3.2 DEFINITION OF THE ANCESTOR FUNCTION

In this section, we provide the formal definition for the function $\mathcal{A}(\cdot, \cdot)$ within the Dynamic Patch Fusion (DPF) module, as introduced in Section 3.2.

Consider the $n$-th coarsest patch, which is initially partitioned at the coarsest granularity, $p_S$. As this coarsest patch is processed by the Dynamic Patch Router, $K$ experts are activated. Let $p_k$ denote the minimum patch size among these activated experts. The coarsest patch is subsequently partitioned into $M_n = \frac{p_S}{p_k}$ non-overlapping finest patches. We denote the $m$-th such finest patch as $\boldsymbol{x}_{n,m}^{p_k}$.

The function $\mathcal{A}(\boldsymbol{x}_{n,m}^{p_k}, p_i)$ is then defined as the operation that retrieves the ancestor patch of size $p_i$ that contains the finest patch $\boldsymbol{x}_{n,m}^{p_k}$. Formally, this is expressed as:

$$\mathcal{A}(\boldsymbol{x}_{n,m}^{p_k}, p_i) = [\boldsymbol{x}_{\lfloor \frac{(m-1)\cdot p_k}{p_i} \rfloor \frac{p_i}{p_k}+1}^{p_k}, \ldots, \boldsymbol{x}_{\lfloor \frac{(m-1)\cdot p_k}{p_i}+1 \rfloor \frac{p_i}{p_k}}^{p_k}]. \tag{15}$$

We now illustrate the computational process with the following example. Let the setup be as follows:

- Set of available patch sizes: $\{p_1, \ldots, p_S\} = \{32, 64, 128\}$.
- Number of null experts: $Z = 2$.
- Number of activated experts: $K = 3$.
- Sequence context length: $T = 2048$.

The sequence is partitioned using the maximum patch size $p_S = 128$, resulting in $N = \lceil \frac{T}{p_S} \rceil = 16$ coarsest patches. For the first coarsest patch ($n = 1$), suppose the activated expert indices are $\{1, 3, 4\}$. Since experts 4 and 5 are designated as null experts, the fusion granularities are determined by the non-null activated experts, corresponding to patch sizes $p_1 = 32$ and $p_3 = 128$. The minimum patch size among these is $p_k = 32$. Consequently, this patch is partitioned into $M_n = \frac{p_S}{p_k} = \frac{128}{32} = 4$ non-overlapping finest patches. For the third finest patch ($m = 3$), the ancestor patches are computed as follows:

- For $i = 1$ and $p_i = p_1 = 32$:

$$\mathcal{A}(\boldsymbol{x}_{1,3}^{32}, 32) = \left[\boldsymbol{x}_{\lfloor \frac{(3-1)\cdot 32}{32} \rfloor \frac{32}{32}+1}^{32}, \ldots, \boldsymbol{x}_{\lfloor \frac{(3-1)\cdot 32}{32}+1 \rfloor \frac{32}{32}}^{32}\right] = [\boldsymbol{x}_3^{32}]. \tag{16}$$

  The ancestor patch is the finest patch itself.
- For $i = 2$ and $p_i = p_2 = 64$:

$$\mathcal{A}(\boldsymbol{x}_{1,3}^{32}, 64) = \left[\boldsymbol{x}_{\lfloor \frac{(3-1)\cdot 32}{64} \rfloor \frac{64}{32}+1}^{32}, \ldots, \boldsymbol{x}_{\lfloor \frac{(3-1)\cdot 32}{64}+1 \rfloor \frac{64}{32}}^{32}\right] = [\boldsymbol{x}_3^{32}, \boldsymbol{x}_4^{32}]. \tag{17}$$

  Expert 2 is not activated, so its normalized gating value, $\alpha_{n,i}$, in Equation 3 would be zero, meaning it does not contribute to the information fusion.
- For $i = 3$ and $p_i = p_3 = 128$:

$$\mathcal{A}(\boldsymbol{x}_{1,3}^{32}, 128) = \left[\boldsymbol{x}_{\lfloor \frac{(3-1)\cdot 32}{128} \rfloor \frac{128}{32}+1}^{32}, \ldots, \boldsymbol{x}_{\lfloor \frac{(3-1)\cdot 32}{128}+1 \rfloor \frac{128}{32}}^{32}\right] = [\boldsymbol{x}_1^{32}, \boldsymbol{x}_2^{32}, \boldsymbol{x}_3^{32}, \boldsymbol{x}_4^{32}]. \tag{18}$$

This mechanism, enabled by the function $\mathcal{A}(\cdot, \cdot)$, allows for the dynamic selection of different granularities for information fusion, tailored to the information density of each coarsest patch. Consequently, it effectively addresses the challenge of time series heterogeneity inherent in TSFMs.

### D.4 INSTANCE-ADAPTIVE ROTARY POSITION EMBEDDING (IARoPE)

In this section, we provide a more detailed description of the IARoPE implementation discussed in Section 3.3. We first introduce the principles of Rotary Position Embedding (RoPE) (Su et al., 2024) in detail, and then provide corresponding supplementary information on the IARoPE implementation.

#### D.4.1 DETAILS OF RoPE

The core idea of RoPE is to rotate segments of the query and key vectors by an angle proportional according to their absolute position in the sequence. This allows the model to discern relative positions through the geometry of these rotations, without needing explicit calculation of relative distances.

Specifically, RoPE operates on vectors of an even dimension, denoted as $D_h$. For a vector $\boldsymbol{e}$ (representing a query $\mathbf{q}$ or a key $\mathbf{k}$ in self-attention) at position $m$, it is transformed by a rotation matrix $\mathbf{R}_m$. This matrix is block-diagonal, composed of $D_h/2$ individual $2 \times 2$ rotation blocks. Each block $\mathbf{R}_{m,j}$ acts on a pair of dimensions $(e_{2j}, e_{2j+1})$ of the vector:

As introduced in Section 3.3, RoPE applies a rotation to input vector. This rotation is applied to pairs of dimensions $(e_{2j}, e_{2j+1})$, and can be concisely expressed using complex form:

$$f_{\text{RoPE}}(\boldsymbol{e}, m) = (e_{2j} + i e_{2j+1}) e^{im\theta_j}, \tag{19}$$

where $\theta_j$ is the angular frequency. This multiplication by $e^{im\theta_j}$ corresponds to a rotation in the complex plane. For the real-valued components $e_{2j}$ and $e_{2j+1}$, this operation is equivalent to applying the following $2 \times 2$ rotation matrix $\mathbf{R}_{m,j}$:

$$\mathbf{R}_{m,j} = \begin{bmatrix} \cos(m\theta_j) & -\sin(m\theta_j) \\ \sin(m\theta_j) & \cos(m\theta_j) \end{bmatrix}, \tag{20}$$

here, $m$ is the absolute position of the token. The term $\theta_j$ represents a predefined angular frequency for the $j$-th pair of dimensions ($j \in [0, D_h/2 - 1]$), typically defined as $\theta_j = b^{-2j/D_h}$ where $b = 10000$ in RoPE (Su et al., 2024) original setting.

The full rotation matrix $\mathbf{R}_m$ for position $m$ is thus:

$$\mathbf{R}_m = \text{diag}(\mathbf{R}_{m,0}, \mathbf{R}_{m,1}, \ldots, \mathbf{R}_{m,D_h/2-1}) \tag{21}$$

$$= \begin{pmatrix} \cos m\theta_0 & -\sin m\theta_0 & 0 & 0 & \cdots & 0 & 0 \\ \sin m\theta_0 & \cos m\theta_0 & 0 & 0 & \cdots & 0 & 0 \\ 0 & 0 & \cos m\theta_1 & -\sin m\theta_1 & \cdots & 0 & 0 \\ 0 & 0 & \sin m\theta_1 & \cos m\theta_1 & \cdots & 0 & 0 \\ \vdots & \vdots & \vdots & \vdots & \ddots & \vdots & \vdots \\ 0 & 0 & 0 & 0 & \cdots & \cos m\theta_{D_h/2-1} & -\sin m\theta_{D_h/2-1} \\ 0 & 0 & 0 & 0 & \cdots & \sin m\theta_{D_h/2-1} & \cos m\theta_{D_h/2-1} \end{pmatrix}. \tag{22}$$

Let $\mathbf{q}_m$ and $\mathbf{k}_n$ be the original query and key vectors for tokens at positions $m$ and $n$ respectively. After applying RoPE, their new representations become $\mathbf{q}'_m = \mathbf{R}_m \mathbf{q}_m$ and $\mathbf{k}'_n = \mathbf{R}_n \mathbf{k}_n$. The dot product for attention is then:

$$(\mathbf{q}'_m)^\top (\mathbf{k}'_n) = (\mathbf{R}_m \mathbf{q}_m)^\top (\mathbf{R}_n \mathbf{k}_n) = \mathbf{q}_m^\top \mathbf{R}_m^\top \mathbf{R}_n \mathbf{k}_n, \tag{23}$$

due to the properties of rotation matrices, $\mathbf{R}_m^\top \mathbf{R}_n = \mathbf{R}_{n-m}$ we can get:

$$(\mathbf{q}'_m)^\top (\mathbf{k}'_n) = \mathbf{q}_m^\top \mathbf{R}_{n-m} \mathbf{k}_n. \tag{24}$$

This equation shows that the dot product between a query and key vector after rotation inherently depends on their original values ($\mathbf{q}_m, \mathbf{k}_n$) and their relative positions ($n - m$). This allows RoPE to integrate relative positional information into the self-attention without any additional learnable parameters or explicit relative position computations.

Equation 24 confirms that the interaction relies on the relative position $n - m$ through the full rotation matrix $\mathbf{R}_{n-m}$. Since this matrix is block-diagonal, the total interaction is actually composed of independent rotations on disjoint 2D subspaces. The following theorem demonstrates the periodicity of the attention score within each subspace.

**Theorem 1** (Periodic dependence of RoPE attention). *Consider the $j$-th 2D subspace of the hidden state under RoPE, with angular frequency $\theta_j$, and let $\text{atten}_j(m, n)$ denote this subspace's contribution to the dot-product attention between positions $m$ and $n$. Then there exist an amplitude $A_j(m, n) \geq 0$ and a phase $\varphi_j(m, n)$, depending only on the content vectors at $m$ and $n$, such that*

$$
\begin{aligned}
\text{atten}_j(m, n) &= A_j(m, n) \cos\big(\theta_j(m - n) + \varphi_j(m, n)\big) \\
&\propto \cos\big(\theta_j(m - n) + \varphi_j(m, n)\big).
\end{aligned}
\tag{25}
$$

*In particular, for fixed content, $\text{atten}_j(m, n)$ is a periodic function of the relative distance $(m - n)$ with period $2\pi/\theta_j$.*

*Proof.* Let $q_{m,j}, k_{n,j} \in \mathbb{C}$ be the complex-valued representations of the $j$-th 2D subspace for the query and key at positions $m$ and $n$, respectively, as defined above. Applying RoPE multiplies these components by phase factors $e^{im\theta_j}$ and $e^{in\theta_j}$, yielding

$$
q'_{m,j} = q_{m,j} e^{im\theta_j}, \qquad k'_{n,j} = k_{n,j} e^{in\theta_j}.
\tag{26}
$$

The contribution of this subspace to the dot-product attention is the real part of $q'_{m,j}(k'_{n,j})^*$:

$$
\text{atten}_j(m, n) = \text{Re}\big[q'_{m,j}(k'_{n,j})^*\big] = \text{Re}\big[q_{m,j}k^*_{n,j}e^{i\theta_j(m-n)}\big].
\tag{27}
$$

Writing $q_{m,j}k^*_{n,j}$ in polar form as $q_{m,j}k^*_{n,j} = A_j(m, n)e^{i\varphi_j(m,n)}$ with $A_j(m, n) \geq 0$ and $\varphi_j(m, n) \in \mathbb{R}$, we obtain

$$
\begin{aligned}
\text{atten}_j(m, n) &= \text{Re}\big[A_j(m, n)e^{i(\theta_j(m-n)+\varphi_j(m,n))}\big] \\
&= A_j(m, n) \cos\big(\theta_j(m - n) + \varphi_j(m, n)\big),
\end{aligned}
\tag{28}
$$

which shows that $\text{atten}_j(m, n)$ is proportional to a cosine function of the relative distance $(m - n)$, with angular frequency $\theta_j$. The periodicity with period $2\pi/\theta_j$ follows immediately from the periodicity of the cosine function. $\square$

### D.4.2 DETAILS OF IARoPE

The initial RoPE frequencies $\theta_{\text{init},j}$ typically exhibit a wide numerical range. For instance, in our setting $\theta_{\text{init},0} = 1.0$ ($j = 0$), while $\theta_{\text{init},31} \approx 1.33 \times 10^{-4}$ ($j = 31$, corresponding to $D_h/2 - 1$ and $D_h = 64$). Directly applying affine modulation to these values could lead to numerical instability or disproportionate adjustments due to the vast scale differences.

To address this and ensure stable and effective modulation across the entire range of base frequencies, our IARoPE performs the adaptation in log-space. The layer-specific modulation parameters, $\gamma_j^{(l)}$ and $\beta_j^{(l)}$, predicted by the Multilayer Perceptron (MLP) for layer $l$ following Algorithm 1, are applied to the log-transformed base frequencies $\log \theta_{\text{init},j}$ via an element-wise affine transformation as

$$
\log \theta'^{(l)}_j = \gamma_j^{(l)} \odot \log \theta_{\text{init},j} + \beta_j^{(l)},
\tag{29}
$$

where $\log \theta'^{(l)}_j$ represents the modulated log-frequencies for layer $l$ and dimension pair $j$. Then, these modulated log-frequencies are transformed back to their original scale by exponentiation to obtain the adaptive rotation frequencies $\theta'^{(l)}_j$ used in the IARoPE calculation for layer $l$ as

$$
\theta'^{(l)}_j = \exp(\log \theta'^{(l)}_j).
\tag{30}
$$

---

**Algorithm 1** Generating Instance Adaptive Parameters $(\gamma^{(l)}, \beta^{(l)})$

---

**Input:** Time series instance $X \in \mathbb{R}^{B \times L}$, mask $M \in \{0,1\}^{B \times L}$, batch size $B$, sequence length $L$,
   FFT feature dimension $w$, MLP network $f_{\text{MLP}}$
**Output:** Adaptive parameters $\gamma^{(l)} \in \mathbb{R}^{B \times D_h/2}, \beta^{(l)} \in \mathbb{R}^{B \times D_h/2}$

1: $X_{\text{masked}} \leftarrow X \odot M$                                   ▷ Apply mask (element-wise product)
2: $F_{\text{result}} \leftarrow \text{RFFT}(X_{\text{masked}}), F_{\text{result}} \in \mathbb{R}^{B \times (\frac{L}{2}+1)}$        ▷ Real FFT along sequence dim
3: $F_{\text{amp}} \leftarrow |F_{\text{result}}|, F_{\text{amp}} \in \mathbb{R}^{B \times (\frac{L}{2}+1)}$               ▷ Amplitude spectrum
4: $X_{\text{FFT}} \leftarrow F_{\text{amp}}[\ldots, :w], X_{\text{FFT}} \in \mathbb{R}^{B \times w}$           ▷ Truncate low frequency
5: $X'_{\text{FFT}} \leftarrow \text{LayerNorm}_{\text{FFT}}(X_{\text{FFT}})$              ▷ Normalize FFT features
6: $\gamma^{(l)}, \beta^{(l)} \leftarrow f_{\text{MLP}}(X'_{\text{FFT}})$           ▷ Get instance-specific parameters
7: **return** $\gamma^{(l)}, \beta^{(l)}$

---

### D.4.3 INTERPRETATION OF IARoPE

From the theorem 1, the base RoPE parameters $\{\theta_{\text{init},j}\}_{j=0}^{D_h/2-1}$ define a fixed bank of sinusoidal kernels over relative positions: each $\theta_{\text{init},j}$ controls the frequency of one cosine kernel. This fixed frequencies works reasonably well when sequences share a homogeneous notion of temporal scale, but it can become suboptimal for a TSFM that must handle highly heterogeneous sampling intervals and spectral patterns across domains.

IARoPE addresses this by making these frequencies instance-adaptive. The FFT-based module in Algorithm 1 extracts simple spectral statistics $X'_{\text{FFT}}$ from each input sequence and maps them, via an MLP, to instance- and layer-specific parameters $\gamma_j^{(l)}$ and $\beta_j^{(l)}$. These parameters modulate the base frequencies in log-space:

$$\log \theta_j'^{(l)} = \gamma_j^{(l)} \odot \log \theta_{\text{init},j} + \beta_j^{(l)}, \quad \theta_j'^{(l)} = \exp \left( \log \theta_j'^{(l)} \right), \tag{31}$$

yielding an adapted frequency grid $\{\theta_j'^{(l)}\}$ for each layer and sequence.

Intuitively, this log-space affine transformation stretches or compresses the original RoPE frequency grid in a sequence-dependent way, while preserving the relative-position nature of RoPE. The resulting $\theta_j'^{(l)}$ still define sinusoidal kernels over relative lags, but their effective frequencies are now gently steered by the spectrum of the current input. Crucially, IARoPE does not aim to recover a single true period for each series; real-world time series are often multi-periodic and non-stationary. Instead, we interpret $\{\theta_j'^{(l)}\}$ as a sequence-dependent frequency profile that shapes how attention depends on relative lags, providing a more flexible and data-driven positional bias than fixed RoPE.

## E ADDITIONAL DETAILS OF EXPERIMENT SETTING

### E.1 PRE-TRAINING DATASETS

We trained KAIROS on the Predictability-Stratified Time Series (PreSTS) corpus, which consists of over 300B real-world time series observations from Chronos (Ansari et al., 2024) and Moirai (Woo et al., 2024) in conjunction with 15B synthetic time points. Following (Das et al., 2024), the training loader samples 80% real data and 20% synthetic data.

**Real-world data.** The real-world datasets were stratified into five tiers based on their predictability. This hierarchical structure dictates the sampling probability during model training, assigning a higher likelihood of selection to datasets with greater predictability. Such a strategy ensures that the model is preferentially trained on high-quality data while preserving its capacity to predict corner cases. Specifically, Tier 1 comprises datasets characterized by pronounced periodicity and trends with low noise. Tier 2 contains datasets with similarly distinct patterns but high noise, whereas Tier 3 includes those with subtle trends and considerable noise. The remaining datasets were classified into Tiers 4 and 5, based on a composite assessment of their size and pattern regularity. The specific details of these datasets, categorized by their respective sampling frequencies, are presented in Tables 5-8.

**Synthetic data.** We build a synthetic data generator that produces two distinct types of time series, each with a length of 4096. The first type consists of composite series, created by the additive combination of seasonal, trend, and noise components. For seasonality, we sample one or two components, where the primary period is drawn from {24, 48, 288, 360}, and a potential second period is a fixed seven-fold multiple of the first; the seasonal patterns manifest as either spike trains or smooth non-sinusoidal templates generated via interpolation. An optional trend is chosen from linear, exponential, or an ARIMA-like process derived from cumulating a stationary ARMA model's output. High-probability white Gaussian noise is also added, with each series guaranteed to contain at least one seasonal or trend component and all magnitudes bounded for stability. The generation process for these composite series is formally detailed in Algorithm 2. Complementing these are idealized industrial signals, which simulate perfectly regular machine cycles. These feature a constant baseline with repeating events like trapezoidal spikes or inverted-U shaped dips, where the period, amplitude, and width of the events remain fixed across the entire series with no random jitter, as outlined in Algorithm 3. Synthetic dataset cases are provided in Appendix H.1.

**Distribution Analysis.** To investigate whether the synthetic dataset supplements distributions absent in the real-world data, we adopted the statistical methodology of Toto (Cohen et al., 2025). We employed the ARCH-LM Statistic and Spectral Entropy to quantify time-varying volatility and information density (complexity), respectively. These metrics align closely with the design motivation for the MoS-DP and IARoPE components of Kairos. As illustrated in Figures 7 (a) and (b), we observed that a significant portion of the real-world dataset exhibits ARCH-LM Statistic values approaching zero and Spectral Entropy values nearing one, indicating high time-varying volatility and complexity. In contrast, the synthetic dataset exhibits the opposite characteristics, exposing the model during training to time series with more constant volatility and greater regularity. This exposure enhances the model's generalization capabilities. Furthermore, we statistically compared the periodicity of the real-world and synthetic datasets by plotting their respective Cumulative Distribution Functions (CDFs). As shown in Figure 7 (c), the dominant periods in the real-world dataset are heavily concentrated. This can cause the model to overfit to these specific periodicities and fail to learn generalizable periodic patterns. Conversely, the synthetic dataset presents a significantly smoother distribution of periods, enabling the model to generalize effectively across arbitrary periodicities. Consequently, the synthetic dataset effectively mitigates these biases present in the real-world data, thereby exposing the model to a more comprehensive and diverse set of distributions.

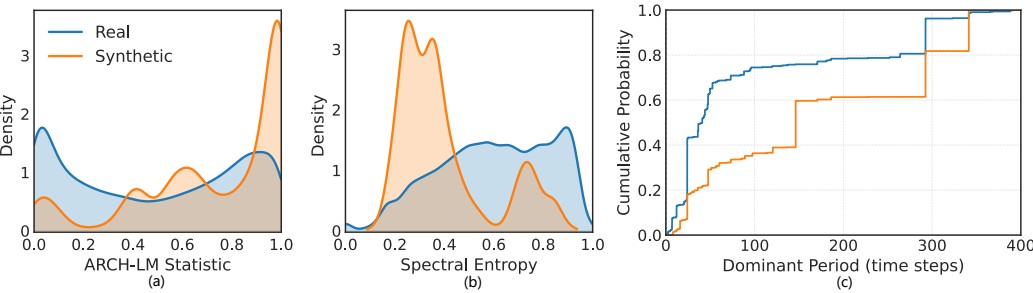

Figure 7: Comparison of real and synthetic datasets across **(a)** ARCH-LM statistic, **(b)** spectral entropy, and **(c)** dominant period distributions, illustrating complementary volatility, complexity, and periodicity characteristics.

### E.2 EVALUATION DATASETS

We select datasets from diverse domains and with varying sampling frequencies as evaluation datasets. The details are summarized in Table 9. For the evaluation on the TSLib benchmark, the datasets were split into training, validation, and test sets. The split for the ETT and Weather datasets follows the configuration adopted by iTransformer (Liu et al., 2023). All other datasets use a 70%/10%/20% ratio for the training, validation, and test sets, respectively.

Table 5: Detailed descriptions of second-level, minute-level, and hourly datasets.

| Dataset | Domain | Frequency | # Time Series | # Time points |
|---|---|---|---|---|
| Wind Power | Energy | 4S | 1 | 7,397,147 |
| Residential Load Power | Energy | T | 813 | 437,983,677 |
| Residential PV Power | Energy | T | 699 | 376,016,850 |
| Los-Loop | Transport | 5T | 207 | 7,094,304 |
| PEMS03 | Transport | 5T | 358 | 9,382,464 |
| PEMS04 | Transport | 5T | 921 | 15,649,632 |
| PEMS07 | Transport | 5T | 883 | 24,921,792 |
| PEMS08 | Transport | 5T | 510 | 9,106,560 |
| PEMS Bay | Transport | 5T | 325 | 16,941,600 |
| Alibaba Cluster Trace 2018 | CloudOps | 5T | 116,818 | 190,385,060 |
| Azure VM Traces 2017 | CloudOps | 5T | 159,472 | 885,522,908 |
| Borg Cluster Data 2011 | CloudOps | 5T | 286,772 | 1,075,105,708 |
| LargeST | Transport | 5T | 42,333 | 4,452,510,528 |
| KDD Cup 2022 | Energy | 10T | 134 | 4,727,519 |
| HZMetro | Transport | 15T | 160 | 380,320 |
| Q-Traffic | Transport | 15T | 45,148 | 264,386,688 |
| SHMetro | Transport | 15T | 576 | 5,073,984 |
| Beijing Subway | Transport | 30T | 552 | 867,744 |
| Elecdemand | Energy | 30T | 1 | 17,520 |
| Australian Electricity Demand | Energy | 30T | 5 | 1,155,264 |
| London Smart Meters | Energy | 30T | 5,560 | 166,528,896 |
| Taxi | Transport | 30T | 2428 | 3,589,798 |
| BDG-2 Bear | Energy | H | 91 | 1,482,312 |
| BDG-2 Fox | Energy | H | 135 | 2,324,568 |
| BDG-2 Panther | Energy | H | 105 | 919,800 |
| BDG-2 Rat | Energy | H | 280 | 4,728,288 |
| Borealis | Energy | H | 15 | 83,269 |
| BDG-2 Bull | Energy | H | 41 | 719,304 |
| China Air Quality | Nature | H | 2,622 | 34,435,404 |
| BDG-2 Cockatoo | Energy | H | 1 | 17,544 |
| Covid19 Energy | Energy | H | 1 | 31,912 |
| ELF | Energy | H | 1 | 21,792 |
| GEF12 | Energy | H | 20 | 788,280 |
| GEF14 | Energy | H | 1 | 17,520 |
| GEF17 | Energy | H | 8 | 140,352 |
| BDG-2 Hog | Energy | H | 24 | 421,056 |
| IDEAL | Energy | H | 217 | 1,255,253 |
| Low Carbon London | Energy | H | 713 | 9,543,553 |
| Oikolab Weather | Climate | H | 8 | 800,456 |
| PDB | Energy | H | 1 | 17,520 |
| Sceaux | Energy | H | 1 | 34,223 |
| SMART | Energy | H | 5 | 95,709 |
| Spanish Energy and Weather | Energy | H | 1 | 35,064 |
| ERCOT Load | Energy | H | 8 | 1,238,976 |
| Mexico City Bikes | Transport | H | 494 | 38,687,004 |
| Beijing Air Quality | Nature | H | 132 | 4,628,448 |
| Pedestrian Counts | Transport | H | 66 | 3,132,346 |
| Rideshare | Transport | H | 2,304 | 859,392 |
| Traffic | Transport | H | 862 | 15,122,928 |
| Taxi (Hourly) | Transport | H | 2,428 | 1,794,292 |
| Uber TLC (Hourly) | Transport | H | 262 | 1,138,128 |
| Wind Farms (Hourly) | Energy | H | 337 | 2,869,414 |
| Weatherbench (Hourly) | Nature | H | 225,280 | 78,992,150,528 |
| Buildings900K | Energy | H | 1,795,256 | 15,728,237,816 |
| ERA5 | Climate | H | 11,059,200 | 96,613,171,200 |
| CMIP6 | Climate | 6H | 14,327,808 | 104,592,998,400 |

Table 6: Detailed descriptions of daily datasets.

| Dataset | Domain | Frequency | # Time Series | # Time points |
|---|---|---|---|---|
| Bitcoin | Econ/Fin | D | 18 | 81,918 |
| Covid Mobility | Transport | D | 362 | 148,602 |
| Extended Web Traffic | Web | D | 145,063 | 370,926,091 |
| Favorita Sales | Sales | D | 111,840 | 139,179,538 |
| Favorita Transactions | Sales | D | 54 | 84,408 |
| Subseasonal | Climate | D | 3,448 | 56,788,560 |
| Subseasonal Precipitation | Climate | D | 862 | 9,760,426 |
| Sunspot | Nature | D | 1 | 73,894 |
| Vehicle Trips | Transport | D | 329 | 32,512 |
| Wiki-Rolling | Web | D | 47,675 | 40,619,100 |
| Dominick | Retail | D | 100,014 | 29,652,492 |
| M5 | Sales | D | 30,490 | 47,649,940 |
| Monash Weather | Climate | D | 3,010 | 43,032,000 |
| NN5 Daily | Econ/Fin | D | 111 | 87,801 |
| Uber TLC Daily | Transport | D | 262 | 47,422 |
| Weatherbench (Daily) | Nature | D | 225,280 | 3,291,336,704 |
| Wiki Daily (100k) | Web | D | 100,000 | 274,100,000 |
| Wind Farms (Daily) | Energy | D | 337 | 119,549 |
| Exchange Rate | Finance | D | 8 | 60,704 |

Table 7: Detailed descriptions of weekly datasets.

| Dataset | Domain | Frequency | # Time Series | # Time points |
|---|---|---|---|---|
| CDC Fluview ILINet | Healthcare | W | 375 | 319,515 |
| CDC Fluview WHO NREVSS | Healthcare | W | 296 | 167,040 |
| Kaggle Web Traffic Weekly | Web | W | 145,063 | 16,537,182 |
| Project Tycho | Healthcare | W | 1,258 | 1,377,707 |
| Traffic Weekly | Transport | W | 862 | 82,752 |
| NN5 Weekly | Econ/Fin | W | 111 | 12,543 |
| Weatherbench (Weekly) | Nature | W | 225,280 | 470,159,360 |

Table 8: Detailed descriptions of monthly, quarterly, and yearly datasets.

| Dataset | Domain | Frequency | # Time Series | # Time points |
|---|---|---|---|---|
| GoDaddy | Econ/Fin | M | 6,270 | 257,070 |
| CIF 2016 | Econ/Fin | M | 72 | 7,108 |
| FRED MD | Econ/Fin | M | 107 | 77,896 |
| M1 Monthly | Econ/Fin | M | 617 | 55,998 |
| M3 Monthly | Econ/Fin | M | 1,428 | 167,562 |
| Tourism Monthly | Econ/Fin | M | 366 | 109,280 |
| M3 Other | Econ/Fin | Q | 174 | 11,933 |
| M1 Quarterly | Econ/Fin | Q | 203 | 9,944 |
| M3 Quarterly | Econ/Fin | Q | 756 | 37,004 |
| Tourism Quarterly | Econ/Fin | Q | 427 | 42,544 |
| M1 Yearly | Econ/Fin | Y | 181 | 4,515 |
| M3 Yearly | Econ/Fin | Y | 645 | 18,319 |
| Tourism Yearly | Econ/Fin | Y | 518 | 12,757 |

Table 9: Detailed descriptions of evaluation datasets.

| Dataset | Domain | Frequency | # Time Series | # Target | # Time points |
|---|---|---|---|---|---|
| ETTh1 | Energy | H | 1 | 7 | 17,420 |
| ETTh2 | Energy | H | 1 | 7 | 17,420 |
| ETTm1 | Energy | 15T | 1 | 7 | 69,680 |
| ETTm2 | Energy | 15T | 1 | 7 | 69,680 |
| Weather | Nature | 10T | 1 | 21 | 52,696 |
| Saugeen (D) | Nature | D | 1 | 1 | 23,741 |
| Saugeen (W) | Nature | W | 1 | 1 | 3,391 |

---

**Algorithm 2** Composite Time Series Generation

---

**Input:** Time series length $L = 4096$, Primary period set $\mathcal{P}_1 = \{24, 48, 288, 360\}$, Harmonic multiplier $n = 7$, Trend types $\mathcal{T}_{\text{types}} = \{\text{linear, exp, ARMA}\}$, Seasonal patterns $\mathcal{S}_{\text{patterns}} = \{\text{spike, interpolated segment}\}$.

**Output:** A synthetic time series $\boldsymbol{x}_{1:L}$.

1: $\boldsymbol{x}, \boldsymbol{s}, \boldsymbol{t}, \boldsymbol{n} \leftarrow \boldsymbol{0}_{1:L}$ ▷ Initialize total series and components (seasonal, trend, noise)
2: Sample flags $f_s, f_t, f_n$ ▷ Determine component inclusion, ensuring $f_s \vee f_t$ is true
3: **if** $f_s$ is true **then**
4:     $k \sim \text{Bernoulli}(0.2)$ ▷ $k = 1$ for double period (20% prob), $k = 0$ for single
5:     $p_1 \sim \mathcal{U}(\mathcal{P}_1)$ ▷ Sample primary period
6:     $\mathcal{P}_{\text{active}} \leftarrow \{p_1\}$
7:     **if** $k = 1$ **then**
8:         $\mathcal{P}_{\text{active}} \leftarrow \mathcal{P}_{\text{active}} \cup \{n \cdot p_1\}$
9:     **end if**
10:    **for** $p$ in $\mathcal{P}_{\text{active}}$ **do**
11:       $a \sim \mathcal{U}(1.0, 3.0)$ ▷ Sample amplitude for this component
12:       $pattern \sim \mathcal{U}(\mathcal{S}_{\text{patterns}})$
13:       $\boldsymbol{c} \leftarrow \text{GeneratePattern}(pattern, p, a)$ ▷ Create a single cycle of the pattern
14:       $\boldsymbol{s} \leftarrow \boldsymbol{s} + \text{Tile}(\boldsymbol{c}, L)$ ▷ Tile the cycle to length $L$ and add to seasonal component
15:    **end for**
16: **end if**
17: **if** $f_t$ is true **then**
18:    $type \sim \mathcal{U}(\mathcal{T}_{\text{types}})$
19:    $\boldsymbol{t} \leftarrow \text{GenerateTrend}(type, L)$
20:    **if** $f_s$ is true **then**
21:       $\lambda \sim \mathcal{U}(0.1, 0.3)$ ▷ Reduce trend strength when seasonality is present
22:       $\boldsymbol{t} \leftarrow \lambda \cdot \boldsymbol{t}$
23:    **end if**
24: **end if**
25: **if** $f_n$ is true **then**
26:    $\sigma_n \sim \mathcal{U}(0.01, 0.1)$
27:    $\boldsymbol{n} \sim \mathcal{N}(\boldsymbol{0}, \sigma_n^2 \mathbf{I})$ ▷ Generate white Gaussian noise
28: **end if**
29: $\boldsymbol{x} \leftarrow \boldsymbol{s} + \boldsymbol{t} + \boldsymbol{n}$
30: **return** $\boldsymbol{x}_{1:L}$

---

**Algorithm 3** Idealized Industrial Signal Generation

---

**Input:** Time series length $L = 4096$, Pattern types $\mathcal{P}_{\text{types}} = \{\text{inverted\_u, spikes}\}$.

**Output:** A synthetic time series $\boldsymbol{x}_{1:L}$.

1: $type \sim \mathcal{U}(\mathcal{P}_{\text{types}})$
2: $(b, p, a, w, \sigma_n) \leftarrow \text{SampleParams}(type)$ ▷ Sample baseline, period, amplitude, width, noise
3: $\boldsymbol{x}_{1:L} \leftarrow b$ ▷ Initialize series with baseline value
4: $\boldsymbol{e} \leftarrow \text{TrapezoidShape}(w, a)$ ▷ Create the event shape of width $w$ and amplitude $a$
5: **if** $type = \text{inverted\_u}$ **then**
6:    $sign \leftarrow -1$
7: **else**
8:    $sign \leftarrow +1$
9: **end if**
10: **for** $i \leftarrow 0, p, 2p, \ldots$ up to $L - 1$ **do**
11:    $start \leftarrow i, end \leftarrow \min(i + w, L)$
12:    $\boldsymbol{x}_{start:end} \leftarrow \boldsymbol{x}_{start:end} + sign \cdot \boldsymbol{e}_{1:end-start}$ ▷ Add or subtract event shape at periodic intervals
13: **end for**
14: **if** $\sigma_n > 0$ **then**
15:    $\boldsymbol{x} \leftarrow \boldsymbol{x} + \mathcal{N}(\boldsymbol{0}, \sigma_n^2 \mathbf{I})$ ▷ Add global Gaussian noise
16: **end if**
17: **return** $\boldsymbol{x}_{1:L}$

### E.3 EVALUATION LENGTH SELECTION

In this section, we provide further details regarding the selection of historical sequence lengths for the various models evaluated in the TSLib benchmark, supplementing the discussion in Section 4.1.1.

To accommodate diverse application scenarios, an increasing number of TSFMs (Liu et al., 2025; Das et al., 2024; Liu et al., 2024c; Woo et al., 2024) have devoted attention to predicting over long contexts. Consequently, we evaluate KAIROS and other TSFMs under a long-context setting. Specifically, we adopt a context length of 2048 time steps and examine four prediction horizons, which are {96,192,336,720}. For TSFMs incapable of processing this context length, we instead employ the context length at which each model achieves its best performance.

For the full-shot deep learning baselines, **a hyperparameter search was conducted independently for each dataset to ensure a fair evaluation**. We compared the performance of each model using the context length from its original publication against a length of 2048, selecting the superior of the two. To be specific, lengths of 96 and 2048 were compared for DLinear (Zeng et al., 2022), iTransformer (Liu et al., 2023), TimesNet (Wu et al., 2022), and Pathformer, while lengths of 336, 512, and 2048 were compared for PatchTST (Nie et al., 2023).

The definitive context lengths adopted for each model are systematically tabulated in Table 10.

Table 10: Context Lengths for Models on the TSLib Benchmark.

| Method | DLinear | iTrans. | TimesNet | PatchTST | Path. | Chronos | Moirai | TimesFM-2.0 | Timer-XL | TTM$_a$ | ChronosBolt | KAIROS |
|---|---|---|---|---|---|---|---|---|---|---|---|---|
| Context length | {96, 2048} | {96, 2048} | {96, 2048} | {336, 512, 2048} | {96, 2048} | 512 | 2048 | 2048 | 2048 | 1536 | 2048 | 2048 |

### E.4 DETAILS OF IARoPE ANALYSIS

In this section, we explain in more detail the setting of the IARoPE analysis experiment discussed in Section 4.4.2. To more definitively ascertain whether these instance-specific modulations truly capture and leverage beneficial positional information derived from an instance's FFT features, we designed a series of "shuffle" experiments. These experiments function as a form of causal intervention analysis. The underlying hypothesis is: if the instance-derived $\theta$ modulations are crucial for IARoPE's performance, then disrupting this linkage by applying $\theta$ modulations from one instance to another (shuffling) should lead to a noticeable degradation in forecasting accuracy.

The details of each group in the experiment are as follows:

- IARoPE: Our proposed method, where each time series instance utilizes its own FFT-derived features to independently modulate its RoPE $\theta$ parameters at each layer.

- Intra-Dataset Shuffle: During inference, the learned layer-specific $\theta$ modulation parameters $(\gamma^{(l)}, \beta^{(l)})$ are randomly permuted among instances within the same batch and originating from the same dataset. The modulated $\theta$ are always shuffled in the same layer.

- Inter-Dataset Shuffle: For instances of a target dataset, layer-specific $\theta$ modulation parameters $(\gamma^{(l)}, \beta^{(l)})$ are randomly sampled from a pre-computed collection derived from a different source dataset and applied to instances of the target dataset. This process also ensures layer-wise correspondence of the applied modulations.

- Fixed RoPE: We use the fully trained IARoPE model but disable its instance-adaptive modulation during inference. Specifically, the prediction of $\gamma^{(l)}$ and $\beta^{(l)}$ is bypassed, and all instances revert to using the initial RoPE $\theta_{\text{init}}$ across all layers. This approach keeps the base model architecture and other learned weights identical to IARoPE, isolating the effect of the modulation.

By comparing IARoPE's performance against these shuffled and fixed $\theta$ configurations, we can better attribute performance gains or losses to the effectiveness of the instance-adaptive $\theta$ modulation process.

## F FULL EVALUATION RESULTS

In this section, we present the detailed results in Section 4.2. Table 11 presents the full results of the zero-shot forecasting experiments conducted at each forecast horizon. For Time-MoE, we report the results as presented in the original paper (Shi et al., 2024).

## G CALCULATION OF INFORMATION DENSITY

In this section, we present the formula for the information density illustrated in Figure 1(b). To effectively characterize and compare different time series datasets, we introduce a method to quantify their information density and the variation of this density over time. For this, similar to TimeMixer (Wang et al., 2024a), we utilize spectral entropy, a metric that measures the complexity and compressibility of a signal. In this context, a signal with low spectral entropy, such as one with a few dominant periodic components, is considered to have low information density due to its redundant and predictable nature. Conversely, a signal with high spectral entropy, resembling white noise, has its power spread broadly across the frequency spectrum, indicating a high degree of randomness and thus a higher information density.

Given the observations $\mathbf{x}_{1:T} = (x_1, \ldots, x_T) \in \mathbb{R}^T$ of a specific dataset, we analyze the time series using a sliding window approach. The series is segmented into $N$ windows, $\mathbf{w}_i$, each of size $M$ with a step size of $S$. In our analysis, we use a non-overlapping configuration where both the window size and the step size are set to 128 (i.e., $M = 128$, $S = 128$). The total number of windows is $N = \lfloor (T-M)/S \rfloor + 1$. To mitigate spectral leakage from windowing, we apply a Hamming window to each segment $\mathbf{w}_i$:

$$h[n] = 0.5 - 0.5 \cos\left(\frac{2\pi n}{M-1}\right), \quad 0 \leq n \leq M - 1. \tag{32}$$

For each resulting windowed segment $\mathbf{w}'_i = \mathbf{w}_i \cdot h$, we compute its normalized power spectral density, $p_i[k]$, which describes how the signal's power is distributed over different frequencies. The spectral entropy for the $i$-th window, $H_{SE}(\mathbf{w}_i)$, is then calculated using the Shannon entropy formula:

$$H_{SE}(\mathbf{w}_i) = -\sum_k p_i[k] \log_2(p_i[k]). \tag{33}$$

This process yields a sequence of entropy values, $(H_{SE}(\mathbf{w}_1), \ldots, H_{SE}(\mathbf{w}_N))$, where each value represents the localized information density of its corresponding time segment.

To obtain a holistic view of the dataset's characteristics, we compute two key statistics from this entropy sequence.

First, the mean spectral entropy ($\mu_{SE}$) serves as a measure of the average information density of the entire dataset. A higher $\mu_{SE}$ suggests that the dataset, on average, contains more complex and less predictable patterns. This allows for a direct comparison of the overall information content between different datasets.

$$\mu_{SE} = \frac{1}{N} \sum_{i=1}^{N} H_{SE}(\mathbf{w}_i). \tag{34}$$

Second, the standard deviation of spectral entropy ($\sigma_{SE}$) quantifies the variability of information density within the dataset. A small $\sigma_{SE}$ indicates that the dataset is stationary in its complexity, with a consistent level of information density throughout. A large $\sigma_{SE}$, however, reveals a non-stationary character, signifying substantial fluctuations in the signal's complexity and information content over time.

$$\sigma_{SE} = \sqrt{\frac{1}{N} \sum_{i=1}^{N} (H_{SE}(\mathbf{w}_i) - \mu_{SE})^2}. \tag{35}$$

Together, $\mu_{SE}$ and $\sigma_{SE}$ provide a concise yet powerful summary of a dataset's informational characteristics, enabling a quantitative assessment of its average complexity and internal dynamics.

Table 11: Full results of zero-shot forecasting experiments.

| Method | Metric | ETTh1 96 | 192 | 336 | 720 | AVG | ETTh2 96 | 192 | 336 | 720 | AVG | ETTm1 96 | 192 | 336 | 720 | AVG | ETTm2 96 | 192 | 336 | 720 | AVG | Weather 96 | 192 | 336 | 720 | AVG | Saugeen(D) 96 | 192 | 336 | 720 | AVG | Saugeen(W) 96 | 192 | 336 | AVG | AVG | Total 1st |
|---|---|---|---|---|---|---|---|---|---|---|---|---|---|---|---|---|---|---|---|---|---|---|---|---|---|---|---|---|---|---|---|---|---|---|---|---|---|
| DLinear | MSE | 0.377 | 0.416 | 0.446 | 0.482 | 0.430 | 0.28 | 0.398 | 0.505 | 0.796 | 0.495 | 0.312 | 0.339 | 0.364 | 0.401 | 0.354 | 0.174 | 0.228 | 0.287 | 0.396 | 0.271 | 0.174 | 0.214 | 0.256 | 0.309 | 0.238 | 0.829 | 0.832 | 0.826 | 0.858 | 0.836 | 0.820 | 0.781 | 0.781 | 0.817 | 0.480 | - |
| | MAE | 0.391 | 0.416 | 0.44 | 0.495 | 0.436 | 0.358 | 0.434 | 0.495 | 0.634 | 0.480 | 0.359 | 0.375 | 0.392 | 0.415 | 0.385 | 0.272 | 0.312 | 0.355 | 0.428 | 0.342 | 0.238 | 0.272 | 0.304 | 0.346 | 0.290 | 0.502 | 0.503 | 0.506 | 0.513 | 0.506 | 0.569 | 0.571 | 0.551 | 0.564 | 0.424 | |
| iTransformer | MSE | 0.383 | 0.441 | 0.478 | 0.47 | 0.443 | 0.282 | 0.367 | 0.402 | 0.406 | 0.364 | 0.327 | 0.363 | 0.390 | 0.424 | 0.376 | 0.336 | 0.369 | 0.398 | 0.428 | 0.383 | 0.171 | 0.218 | 0.278 | 0.357 | 0.256 | 0.778 | 0.795 | 0.800 | 0.839 | 0.803 | 0.988 | 1.041 | 1.007 | 1.012 | 0.501 | - |
| | MAE | 0.396 | 0.429 | 0.449 | 0.469 | 0.436 | 0.344 | 0.392 | 0.422 | 0.433 | 0.398 | 0.373 | 0.381 | 0.411 | 0.442 | 0.397 | 0.355 | 0.381 | 0.411 | 0.442 | 0.397 | 0.21 | 0.256 | 0.299 | 0.351 | 0.279 | 0.457 | 0.463 | 0.482 | 0.490 | 0.473 | 0.626 | 0.64 | 0.642 | 0.636 | 0.423 | |
| TimesNet | MSE | 0.408 | 0.456 | 0.508 | 0.567 | 0.485 | 0.315 | 0.435 | 0.481 | 0.455 | 0.422 | 0.358 | 0.428 | 0.494 | 0.550 | 0.458 | 0.175 | 0.251 | 0.307 | 0.412 | 0.286 | 0.149 | 0.195 | 0.254 | 0.360 | 0.261 | 1.177 | 1.268 | 1.129 | 1.136 | 1.178 | 0.817 | 0.777 | 0.751 | 0.782 | 0.544 | - |
| | MAE | 0.426 | 0.451 | 0.496 | 0.521 | 0.469 | 0.359 | 0.422 | 0.456 | 0.461 | 0.425 | 0.373 | 0.412 | 0.447 | 0.480 | 0.428 | 0.255 | 0.305 | 0.343 | 0.407 | 0.328 | 0.222 | 0.265 | 0.293 | 0.352 | 0.286 | 0.546 | 0.561 | 0.534 | 0.531 | 0.543 | 0.517 | 0.494 | 0.501 | 0.504 | 0.423 | |
| PatchTST | MSE | 0.382 | 0.416 | 0.443 | 0.466 | 0.427 | 0.276 | 0.365 | 0.420 | 0.424 | 0.361 | 0.278 | 0.351 | 0.363 | 0.414 | 0.346 | 0.162 | 0.225 | 0.290 | 0.414 | 0.273 | 0.152 | 0.200 | 0.258 | 0.341 | 0.238 | 0.827 | 0.83 | 0.819 | 0.848 | 0.831 | 0.849 | 0.847 | 0.751 | 0.826 | 0.456 | - |
| | MAE | 0.403 | 0.426 | 0.444 | 0.474 | 0.437 | 0.334 | 0.386 | 0.428 | 0.443 | 0.398 | 0.330 | 0.361 | 0.388 | 0.423 | 0.376 | 0.249 | 0.288 | 0.323 | 0.423 | 0.317 | 0.192 | 0.237 | 0.281 | 0.335 | 0.261 | 0.474 | 0.482 | 0.488 | 0.491 | 0.484 | 0.545 | 0.526 | 0.500 | 0.524 | 0.397 | |
| PathFormer | MSE | 0.405 | 0.466 | 0.506 | 0.494 | 0.468 | 0.302 | 0.356 | 0.395 | 0.463 | 0.379 | 0.302 | 0.356 | 0.395 | 0.463 | 0.379 | 0.162 | 0.225 | 0.290 | 0.382 | 0.278 | 0.200 | 0.241 | 0.296 | 0.341 | 0.238 | 1.168 | 1.266 | 1.142 | 1.149 | 1.181 | 0.932 | 0.949 | 0.877 | 0.919 | 0.533 | - |
| | MAE | 0.398 | 0.427 | 0.447 | 0.463 | 0.434 | 0.325 | 0.361 | 0.388 | 0.429 | 0.376 | 0.325 | 0.361 | 0.388 | 0.429 | 0.376 | 0.244 | 0.288 | 0.333 | 0.391 | 0.317 | 0.213 | 0.256 | 0.281 | 0.335 | 0.261 | 0.501 | 0.535 | 0.508 | 0.499 | 0.511 | 0.532 | 0.532 | 0.533 | 0.532 | 0.399 | |
| Chronos$_s$ | MSE | 0.516 | 0.599 | 0.613 | 0.603 | 0.583 | 0.304 | 0.407 | 0.427 | 0.450 | 0.397 | 0.504 | 0.601 | 0.667 | 0.728 | 0.625 | 0.207 | 0.277 | 0.343 | 0.468 | 0.324 | 0.209 | 0.267 | 0.328 | 0.397 | 0.300 | 1.108 | 1.142 | 1.105 | 1.143 | 1.125 | 1.183 | 1.227 | 1.165 | 1.192 | 0.629 | 0 |
| | MAE | 0.416 | 0.464 | 0.487 | 0.512 | 0.470 | 0.347 | 0.401 | 0.428 | 0.457 | 0.408 | 0.398 | 0.457 | 0.498 | 0.538 | 0.473 | 0.277 | 0.323 | 0.364 | 0.437 | 0.350 | 0.213 | 0.265 | 0.328 | 0.356 | 0.285 | 0.477 | 0.503 | 0.512 | 0.512 | 0.500 | 0.520 | 0.524 | 0.526 | 0.523 | 0.426 | |
| Chronos$_b$ | MSE | 0.526 | 0.585 | 0.593 | 0.608 | 0.578 | 0.322 | 0.411 | 0.440 | 0.443 | 0.404 | 0.422 | 0.523 | 0.600 | 0.686 | 0.558 | 0.206 | 0.272 | 0.329 | 0.436 | 0.311 | 0.184 | 0.241 | 0.296 | 0.375 | 0.274 | 1.172 | 1.224 | 1.202 | 1.249 | 1.212 | 1.166 | 1.222 | 1.157 | 1.182 | 0.626 | 0 |
| | MAE | 0.413 | 0.447 | 0.461 | 0.491 | 0.453 | 0.343 | 0.392 | 0.422 | 0.437 | 0.399 | 0.374 | 0.434 | 0.477 | 0.520 | 0.451 | 0.270 | 0.316 | 0.354 | 0.420 | 0.340 | 0.206 | 0.256 | 0.298 | 0.353 | 0.278 | 0.473 | 0.505 | 0.502 | 0.505 | 0.496 | 0.515 | 0.517 | 0.508 | 0.513 | 0.415 | |
| Chronos$_l$ | MSE | 0.493 | 0.601 | 0.601 | 0.653 | 0.596 | 0.328 | 0.434 | 0.480 | 0.522 | 0.442 | 0.452 | 0.536 | 0.613 | 0.671 | 0.568 | 0.203 | 0.268 | 0.316 | 0.424 | 0.303 | 0.195 | 0.254 | 0.311 | 0.395 | 0.289 | 1.178 | 1.229 | 1.194 | 1.232 | 1.208 | 1.213 | 1.269 | 1.241 | 1.241 | 0.642 | 0 |
| | MAE | 0.400 | 0.449 | 0.475 | 0.503 | 0.457 | 0.348 | 0.407 | 0.448 | 0.482 | 0.421 | 0.379 | 0.430 | 0.470 | 0.512 | 0.448 | 0.263 | 0.309 | 0.342 | 0.408 | 0.331 | 0.211 | 0.264 | 0.305 | 0.360 | 0.285 | 0.481 | 0.508 | 0.499 | 0.499 | 0.497 | 0.523 | 0.534 | 0.544 | 0.534 | 0.420 | |
| Moirai$_s$ | MSE | 0.395 | 0.430 | 0.445 | 0.430 | 0.425 | 0.274 | 0.341 | 0.364 | 0.378 | 0.339 | 0.362 | 0.385 | 0.410 | 0.450 | 0.402 | 0.192 | 0.258 | 0.307 | 0.391 | 0.287 | 0.160 | 0.207 | 0.258 | 0.327 | 0.238 | 1.082 | 1.112 | 1.085 | 1.105 | 1.096 | 1.272 | 1.284 | 1.177 | 1.244 | 0.551 | 4 |
| | MAE | 0.404 | 0.427 | 0.436 | 0.445 | 0.428 | 0.324 | 0.369 | **0.386** | **0.405** | 0.371 | 0.360 | 0.379 | 0.399 | 0.422 | 0.390 | 0.263 | 0.308 | 0.340 | 0.397 | 0.327 | 0.194 | 0.238 | 0.275 | 0.324 | 0.258 | 0.479 | 0.493 | 0.491 | 0.493 | 0.489 | 0.560 | 0.562 | 0.535 | 0.552 | 0.397 | |
| Moirai$_b$ | MSE | 0.429 | 0.494 | 0.531 | 0.544 | 0.500 | 0.275 | 0.347 | 0.379 | 0.388 | 0.347 | 0.456 | 0.459 | 0.482 | 0.518 | 0.479 | 0.194 | 0.251 | 0.292 | 0.354 | 0.273 | 0.170 | 0.218 | 0.270 | 0.344 | 0.251 | 1.129 | 1.175 | 1.142 | 1.177 | 1.156 | 1.208 | 1.243 | 1.171 | 1.207 | 0.579 | 1 |
| | MAE | 0.412 | 0.447 | 0.467 | 0.501 | 0.457 | 0.319 | 0.365 | 0.388 | 0.408 | 0.370 | 0.385 | 0.400 | 0.415 | 0.435 | 0.409 | 0.266 | 0.306 | 0.333 | 0.375 | 0.320 | 0.203 | 0.249 | 0.288 | 0.339 | 0.270 | 0.472 | 0.495 | 0.485 | 0.485 | 0.484 | 0.557 | 0.550 | 0.538 | 0.548 | 0.403 | |
| Moirai$_l$ | MSE | 0.464 | 0.523 | 0.546 | 0.549 | 0.521 | 0.298 | 0.403 | 0.464 | 0.530 | 0.424 | 0.522 | 0.541 | 0.563 | 0.611 | 0.559 | 0.195 | 0.257 | 0.308 | 0.395 | 0.289 | 0.170 | 0.222 | 0.277 | 0.361 | 0.258 | 1.102 | 1.140 | 1.102 | 1.122 | 1.117 | 1.213 | 1.252 | 1.179 | 1.215 | 0.604 | 0 |
| | MAE | 0.433 | 0.470 | 0.486 | 0.511 | 0.475 | 0.346 | 0.412 | 0.448 | 0.498 | 0.426 | 0.401 | 0.423 | 0.442 | 0.465 | 0.433 | 0.267 | 0.311 | 0.342 | 0.395 | 0.329 | 0.203 | 0.253 | 0.293 | 0.347 | 0.274 | 0.464 | 0.486 | 0.484 | 0.486 | 0.480 | 0.541 | 0.543 | 0.534 | 0.539 | 0.418 | |
| TimesFM-2.0 | MSE | 0.424 | 0.444 | 0.443 | 0.438 | 0.437 | 0.256 | 0.328 | 0.366 | 0.391 | 0.335 | 0.362 | 0.410 | 0.429 | 0.470 | 0.418 | 0.175 | 0.239 | 0.294 | 0.369 | 0.269 | 0.167 | 0.213 | 0.266 | 0.340 | 0.247 | 1.099 | 1.106 | 1.058 | 1.088 | 1.088 | 1.277 | 1.448 | 3.971 | 2.232 | - | 0 |
| | MAE | 0.397 | 0.417 | 0.424 | 0.442 | 0.420 | 0.308 | 0.361 | 0.389 | 0.413 | 0.368 | 0.340 | 0.373 | 0.395 | 0.424 | 0.383 | 0.245 | 0.294 | 0.332 | 0.385 | 0.314 | 0.221 | 0.264 | 0.304 | 0.354 | 0.286 | 0.449 | 0.483 | 0.492 | 0.522 | 0.487 | 0.649 | 0.724 | 1.362 | 0.912 | - | |
| Timer-XL | MSE | 0.394 | 0.436 | 0.469 | 0.493 | 0.425 | 0.273 | 0.342 | 0.377 | 0.382 | 0.344 | 0.325 | 0.375 | 0.414 | 0.512 | 0.406 | 0.187 | 0.240 | 0.285 | 0.392 | 0.272 | 0.160 | 0.210 | 0.274 | 0.418 | 0.247 | 1.043 | 1.126 | 1.127 | 1.127 | 1.106 | 1.112 | 1.128 | 1.043 | 1.094 | 0.537 | 0 |
| | MAE | 0.395 | 0.419 | 0.437 | 0.457 | 0.425 | 0.336 | 0.380 | 0.402 | 0.416 | 0.384 | 0.354 | 0.385 | 0.409 | 0.441 | 0.401 | 0.272 | 0.310 | 0.338 | 0.392 | 0.328 | 0.214 | 0.260 | 0.309 | 0.405 | 0.286 | 0.574 | 0.612 | 0.615 | 0.606 | 0.602 | 0.626 | 0.620 | 0.594 | 0.613 | 0.427 | |
| Time-MoE$_b$ | MSE | 0.357 | 0.384 | 0.411 | 0.449 | 0.400 | 0.305 | 0.351 | 0.391 | 0.419 | 0.367 | 0.338 | 0.355 | 0.381 | 0.504 | 0.394 | 0.201 | 0.258 | 0.324 | 0.488 | 0.318 | 0.159 | 0.215 | 0.291 | 0.415 | 0.266 | - | - | - | - | - | - | - | - | - | - | 3 |
| | MAE | 0.384 | 0.404 | 0.411 | 0.449 | 0.394 | 0.359 | 0.386 | 0.418 | 0.464 | 0.404 | 0.368 | 0.388 | 0.413 | 0.493 | 0.404 | 0.291 | 0.334 | 0.373 | 0.464 | 0.366 | 0.213 | 0.266 | 0.322 | 0.405 | 0.297 | - | - | - | - | - | - | - | - | - | - | |
| Time-MoE$_l$ | MSE | 0.350 | 0.388 | 0.411 | 0.427 | 0.394 | 0.302 | 0.364 | 0.417 | 0.537 | 0.405 | 0.309 | 0.346 | 0.373 | 0.475 | 0.376 | 0.197 | 0.250 | 0.337 | 0.480 | 0.316 | 0.167 | 0.215 | 0.291 | 0.400 | 0.270 | - | - | - | - | - | - | - | - | - | - | 2 |
| | MAE | 0.382 | 0.412 | 0.430 | 0.427 | 0.420 | 0.354 | 0.385 | 0.425 | 0.496 | 0.415 | 0.357 | 0.381 | 0.408 | 0.477 | 0.406 | 0.286 | 0.322 | 0.375 | 0.461 | 0.361 | 0.214 | 0.266 | 0.322 | 0.400 | 0.300 | - | - | - | - | - | - | - | - | - | - | |
| TTM$_a$ | MSE | 0.373 | 0.396 | 0.406 | 0.411 | 0.397 | 0.262 | 0.326 | 0.354 | 0.376 | 0.330 | 0.329 | 0.364 | 0.381 | 0.413 | 0.372 | 0.167 | 0.227 | 0.277 | 0.360 | 0.258 | 0.146 | 0.193 | 0.249 | 0.328 | 0.229 | 0.929 | 0.926 | 0.904 | 0.939 | 0.925 | 1.114 | 1.134 | 1.051 | 1.156 | 0.494 | 11 |
| | MAE | 0.397 | 0.415 | 0.424 | 0.441 | 0.419 | 0.334 | 0.378 | 0.400 | 0.421 | 0.383 | 0.350 | 0.374 | 0.390 | 0.410 | 0.381 | 0.254 | 0.296 | 0.329 | 0.382 | 0.315 | 0.181 | 0.229 | 0.284 | 0.336 | 0.265 | 0.511 | 0.518 | 0.523 | 0.541 | 0.523 | 0.628 | 0.628 | 0.608 | 0.621 | 0.408 | |
| ChronosBolt$_s$ | MSE | 0.405 | 0.475 | 0.514 | 0.516 | 0.478 | 0.273 | 0.339 | 0.388 | 0.393 | 0.346 | 0.357 | 0.400 | 0.420 | 0.512 | 0.404 | 0.166 | 0.231 | 0.289 | 0.400 | 0.272 | 0.152 | 0.205 | 0.264 | 0.357 | 0.245 | 0.863 | 0.956 | 1.001 | 1.103 | 0.981 | 0.762 | 0.802 | 0.749 | 0.771 | 0.489 | 3 |
| | MAE | 0.388 | 0.426 | 0.453 | 0.485 | 0.438 | 0.333 | 0.362 | 0.398 | 0.408 | 0.371 | 0.378 | 0.418 | 0.389 | 0.441 | 0.376 | 0.237 | 0.284 | 0.324 | 0.393 | 0.310 | 0.186 | 0.237 | 0.279 | 0.338 | 0.260 | 0.410 | 0.458 | 0.492 | 0.551 | 0.478 | 0.433 | 0.446 | 0.473 | 0.451 | 0.381 | |
| ChronosBolt$_b$ | MSE | 0.420 | 0.486 | 0.517 | 0.493 | 0.479 | 0.255 | 0.333 | 0.381 | 0.393 | 0.341 | 0.303 | 0.370 | 0.410 | 0.497 | 0.395 | 0.164 | 0.226 | 0.278 | 0.382 | 0.262 | 0.150 | 0.197 | 0.255 | 0.329 | 0.237 | 0.905 | 0.959 | 0.957 | 1.049 | 0.968 | **0.752** | **0.791** | **0.725** | **0.756** | 0.483 | _14_ |
| | MAE | 0.388 | 0.424 | 0.445 | 0.457 | 0.429 | 0.305 | 0.355 | 0.389 | 0.408 | 0.364 | 0.311 | 0.352 | 0.382 | 0.428 | 0.368 | 0.232 | 0.280 | 0.323 | 0.391 | 0.307 | 0.183 | 0.230 | 0.272 | 0.329 | 0.254 | 0.417 | 0.439 | 0.441 | 0.467 | 0.441 | **0.423** | **0.437** | **0.462** | **0.441** | 0.369 | |
| KAIROS$_m$ | MSE | 0.386 | 0.436 | 0.460 | 0.426 | 0.427 | 0.269 | 0.340 | 0.380 | 0.394 | 0.335 | 0.304 | 0.358 | 0.383 | 0.408 | 0.363 | 0.168 | 0.228 | 0.279 | 0.366 | 0.260 | 0.144 | 0.193 | 0.254 | 0.353 | 0.236 | 0.976 | 0.997 | 0.982 | 1.015 | 0.993 | 0.789 | 0.840 | 0.788 | 0.806 | 0.477 | 4 |
| | MAE | 0.384 | 0.411 | 0.430 | 0.422 | 0.411 | 0.320 | 0.360 | 0.391 | 0.416 | 0.369 | 0.329 | 0.362 | 0.384 | 0.407 | 0.371 | 0.246 | 0.290 | 0.323 | 0.378 | 0.309 | 0.181 | 0.229 | 0.273 | 0.331 | 0.254 | 0.424 | 0.439 | 0.443 | 0.450 | 0.439 | 0.437 | 0.444 | 0.436 | 0.436 | 0.367 | |
| KAIROS$_s$ | MSE | 0.385 | 0.429 | 0.446 | 0.422 | 0.421 | 0.276 | 0.353 | 0.385 | 0.386 | 0.362 | 0.305 | 0.356 | 0.381 | 0.404 | 0.362 | 0.163 | 0.226 | 0.278 | 0.382 | 0.262 | 0.151 | 0.197 | 0.253 | 0.340 | 0.235 | 0.920 | 0.943 | 0.942 | 1.002 | 0.952 | 0.808 | 0.851 | 0.815 | 0.825 | 0.473 | 4 |
| | MAE | 0.383 | 0.408 | 0.418 | 0.424 | 0.408 | 0.319 | 0.369 | 0.394 | 0.416 | 0.368 | 0.326 | 0.360 | 0.382 | 0.404 | 0.368 | 0.238 | 0.284 | 0.320 | 0.384 | 0.307 | 0.186 | 0.232 | 0.274 | 0.329 | 0.255 | 0.410 | 0.426 | 0.426 | 0.442 | 0.427 | 0.447 | 0.446 | 0.429 | 0.441 | 0.366 | |
| KAIROS$_b$ | MSE | 0.396 | 0.434 | 0.452 | 0.426 | 0.427 | 0.276 | 0.353 | 0.385 | 0.386 | 0.350 | 0.295 | 0.336 | 0.367 | 0.393 | 0.348 | 0.160 | 0.220 | 0.271 | 0.373 | 0.248 | 0.146 | 0.192 | 0.248 | 0.338 | 0.231 | 0.927 | 0.955 | 0.945 | 0.989 | 0.954 | 0.804 | 0.843 | 0.815 | 0.821 | **0.471** | **28** |
| | MAE | 0.385 | 0.408 | 0.418 | 0.427 | 0.410 | 0.319 | 0.369 | 0.394 | 0.412 | 0.374 | 0.322 | 0.355 | 0.380 | 0.402 | 0.365 | 0.238 | 0.283 | 0.317 | 0.373 | 0.270 | 0.182 | 0.228 | 0.270 | 0.330 | 0.253 | 0.414 | 0.425 | 0.427 | 0.432 | 0.425 | 0.446 | 0.449 | 0.434 | 0.443 | **0.364** | |

## H   SHOWCASES

### H.1   SHOWCASES OF SYNTHETIC DATA

Figure 8 showcases representative examples generated by the algorithm using the synthetic dataset. As detailed in Appendix E.1, the generated synthetic dataset is classified into two categories. The first is a composite type, formed from a combination of seasonal, trend, and noise components, which is designated Custom in the figures. The second consists of idealized industrial signals, designated Perfect periodic.

### H.2   SHOWCASES OF KAIROS

We present several prediction examples generated by KAIROS during testing, as illustrated in Figure 9.

## I   BROADER IMPACTS

Our work on KAIROS is foundational research focused on advancing time series modeling. While KAIROS itself is not designed for direct societal applications with immediate negative impacts, any powerful predictive technology could be misused. So beyond general risks of advanced AI, our model has no specific negative societal impacts need to be discussed here.

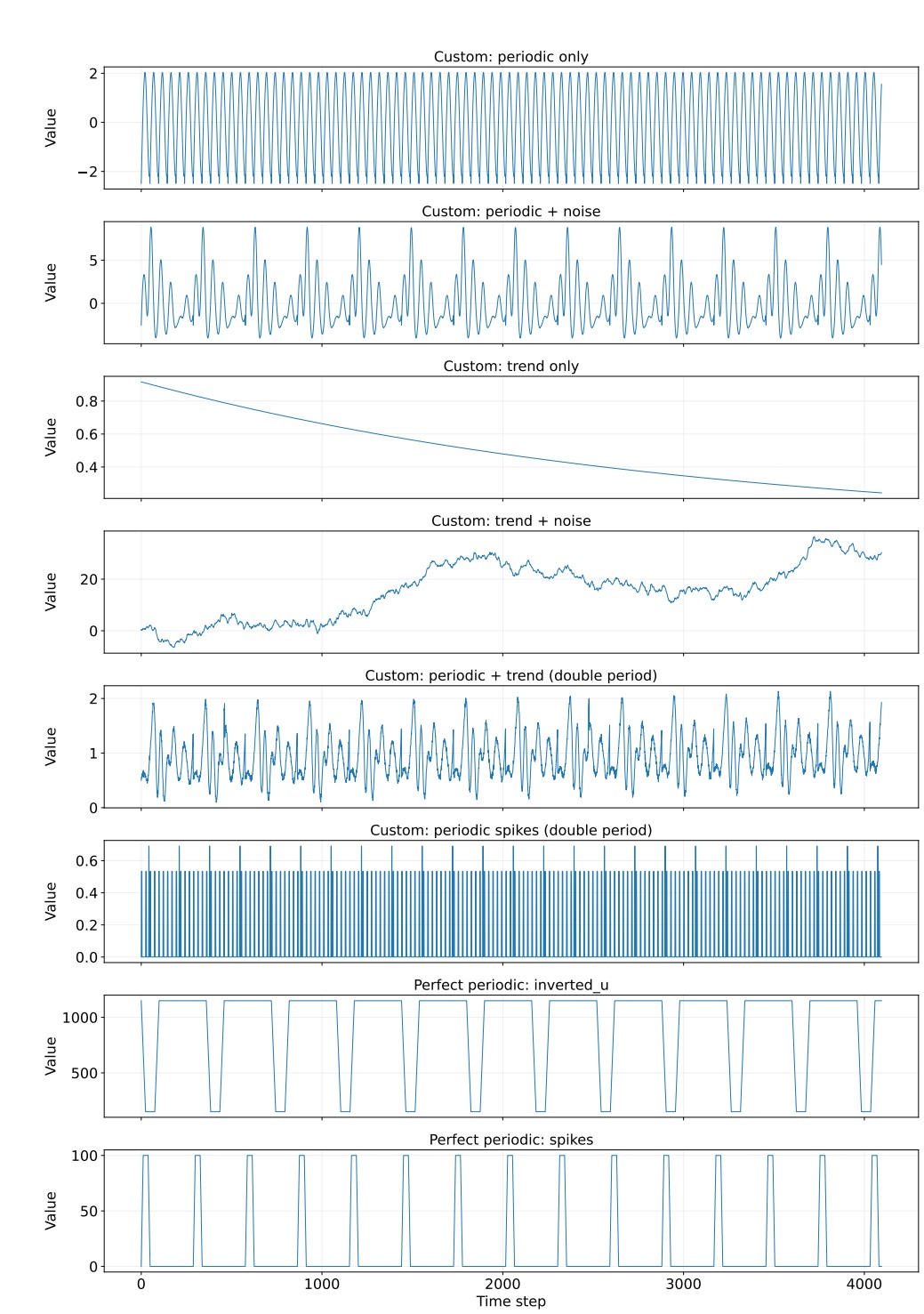

Figure 8: Several distinct case types are generated by the synthetic dataset algorithm. Specifically, the Custom designation refers to a composite signal type, which is systematically constructed by combining seasonal, trend, and noise components. Concurrently, the Perfect periodic designation denotes synthesized, idealized industrial signals.

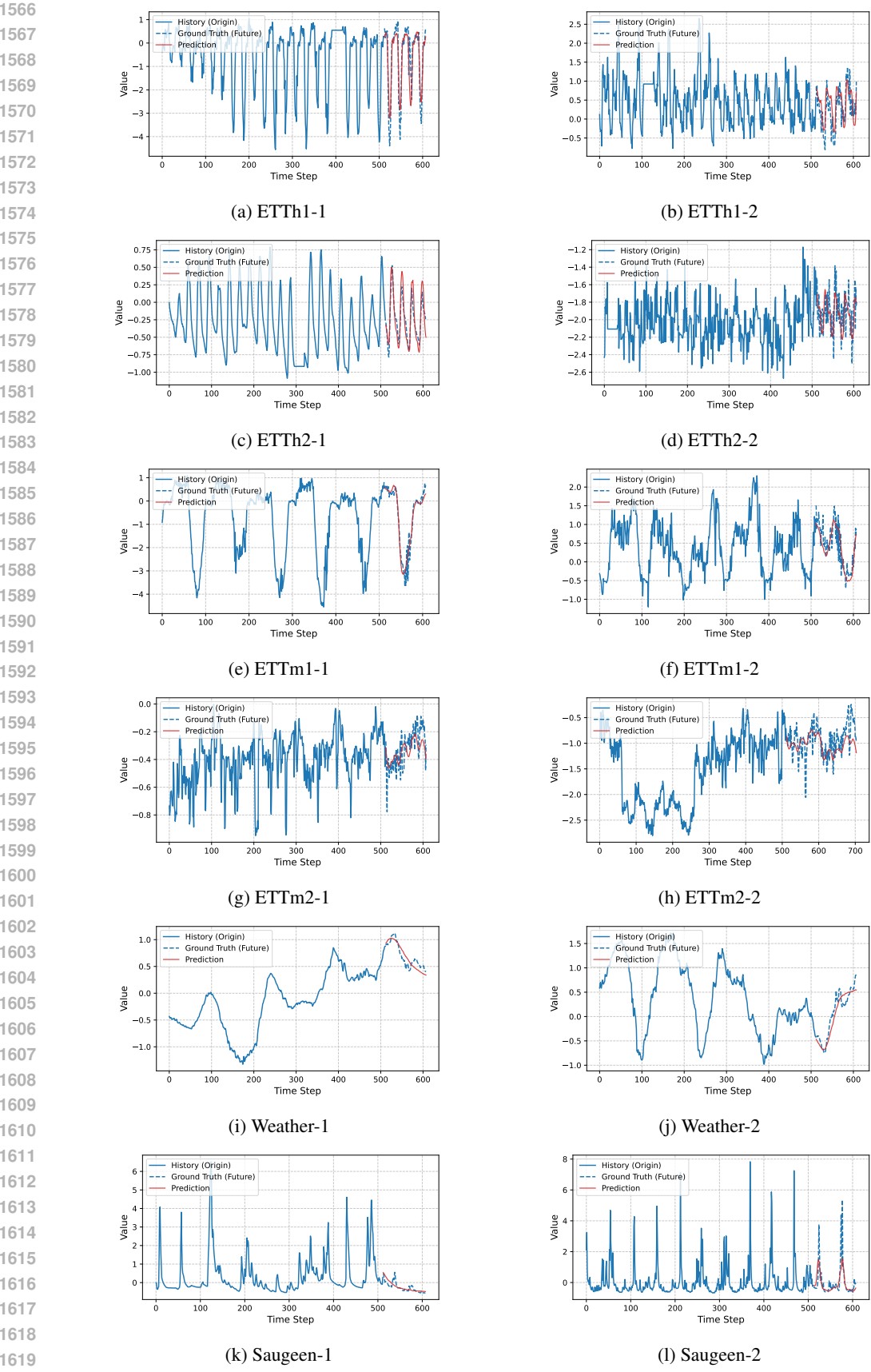

Figure 9: Example of forecasts from KAIROS$_b$ on the test datasets used in experiments.

