# OpenReview forum: "Kairos: Towards Adaptive and Generalizable Time Series Foundation Models"
_ICLR.cc/2026/Conference — Submitted to ICLR 2026_

### Official Review · Reviewer_nXjk · 2025-10-24

**Soundness:** 2
**Presentation:** 2
**Contribution:** 2
**Rating:** 4
**Confidence:** 4

**Summary:**

This paper proposes Kairos, a new Time Series Foundation Model designed to handle heterogeneous and dynamic time series data. It introduces two key components: Mixture-of-Size Dynamic Patching, which adaptively adjusts tokenization granularity, and Instance-Adaptive Rotary Position Embedding, which customizes positional encoding for each instance. Kairos also adopts a multi-patch prediction strategy and leverages a large PreSTS corpus (300B points) for pretraining. Experiments on GIFT-Eval and TSLib show strong zero-shot forecasting performance with fewer parameters than existing models.

**Strengths:**

1. The introduction of adaptive mechanisms, such as dynamic patching and instance-specific positional encoding, represents an impressive and well-motivated approach to address several key limitations present in existing time-series foundation models.

2. The model exhibits consistently strong zero-shot forecasting performance across a diverse set of benchmarks, demonstrating its robustness and generalizability to various time-series tasks.

3. Remarkably, it achieves competitive—and in some cases even superior—results, all while using a substantially smaller number of parameters compared to existing models, highlighting its efficiency and scalability.

**Weaknesses:**

1. The use of 𝑃 = {32,64,128}, k=3 and z=2 indicates that the proposed dynamic tokenization strategy is not genuinely flexible. In practice, the method behaves very similarly to a fixed tokenization with P=32, where different patches are merely assigned different weights. While the approach may appear more sophisticated at first glance, its operations are ultimately limited to combining a small set of predefined patch sizes, which constrains its adaptability and overall novelty.  A truly dynamic tokenization should produce patches of arbitrary lengths. Ideally, it should generate very long patches in smooth regions and patches whose lengths are roughly comparable to the fluctuation period in regions with rapid changes, rather than being limited to a few predefined lengths such as 32, 64, or 128.

2. The function and purpose of the null expert remain unclear. Specifically, it is not evident what contributions it makes to the model’s performance, and no ablation study has been presented to quantify its effects or justify its inclusion.

3. The proposed IARoPE method lacks in-depth theoretical analysis or thorough empirical investigation that would clarify why it performs effectively. As a result, the source of its improvement and the underlying intuition behind its success are insufficiently explained, leaving readers without a strong understanding of its operational principles.

4. While the experiments provide a comparison between Kairos and Sundial on the GIFT-Eval benchmark, they omit a corresponding evaluation on the TSLib benchmark. This absence of broader benchmarking reduces the strength of the empirical validation and makes it harder to assess the generalizability of the proposed method across diverse datasets.

**Questions:**

1. Is the null expert’s s‘n initialized randomly, or is there a specific initialization strategy that influences its behavior?

2. Could the FFT-based scaling used in IARoPE become unstable when applied to noisy or non-stationary signals? Additionally, have you conducted any experiments to assess its robustness in the presence of outliers or spectral distortions?

3. How computationally demanding is the MoS-DP routing and fusion process in comparison to fixed-patch baselines?

---

> ### Author Response · Authors · 2025-11-21
> **Response to Reviewer nXjk (Part 1 / 3)**
>
> > W1: The use of 𝑃 = {32,64,128}, k=3 and z=2 indicates that the proposed dynamic tokenization strategy is not genuinely flexible. In practice, the method behaves very similarly to a fixed tokenization with P=32, where different patches are merely assigned different weights. While the approach may appear more sophisticated at first glance, its operations are ultimately limited to combining a small set of predefined patch sizes, which constrains its adaptability and overall novelty. A truly dynamic tokenization should produce patches of arbitrary lengths. Ideally, it should generate very long patches in smooth regions and patches whose lengths are roughly comparable to the fluctuation period in regions with rapid changes, rather than being limited to a few predefined lengths such as 32, 64, or 128.
>
> We appreciate the reviewer’s thoughtful comment on the design of our dynamic tokenization.
>
> We would like to clarify two points: (i) **why our method is more than a fixed tokenization with (P=32)**, and (ii) **why we deliberately use a small set of patch sizes instead of arbitrary lengths**. We also add new experiments to support this choice.
>
> **First, our method is not equivalent to a fixed (P=32) tokenizer with learned weights.** Due to the presence of null experts, even when the number of activated experts $K=3$, the model does not always activate all the real experts. For instance, the model may activate an expert of 128-patch size granularity, an expert of 64-patch size granularity, and a null expert. In this case, the coarsest patch (128-patch size) will be split into smaller segments of 64-patch size, with the information from only these two granularities being fused. Thus, the existence of null experts provides the model with the flexibility to **activate any combination of granularities**, rather than always splitting at the 32-patch size granularity.
>
> Thus, the total number of tokens for the sequence is **variable**, with the upper limit set by partitioning the entire sequence at the 32-patch size (assuming the 32-granularity expert is activated at every routing), and the lower limit set by partitioning only at the 128-patch size (where only the 128-granularity real expert is activated at every routing).
>
> **Second, the use of a finite, predefined set of patch sizes is a deliberate and necessary tradeoff between the ideal flexibility you described and practical, real-world viability.** A truly dynamic tokenization that produces arbitrary-length patches is an interesting research direction. However, it introduces severe practical challenges:
>
> * It would be extremely difficult to optimize and train effectively.
> * The computational overhead would be prohibitively high, which is unacceptable for inference in real-world applications.
>
> To see whether adding more patch options helps in practice, we extended the patch set to include finer granularities:
>
> - Setting 1: $P=\\{16,32,64,128\\}$ (with $K=3$,$Z=2$)
> - Setting 2: $P=\\{8,16,32,64,128\\}$ (with $K=3$,$Z=2$)
>
> As illustrated in the table below, on GIFT-Eval (normalized MASE), we observe that adding more patch sizes brings **no overall gain**: a likely reason is that very small patches make the model overly sensitive to high-frequency noise, which hurts generalization. Given this, the additional flexibility does not translate into better performance, while it clearly harms efficiency.
>
> **In summary, our original choice is flexible enough to let the model adapt its effective patch length and fuse multi-scale information.** The new ablations suggest that simply adding more patch sizes moves us away from this well-balanced configuration, both in accuracy and in latency.
>
> | Task (Norm. MASE)  | $P=\\{32,64,128\\}$ (Ours) | $P=\\{16,32,64,128\\}$ | $P=\\{8,16,32,64,128\\}$ |
> | :----------------- | :----------------------: | :------------------: | :--------------------: |
> | **Short**          |           0.719          |       **0.717**      |          0.725         |
> | **Medium**         |         **0.759**        |         0.772        |          0.789         |
> | **Long**           |         **0.789**        |         0.817        |          0.845         |
> | **Average**        |         **0.742**        |         0.749        |          0.763         |
> | **Inference Time** |        **0.065s**        |    0.074s (+13.8%)   |     0.099s (+52.3%)    |

---

> ### Author Response · Authors · 2025-11-21
> **Response to Reviewer nXjk (Part 2 / 3)**
>
> > W2: The function and purpose of the null expert remain unclear. Specifically, it is not evident what contributions it makes to the model’s performance, and no ablation study has been presented to quantify its effects or justify its inclusion.
>
> Thank you for this valuable suggestion.
>
> **The purpose of the null expert is to fundamentally enhance the flexibility and adaptive capacity of the MoS-DP module.** Without the null expert, the number of active real experts is fixed at $K$ for each patch. By adding the null expert, we remove this rigid constraint, allowing the model to activate a variable number of real experts. This means the model can freely choose to use only the coarsest granularity, only the finest granularity, or a combination of multiple granularities to model a specific patch.
>
> **To demonstrate the impact of the null expert, we have conducted a new ablation study comparing the full Kairos model against a variant without the null expert.** As shown in the table below, we evaluated this on the GIFT-Eval benchmark using normalized MASE. The results clearly show that **removing the null expert causes a significant performance degradation due to this loss of adaptability, especially on the more challenging long-term forecasting horizons.**
>
> | Task (Norm. MASE) | w/ Null Experts | w/o Null Experts |
> | --- | --- | --- |
> | Short | **0.719** | 0.720 |
> | Medium | **0.759** | 0.774 |
> | Long | **0.789** | 0.836 |
> | Average | **0.742** | 0.755 |
>
> ---
> > W3: The proposed IARoPE method lacks in-depth theoretical analysis or thorough empirical investigation that would clarify why it performs effectively. As a result, the source of its improvement and the underlying intuition behind its success are insufficiently explained, leaving readers without a strong understanding of its operational principles.
>
> Thank you for highlighting the lack of explanation. **In the revised version, we have added a theoretical analysis (Appendix D.4).** Summary as follows:
>
> **(1) RoPE as frequency-selective kernels over relative positions.**
> Theorem 1 in Appendix D.4.1 derives the attention score under RoPE at the level of each 2D subspace. For the $j$-th pair of dimensions and positions $(m, n)$, we show that the contribution of this subspace to the dot-product attention can be written as (Eq. 28):
> $$
> \mathrm{atten}\_j(m, n) = A\_j(m, n) \cdot \cos\big(\theta\_j (m-n) + \varphi\_j(m, n)\big),
> $$
> where $\theta\_j$ is the RoPE angular frequency for this subspace, and $A\_j$, $\varphi\_j$ depend on the content vectors. Thus, for each $j$, RoPE induces a **cosine-shaped kernel over the relative distance** $(m-n)$, whose frequency is controlled by $\theta\_j$. The full attention score is a superposition of such sinusoidal kernels across all $j$ and heads. In this sense, RoPE imposes a **frequency-selective bias over relative lags**, with different $\theta\_j$ emphasizing different temporal ranges.
>
> **(2) IARoPE effectively aligns rotation frequencies with instance-specific patterns**
>
> From the analysis above, **the base RoPE parameters $\\{\theta\_{\text{init},j}\\}$ define a fixed bank of sinusoidal weighting functions over relative positions, shared by all sequences.** This global frequency grid can be suboptimal in a TSFM setting where different series have very different spectral characteristics.
>
> IARoPE makes these frequencies instance-adaptive in a way consistent with Eq. 5 in the main text. For each input sequence $x$, we compute low-frequency FFT features and feed them into a small MLP to obtain modulation parameters $\gamma\_j(x)$ and $\beta\_j(x)$, which adjust the base RoPE frequencies via
> $$
> \theta'\_j(x) = \gamma\_j(x) \cdot \theta\_{\text{init},j} + \beta\_j(x).
> $$
> The resulting $\theta'\_j(x)$ still parameterize sinusoidal kernels over relative lags, so the relative-position nature of RoPE is preserved. However, the **effective rotation frequencies are now adapted to the current input sequence**, providing a sequence-dependent RoPE frequency profile rather than a fixed one.
>
> We hope that the added derivations in Appendix D.4 now make the operational principles and effectiveness of IARoPE much clearer.

---

> ### Author Response · Authors · 2025-11-21
> **Response to Reviewer nXjk (Part 3 / 3)**
>
> > W4: While the experiments provide a comparison between Kairos and Sundial on the GIFT-Eval benchmark, they omit a corresponding evaluation on the TSLib benchmark. This absence of broader benchmarking reduces the strength of the empirical validation and makes it harder to assess the generalizability of the proposed method across diverse datasets.
>
> Thank you for this suggestion. We have conducted the corresponding zero-shot forecasting evaluation of Kairos and Sundial [1] on the TSLib benchmark.
>
> The results are presented in the table below, reporting the average MAE across forecast lengths of {96, 192, 336, 720}. **As the results demonstrate, Kairos consistently and comprehensively outperforms Sundial on the TSLib benchmark as well, even while having significantly fewer parameters.**
>
> | Dataset | Kairos (50M) | Sundial (128M) |
> | --- | --- | --- |
> | ETTh1 | **0.410** | 0.434 |
> | ETTh2 | **0.374** | 0.387 |
> | ETTm1 | **0.365** | 0.377 |
> | ETTm2 | **0.303** | 0.320 |
> | weather |**0.253** | 0.270 |
> | Average | **0.341** | 0.358 |
>
> ---
> > Q1: Is the null expert’s s‘n initialized randomly, or is there a specific initialization strategy that influences its behavior?
>
> We follow the initialization setup for null experts used in Large Language Models [2, 3]. The logits $s^{\prime}\_n$ for the null expert are initialized randomly, using the exact same strategy as the real experts. The sole distinction is functional: the null expert performs no computation when it is activated by the router.
>
> ---
> > Q2: Could the FFT-based scaling used in IARoPE become unstable when applied to noisy or non-stationary signals? Additionally, have you conducted any experiments to assess its robustness in the presence of outliers or spectral distortions?
>
> Thank you for raising this question about the stability of the FFT-based scaling in IARoPE under noisy or non-stationary signals.
>
> As detailed in Appendix D.4.2 and Algorithm 1, our adaptation is computed in the frequency domain and then **normalized by a LayerNorm over FFT features** (step 5). Concretely, for each sequence we compute the amplitude spectrum, truncate to the lower frequencies, and apply LayerNorm along the frequency dimension before feeding it to the MLP that produces $\left(\gamma,\, \beta \right)$. This means the modulation depends mainly on the relative shape of the spectrum rather than on its absolute scale, so sporadic spikes or outliers in a few frequency bins cannot dominate the adaptive scaling of RoPE frequencies.
>
> To empirically test robustness, we inject synthetic anomalies into the GIFT-Eval benchmark at different rates: for each series, we randomly corrupt $p$ % of time steps ($p$ % $\in$ \{1%,2%,5%\}) by adding a large perturbation ($x\_t \leftarrow x\_t + k\cdot\mathrm{std}(x)$) with $k\in\\{-5,5\\}$, and then evaluate zero-shot forecasting with RoPE and IARoPE. The aggregated MASE results are:
>
> | Anomaly Percentage |    IARoPE |  RoPE |
> | :----------------- | ---------:|-----: |
> | 0%                 | **0.742** | 0.759 |
> | 1%                 | **0.779** | 0.788 |
> | 2%                 | **0.804** | 0.809 |
> | 5%                 | **0.874** | 0.879 |
>
> As expected, both methods degrade as the anomaly rate increases, but **IARoPE consistently outperforms RoPE at every noise level**, and its performance drop is never larger than that of RoPE. Together with the normalization design above, these results indicate that the FFT-based scaling in IARoPE is **stable and robust** even in the presence of outliers and spectral distortions.
>
> ---
> > Q3: How computationally demanding is the MoS-DP routing and fusion process in comparison to fixed-patch baselines?
>
> To quantify the computational cost, we conducted an ablation study comparing the full Kairos model against a variant with the MoS-DP routing and fusion process removed.
>
> The experiment was run on a single NVIDIA TITAN RTX GPU with an input sequence length of 2048 and a prediction horizon of 96. **As shown in the table below, removing the MoS-DP process reduces inference time by approximately 18.5%. However, this gain in speed comes at the cost of a severe degradation in forecasting performance, as measured by the normalized MASE on GIFT-Eval.**
>
> | Task | w/ MoS-DP | w/o MoS-DP |
> | --- | --- | --- |
> | Inference Time (s) | 0.065 | **0.053** |
> | Short (MASE) | **0.719** | 0.725 |
> | Medium (MASE) | **0.759** | 0.779 |
> | Long (MASE) | **0.789** | 0.813 |
> | Average (MASE) | **0.742** | 0.755 |
>
>
> ---
>
> [1] Sundial: A Family of Highly Capable Time Series Foundation Models (ICML 2025)
>
> [2] AdaMoE: Token-Adaptive Routing with Null Experts for Mixture-of-Experts Language Models (EMNLP 2024)
>
> [3] MoE++: Accelerating Mixture-of-Experts Methods with Zero-Computation Experts (ICLR 2025)

---

> > ### Comment · Reviewer_nXjk · 2025-11-27
> >
> > We sincerely appreciate the authors’ responses and the additional experiments. After reviewing all reviewers’ comments and the authors’ replies, I still have significant concerns regarding the proposed method.
> >
> > 1. As I previously mentioned, essentially $e_{n,m} = f(x_{n,m}, ..., x_{n, Mn})$, which is not fundamentally different from simply assigning weights to different patches, the range of the patches remains limited. The authors’ further experiments are even more surprising: when refining the set of patch choices, adding smaller patches actually decreases performance. If the proposed method were truly effective, adding smaller patches would create a superset of the original method, offering more flexibility, but the results instead degrade. Moreover, performance further drops when including patch size 8. This suggests that the proposed method cannot adaptively select the truly optimal patch size and is essentially performing a simple fitting rather than genuine adaptive tokenization.
> >
> > 2. According to the authors’ results, MoS-DP increases inference time by 18% while improving performance by only 1.7%. This trade-off does not appear particularly favorable. Directly adopting approaches such as Pathformer and training multiple tokenizers might yield greater improvements with higher efficiency.
> >
> > 3. Since tokenization is the most critical component of the paper, the substantial issues with this method make the paper entirely unacceptable.

---

> > > ### Author Response · Authors · 2025-11-29
> > > **Response to Reviewer nXjk (Part 1 / 2)**
> > >
> > > Thank you for your reply. For your remaining concerns, we provide the following response:
> > >
> > > > As I previously mentioned, essentially $e_{n,m} = f(x_{n,m}, ..., x_{n, Mn})$, which is not fundamentally different from simply assigning weights to different patches, the range of the patches remains limited.
> > >
> > > Thank you for your valuable feedback. We would like to clarify the differences between Kairos and previous work [1, 2] in terms of patching, fusion, and flexibility:
> > >
> > > 1. **Different Patching Strategy**: Previous works typically tokenize the entire sequence using several different patch sizes and then fuse these representations. This ignores the local heterogeneity within the sequence. Kairos, in contrast, first blocks the series with the coarsest granularity and then **dynamically and adaptively selects the most appropriate finest granularity within that local block**. This means Kairos can use **different patch granularities in different regions of the same time series** (e.g., coarse for smooth sections, fine for abrupt changes), allowing each token to represent a **different effective time-step length**.
> > > 2. **Different Fusion Mechanism**: Unlike the simple multi-scale fusion in prior work, each token in Kairos is generated via a hierarchical, multi-scale weighted sum, **fusing features from all of its activated ancestor patches**. This means the specific fusion weights are unique to each patch, enabling a sophisticated adaptation to the local information density, rather than a single, global weighting scheme.
> > > 3. **Superior Flexibility**: Previous methods typically fuse a fixed number of granularities. By incorporating the null experts, Kairos removes this constraint. The number of activated granularities for each patch is **dynamic**. This allows the model to **choose any combination of granularities (from just one to all of them) for any given patch**, effectively balancing computational load and performance.
> > >
> > > **As we demonstrated in Section 4.4.1, Kairos learns this adaptive behavior: it assigns coarser granularities to simpler regions and finer granularities to complex, fluctuating regions.** This confirms that our mechanism is far more than a simple re-weighting of fixed patches; **it is a more flexible and adaptive modeling approach suitable for a foundation model.**
> > >
> > > > The authors’ further experiments are even more surprising: when refining the set of patch choices, adding smaller patches actually decreases performance. If the proposed method were truly effective, adding smaller patches would create a superset of the original method, offering more flexibility, but the results instead degrade. Moreover, performance further drops when including patch size 8. This suggests that the proposed method cannot adaptively select the truly optimal patch size and is essentially performing a simple fitting rather than genuine adaptive tokenization.
> > >
> > > Thank you for your thoughtful observation. We will explain below why increasing the patch size choices can worsen performance, and whether the model can adaptively select the truly optimal patch size.
> > >
> > > 1. **Increasing the search space does not necessarily guarantee improved performance.** The current selection of patch sizes represents **a trade-off between optimization difficulty and performance**. Adding a patch size of 8 increases the model's optimization complexity. Furthermore, smaller patch sizes are highly sensitive to noise, and the features extracted by such experts may overfit high-frequency noise rather than capturing meaningful local patterns.
> > > 2. As demonstrated in Section 4.4.1, **Kairos assigns larger patch sizes to smoother regions and smaller patch sizes to more complex, dynamic regions**. This clearly shows Kairos has an **adaptive capability to varying information densities**, enabling it to model complex temporal dynamics at the appropriate granularity.

---

> > > > ### Author Response · Authors · 2025-11-29
> > > > **Response to Reviewer nXjk (Part 2 / 2)**
> > > >
> > > > > According to the authors’ results, MoS-DP increases inference time by 18% while improving performance by only 1.7%. This trade-off does not appear particularly favorable. Directly adopting approaches such as Pathformer and training multiple tokenizers might yield greater improvements with higher efficiency.
> > > >
> > > > Thank you for this sharp observation.
> > > >
> > > > 1. **The improvement in the metric we report is challenging to achieve.** The metric we use is the normalized MASE, which is normalized by the Seasonal Naïve baseline across the full 97-task GIFT-Eval benchmark [3]. This ensures that the results are statistically significant and robust. **As shown in Figure 7 of Sundial [4], increasing the number of sampling steps from 5 to 50 results in a theoretical 10-fold increase in the forward pass time, but only a 0.95% performance gain.** This highlights the inherent difficulty of achieving significant improvements in this metric.
> > > > 2. **As detailed in Appendix C, the inference speed of Kairos is competitive compared to other time-series models.** The 18% increase in inference time does *not* notably affect the practical efficiency of Kairos in real-world applications.
> > > > 3. **Regarding the Pathformer approach, its tokenizer method does not outperform Kairos.** In our experiment in Section 4.2, Kairos's zero-shot performance significantly surpasses the full-shot performance of Pathformer.
> > > > 4. **Kairos has a major parameter advantage.** Compared to other large models like Sundial (128M) and Toto (151M) [5], Kairos (10M, 23M, 50M) is exceptionally lightweight. This makes Kairos far easier to deploy and use in practical applications.
> > > >
> > > > Thank you again for your valuable feedback. We hope this explanation resolves your concerns. Should you have any further questions, we would be happy to discuss them.
> > > >
> > > > ---
> > > > [1] Pathformer: Multi-scale Transformers with Adaptive Pathways for Time Series Forecasting (ICLR 2024)
> > > >
> > > > [2] ElasTST: Towards Robust Varied-Horizon Forecasting with Elastic Time-Series Transformer (NeurIPS 2024)
> > > >
> > > > [3] GIFT-Eval: A Benchmark For General Time Series Forecasting Model Evaluation
> > > >
> > > > [4] Sundial: A Family of Highly Capable Time Series Foundation Models (ICML 2025)
> > > >
> > > > [5] This Time is Different: An Observability Perspective on Time Series Foundation Models (NeurIPS 2025)

---

### Official Review · Reviewer_WLGx · 2025-10-28

**Soundness:** 2
**Presentation:** 2
**Contribution:** 2
**Rating:** 4
**Confidence:** 4

**Summary:**

In this study, the authors investigate the tokenisation of time series in the context of forecasting. In particular, they propose (a) Mixture-of-Size Dynamic Patching (MoS-DP) to obtain patches of varying size, depending on the information density of the time series, and (b) Instance-adaptive Rotary Position Embedding (IARoPE) to adequatly encode such patches of varying size with positional information. The authors evaluate their method across two benchmarks and conduct an ablation study to assess the importance of MoS-DP and IARoPE.

**Strengths:**

1) The paper is well structured.
2) The authors evaluate their method on two established forecasting benchmarks.
3) The authors conduct ablation studies to provide insights on the effectiveness of the proposed components.
4) The authors provide their code to support reproducibility.

**Weaknesses:**

Related Works:

1) The authors do not discuss the most recent works in the field of time series tokenisation. For instance, works on domain-specific tokenisation [1] and wavelet-based tokenisation [2] should be included to provide a representative overview of the current literature.

Terminology:

2) The term 'general-purpose time series foundation model (TSFM)' (see line 039) is misleading, as the proposed model is only able to perform forecasting. This is also the case for the title. A more suitable term in the context of this study would be 'foundational forecasting model', since general-purpose time series models such as MOMENT [3] are able to perform classification and regression in addition to forecasting.

Methodology:

3) The authors measure the information density of a time series via spectral entropy. They claim a direct relationship between spectral entropy (SE) and information density (ID), i.e. low SE indicating low ID and high SE indicating high ID, as suggested in Figure 1.b. However, I believe the relationship between ID and SE to be inverse. For example, the ID of a pure sine wave should be higher than the ID of a sine wave overlaid with white noise, which is not the case in Figure 1.b.
4) Furthermore, the ID of a time series is strongly coupled to the sampling frequency (SF). For instance, ETTm1 and ETTh1 are recordings of the exact same underlying signal, however, ETTm1 has a higher ID since it is recorded at a higher SF. While an interesting approach, the usage of spectral entropy as a measure of information density might not be the best fit.
5) The overview of the proposed method in Figure 2 is not self-containing and thus hard to follow for the reader.
6) The proposed method is designed for univariate time series analysis and thus cannot model dependencies between variates. However, previous works [1] have shown that inter-variate dependencies are critical in time series analysis and should be considered during the tokenisation process.

Experiments:

7) The authors compare the proposed tokeniser against several baselines, however, they do not include recent works in the field of time series tokenisation. Particularly, OTIS [1] and WaveToken [2] should be included as baselines to ensure a fair comparison of the proposed method.
8) The authors argue that the proposed multi-patch prediction strategy 'effectively mitigates the cumulative errors inherent in autoregressive precesses' (see ll. 289-299). However, in Figure 6 (appendix C) they show that predicting only the next two patches achieves best performance. In light of this, the authors seem to overstate the importance of the multi-patch prediction strategy.
9) The authors ablate the proposed components MoS-DP and IARoPE in Section 4.3. While their work primarily focuses on how "fixed-size patching enforce rigid observational granularity, limiting their ability to adapt to varying information densities" (see ll 17-19), the results presented in the last two rows of Table 2 suggest otherwise. Specifically, the proposed dynamic patching tokeniser MoS-DP yields only marginal improvements of 0.68%, 0.13%, and 0.12% across short-, medium-, and long-term prediction horizons, respectively. This modest gain raises the question of whether fixed-size patching truly constitutes a limitation of current methods in the first place.
10) The authors analyse the proposed IARoPE in Section 4.4.2 and show that Intra-Dataset shuffle performs better than the original RoPE, which seems to be counter-intuitive and requires further elaboration.

Discussion:

11) The discussion of the experiments is limited. The experiments section only provides numbers without their interpretation, while the conclusion section only provides a brief summary of the study.
12) The authors do not discuss the limitations of the study in the main manuscript.
13) The authors do not discuss topics relevant for future work in the main manuscript.

___
[1] Turgut, Ö. et al. "Towards generalisable time series understanding across domains." arXiv preprint arXiv:2410.07299 (2024).

[2] Masserano, L. et al. "Enhancing Foundation Models for Time Series Forecasting via Wavelet-based Tokenization." ICML (2025).

[3] Goswami, M. et al. "MOMENT: A Family of Open Time-series Foundation Models." ICML (2024).

**Questions:**

1) Could the authors visualise examples of the synthetic data included in the pre-training corpus?
2) How do the authors determine the perfect balance between real-world and synthetic data used for pre-training?
3) What is the advantage of infusing synthetic data into the pre-training corpus? The authors state that 'PreSTS comprises real-world time series spanning multiple domains and is augmented by a synthetic dataset [...] to ensure comprehensive coverage' (see ll. 320-321). However, the authors do not explain how 'comprehensive coverage' is defined or measured in this context.
4) Could the authors evaluate their method against multi-variate approaches such as [1]? This would enable a comparison of different tokenisation approaches that address the problem of varying temporal dependencies across domains. For instance in [1], the authors introduce domain-specific positional embeddings for variates and show that these embeddings can successfully capture temporal dependencies (see Figure 11).
5) Have the authors experimented with any other measure than spectral entropy to measure the information density of a time series?
___
[1] Turgut, Ö. et al. "Towards generalisable time series understanding across domains." arXiv preprint arXiv:2410.07299 (2024).

---

> ### Author Response · Authors · 2025-11-21
> **Response to Reviewer WLGx (Part 1 / 5)**
>
> > W1: The authors do not discuss the most recent works in the field of time series tokenisation. For instance, works on domain-specific tokenisation [1] and wavelet-based tokenisation [2] should be included to provide a representative overview of the current literature.
>
> Thank you for this valuable suggestion. We have added a discussion of these important works on domain-specific [1] and wavelet-based [2] tokenization to our related work section in the revised paper.
>
> ---
> > W2: The term 'general-purpose time series foundation model (TSFM)' (see line 039) is misleading, as the proposed model is only able to perform forecasting. This is also the case for the title. A more suitable term in the context of this study would be 'foundational forecasting model', since general-purpose time series models such as MOMENT [3] are able to perform classification and regression in addition to forecasting.
>
> Thank you for your insightful comment on this important terminological point. We apologize for any confusion.
>
> 1. **Our use of the term "general-purpose time series foundation model" was intended to align with previous works (e.g., TimesFM [4], Sundial [5], Toto [6])**, which also introduced "Time Series Foundation Models" while focusing specifically on the forecasting task. Our title and terminology follow this established convention.
> 2. **We respectfully believe that forecasting represents the most fundamental benchmark for a time series foundation model.** This is because forecasting is inherently a generative task that closely mirrors the sequential and autoregressive nature of time series data. Successful forecasting requires the model to learn underlying patterns, trends, and seasonality, which we consider to be the core capabilities of a foundational model.
>
> While we agree that models like MOMENT, which also perform classification and regression, represent a valuable direction for general-purpose models, we believe our in-depth forecasting evaluation provides a strong and sufficient validation of our model's core contributions. We consider the extension to these other tasks as valuable future work.
>
> ---
> > W3: The authors measure the information density of a time series via spectral entropy. They claim a direct relationship between spectral entropy (SE) and information density (ID), i.e. low SE indicating low ID and high SE indicating high ID, as suggested in Figure 1.b. However, I believe the relationship between ID and SE to be inverse. For example, the ID of a pure sine wave should be higher than the ID of a sine wave overlaid with white noise, which is not the case in Figure 1.b.
>
> We sincerely apologize for the terminological ambiguity that caused this confusion. **We have clarified our definition of information density in the revised paper.**
>
> The term Information Density can be interpreted in two ways:
> 1. The density of useful semantic signal (where noise has low density).
> 2. The quantity of information entropy (where noise has high density).
>
> **In our work, we adopt the latter definition rooted in Information Theory.** Specifically, we use Spectral Entropy (SE) to measure the uncertainty of the time series data.
>
> Additionally, **our goal in this section is not just to focus on the specific information density values but to highlight that even within the same dataset, the variation in information density can be significant.** This motivates the need for finer-grained modeling of such varying information densities, which is why we introduced the MoS-DP module to achieve multi-scale information fusion at the patch level.

---

> ### Author Response · Authors · 2025-11-21
> **Response to Reviewer WLGx (Part 2 / 5)**
>
> > W4: Furthermore, the ID of a time series is strongly coupled to the sampling frequency (SF). For instance, ETTm1 and ETTh1 are recordings of the exact same underlying signal, however, ETTm1 has a higher ID since it is recorded at a higher SF. While an interesting approach, the usage of spectral entropy as a measure of information density might not be the best fit.
>
> Thank you for this insightful observation. Indeed, this is a **reasonable** phenomenon. Since we calculate the spectral entropy (information density) for each window using a sliding window approach with **a fixed window size of 128**, **when the sampling frequency differs, each window will correspond to a different time span**. Therefore, the spectral entropy values associated with different sampling frequencies are ultimately determined by the post-sampling time series in each fixed window.
>
> In fact, this is exactly the issue that motivates our design: **with fixed-size patches, changing the sampling frequency automatically changes how much real time each patch actually covers.** Kairos addresses this by dynamically fusing multiple patch sizes, so it can capture patterns across different temporal scales instead of being tied to a single patch size.
>
> We also clarify that spectral entropy is only used in Figure 1 as a descriptive and motivational tool: it illustrates that even within a single dataset, local spectral complexity varies significantly, which motivates modeling heterogeneous information densities. **Our Kairos model does not use spectral entropy at training or inference time (it is not an input, target, or regularizer), and all components, including MoS-DP and IARoPE, are learned end-to-end from the raw time series.**
>
> ---
>
> > W5: The overview of the proposed method in Figure 2 is not self-containing and thus hard to follow for the reader.
>
> We apologize for any confusion. Figure 2 is intended as a high-level overview of Kairos that complements the methodological description in Section 3 and its caption, rather than as a fully standalone explanation of all details. We agree that additional guidance can make the figure easier to follow. In the revised paper, we have (i) **recolored several components to reduce visual clutter**, and (ii) **expanded the caption to more clearly explain the roles of MoS-DP and the semantics of the color coding** (e.g., the distinction between selected and unselected patches/experts). We hope that, with these changes, readers can now follow the overall pipeline more easily when viewing Figure 2 together with the main text.
>
> ---
> > W6: The proposed method is designed for univariate time series analysis and thus cannot model dependencies between variates. However, previous works [1] have shown that inter-variate dependencies are critical in time series analysis and should be considered during the tokenisation process.
>
> Thank you for your valuable suggestion. **We fully agree that modeling inter-variate dependencies is a valuable research direction and have already discussed this point in the limitations section of Appendix B in our original manuscript.** Based on your suggestion, we have now **moved this discussion to the main paper's Conclusion for greater visibility**.
>
> We are certainly considering the extension to multivariate modeling as a key area for our future work. However, we would like to emphasize that **this is an additional improvement and is orthogonal to the current focus of our work.**

---

> ### Author Response · Authors · 2025-11-21
> **Response to Reviewer WLGx (Part 3 / 5)**
>
> > W7: The authors compare the proposed tokeniser against several baselines, however, they do not include recent works in the field of time series tokenisation. Particularly, OTIS [1] and WaveToken [2] should be included as baselines to ensure a fair comparison of the proposed method.
>
> > Q4. Could the authors evaluate their method against multi-variate approaches such as [1]? This would enable a comparison of different tokenisation approaches that address the problem of varying temporal dependencies across domains. For instance in [1], the authors introduce domain-specific positional embeddings for variates and show that these embeddings can successfully capture temporal dependencies (see Figure 11).
>
> Thank you for suggesting these highly relevant works. **We have conducted fair comparisons on the respective benchmarks for both OTIS [1] and WaveToken [2]**, with the results shown in the tables below.
>
> 1. For OTIS, We followed the protocol from the OTIS paper, which involves finetuning on each specific dataset's training set and evaluating the 96-step forecast MSE. **Kairos converges in far fewer than one epoch and significantly outperforms OTIS on the majority of datasets in terms of prediction performance.**
> 2. For WaveToken, since the authors did not provide open-source code, we were unable to test on our benchmark. Instead, we used their in-domain benchmark for MASE evaluation and applied the same evaluation methodology to ensure fairness. **The results show that Kairos consistently outperforms WaveToken in forecasting performance, even though Kairos has far fewer parameters.** Additionally, we must emphasize that the forecasting length in this benchmark is relatively short, with the maximum being only 48 time steps. In contrast, **in the more challenging GIFT-Eval benchmark, Kairos achieves a normalized MASE of 0.742, far surpassing Chronos-Large’s 0.870. Notably, Chronos-Large demonstrated stronger performance than WaveToken on this benchmark, further highlighting Kairos's performance advantage.**
>
> Furthermore, we greatly appreciate OTIS's approach to modeling multivariate time series, and we plan to explore modeling inter-variate relationships in **our future work**. However, we believe this is **orthogonal** to the current focus of our work.
>
> **Table 1: Comparison between Kairos and OTIS**
> | Datasets | Kairos (50M) | OTIS-Base (8M) | OTIS-Large (44M) | OTIS-Huge (131M) |
> | --- | --- | --- | --- | --- |
> | ETTh1 | **0.365** | 0.424 | 0.446 | 0.461 |
> | ETTh2 | 0.255 | 0.212 | **0.205** | 0.215 |
> | ETTm1 | **0.247** | 0.337 | 0.362 | 0.384 |
> | ETTm2 | **0.148** | 0.161 | 0.173 | 0.181 |
> | Weather | **0.137** | 0.139 | 0.142 | 0.149 |
> | AVG | **0.230** | 0.255 | 0.266 | 0.278 |
>
> **Table 2: Comparison between Kairos and WaveToken**
> | Datasets | Kairos (50M) | WaveToken-mini (19.2M) | WaveToken-small (44.5M) | WaveToken-base (199M) | WaveToken-large (705.8M) |
> | --- | --- | --- | --- | --- | --- |
> | Electricity (Weekly) | **1.535** | 1.938 | 1.890 | 1.879 | 1.864 |
> | KDD Cup 2018 | **0.606** | 0.618 | 0.624 | 0.631 | 0.654 |
> | London Smart Meters | **0.740** | 0.771 | 0.759 | 0.748 | **0.740** |
> | Taxi (30 Min.) | **0.777** | 0.902 | 0.861 | 0.848 | 0.823 |
> | Uber TLC (Daily) | **0.760** | 0.952 | 0.938 | 0.887 | 0.876 |
> | AVG | **0.884** | 1.036 | 1.014 | 0.999 | 0.991 |
>
> ---
> > W8: The authors argue that the proposed multi-patch prediction strategy 'effectively mitigates the cumulative errors inherent in autoregressive processes' (see ll. 289-299). However, in Figure 6 (appendix C) they show that predicting only the next two patches achieves best performance. In light of this, the authors seem to overstate the importance of the multi-patch prediction strategy.
>
> Thank you for this sharp observation. We appreciate the opportunity to clarify our claim.
>
> 1. **Predicting two patches instead of one already leads to a substantial difference, both in effective forecast length and in performance.** Concretely, when each patch corresponds to 64 time steps, moving from one to two patches doubles the forecast horizon from 64 to 128, and we observe a clear reduction in cumulative error and improved performance, especially for the most challenging long-term forecasting setting (MASE from 0.835 to 0.789).
> 2. **The optimal number of decoded patches is a model-specific trade-off.** Decoding too many patches at once can increase the training optimization difficulty and introduce additional noise, which may ultimately harm performance. For Kairos, the optimal balance happens at two patches, but for other models this optimal number may differ.
>
> Therefore, the fact that we choose to decode two patches per step does *not* diminish the importance of the proposed multi-patch prediction strategy. Rather, it shows that **our method offers a simple and effective way to mitigate cumulative errors in autoregressive decoding for time series foundation models, while still leaving room for further exploration of this design space**.

---

> ### Author Response · Authors · 2025-11-21
> **Response to Reviewer WLGx (Part 4 / 5)**
>
> > W9: The authors ablate the proposed components MoS-DP and IARoPE in Section 4.3. While their work primarily focuses on how "fixed-size patching enforce rigid observational granularity, limiting their ability to adapt to varying information densities" (see ll 17-19), the results presented in the last two rows of Table 2 suggest otherwise. Specifically, the proposed dynamic patching tokeniser MoS-DP yields only marginal improvements of 0.68%, 0.13%, and 0.12% across short-, medium-, and long-term prediction horizons, respectively. This modest gain raises the question of whether fixed-size patching truly constitutes a limitation of current methods in the first place.
>
> Thank you for this sharp observation.
>
> **MoS-DP and IARoPE are complementary components.** MoS-DP and IARoPE are responsible for adaptive modeling at the Intra-Patch and Inter-Patch levels, respectively. **However, RoPE applies the same positional information across completely different, heterogeneous time series, which cannot fully capture the adaptive time series information density tokens generated by MoS-DP.** As a result, the improvement from MoS-DP is more modest when RoPE is used. When IARoPE is used for positional encoding, the improvement from MoS-DP becomes much more significant. For example, **in the most challenging long-term forecasting task, the normalized MASE improves from 0.813 to 0.789.**
>
> Additionally, the metric used is the geometric mean of the MASE values across 97 tasks, normalized by the Seasonal Naïve baseline. This makes the metric **robust and statistically significant**. Even small improvements in this metric indicate a **stable enhancement in performance**.
>
> ---
> > W10: The authors analyse the proposed IARoPE in Section 4.4.2 and show that Intra-Dataset shuffle performs better than the original RoPE, which seems to be counter-intuitive and requires further elaboration.
>
> Thank you for pointing this out. **We agree that this result may appear counter-intuitive at first glance, but it is in fact reasonable.** For many datasets, different samples within the same dataset (and the same channel) tend to share very similar spectral characteristics. As a result, under the Intra-Dataset shuffle setting, the modulated RoPE frequencies $\theta$ assigned to each sample are still quite similar to those it would have received without shuffling, and are overall more suitable than the fixed, non-adaptive frequencies used in the original RoPE. **In other words, Intra-Dataset shuffle still preserves a form of dataset-level adaptivity, even though it loses true instance-level adaptation, so its performance lying between Original RoPE and full IARoPE is actually consistent with our expectations.**
>
> ---
> > W11: The discussion of the experiments is limited. The experiments section only provides numbers without their interpretation, while the conclusion section only provides a brief summary of the study.
> W12: The authors do not discuss the limitations of the study in the main manuscript.
> W13: The authors do not discuss topics relevant for future work in the main manuscript.
>
> We appreciate these valuable suggestions. We acknowledge that due to strict space constraints in the initial submission, our discussion was condensed. In the revised version, we have addressed these points as follows:
>
> 1. **We have enriched the interpretation of our results.** Specifically, we provide a more detailed analysis in the IARoPE shuffle experiment (as discussed in our response to W10).
> 2. **We have explicitly included a discussion on the limitations of the current study and outlined promising directions for future research in the Conclusion section.**
>
> ---
>
> > Q1. Could the authors visualise examples of the synthetic data included in the pre-training corpus?
>
> Thank you for this suggestion. We have added sample visualizations of our synthetic data in Appendix H.1 of the revised paper.
>
> ---
> > Q2. How do the authors determine the perfect balance between real-world and synthetic data used for pre-training?
>
> Thank you for this question. We did not specifically adjust the ratio between real-world and synthetic data used for pre-training. Instead, we followed the ratio settings used in TimesFM [4].

---

> ### Author Response · Authors · 2025-11-21
> **Response to Reviewer WLGx (Part 5 / 5)**
>
> > Q3. What is the advantage of infusing synthetic data into the pre-training corpus? The authors state that 'PreSTS comprises real-world time series spanning multiple domains and is augmented by a synthetic dataset [...] to ensure comprehensive coverage' (see ll. 320-321). However, the authors do not explain how 'comprehensive coverage' is defined or measured in this context.
>
> Thank you for this valuable suggestion.
>
> Similar to Chronos [7], TimesFM [4], and Sundial [5], we use synthetic data to **supplement** the training corpus in order to **enhance the diversity of the training data**.
>
> To illustrate this, we follow the approach of Toto [6] by calculating the statistical features of both the real-world and synthetic datasets in the training set and comparing their distributions. **We present these comparisons in Appendix E.1 of the revised paper**, where we show that the synthetic dataset effectively complements the model by covering distributions not seen in the real-world datasets.
>
> ---
> > Q5. Have the authors experimented with any other measure than spectral entropy to measure the information density of a time series?
>
> This is a valuable suggestion. **We use the Shannon entropy of the power spectral density calculated over a sliding window as a measure of information density. However, we would like to emphasize that even within the same dataset, information density can vary significantly.** We demonstrate this by computing the standard deviation of information density across different windows. To address this, we proposed the MoS-DP module to adapt to varying information densities within the sequence.
>
> In concurrent work [8], the authors use the **variance of the amplitude spectrum** over a sliding window to measure pattern heterogeneity. In the table below, we compare these two approaches and show the relative magnitudes of information density differences across datasets. **Our results are generally consistent with theirs, which validates that our metric is both reasonable and robust.**
>
> | Datasets | Standard Deviation of Spectral Entropy (Ours) | Variance of Amplitude Spectrum |
> | --- | --- | --- |
> | ETTh1 | 0.2704 | 103.4527 |
> | ETTh2 | 0.4974 | 163.9889 |
> | ETTm1 | 0.3524 | 144.2234 |
> | ETTm2 | 0.5512 | 221.6870 |
> | Weather | 0.4362 | 188.3436 |
> | Saugeen (D) | 0.3539 | 96.9169 |
> | Saugeen (W) | 0.1189 | 17.7301 |
> | Pure Sine Wave | 0.0502 | 0.0688 |
> | Sine Shifting to Noise | 1.1637 | 261.7420 |
>
> ---
>
> [1] Towards generalisable time series understanding across domains
>
> [2] Enhancing Foundation Models for Time Series Forecasting via Wavelet-based Tokenization (ICML 2025)
>
> [3] MOMENT: A Family of Open Time-series Foundation Models (ICML 2024)
>
> [4] A decoder-only foundation model for time-series forecasting (ICML 2024)
>
> [5] Sundial: A Family of Highly Capable Time Series Foundation Models (ICML 2025)
>
> [6] This Time is Different: An Observability Perspective on Time Series Foundation Models (NeurIPS 2025)
>
> [7] Chronos: Learning the Language of Time Series (TMLR 2024)
>
> [8] Accuracy Law for the Future of Deep Time Series Forecasting

---

> ### Comment · Reviewer_WLGx · 2025-11-26
> **Reviewer's Response**
>
> I would like to thank the authors for their responses. I acknowledge that I have read the rebuttal and the responses to the other reviewers. While an interesting stream of research, I remain unconvinced and believe the proposed study is not ready for publication. For instance, the authors appear to overstate the general applicability of the model. In the referenced works, the generalisability is clearly defined: „general purpose zero-shot forecaster“ in TimesFM (page 3), „general purpose time series forecasting“ in TOTO (abstract), while the term general-purpose is not even claimed in Sundial. More importantly, the roles and interpretations of key proposed metrics, specifically information density and spectral entropy, are still not clear. Given such concerns, I cannot recommend accepting the study at this time, and thus leave my scores unchanged. However, I strongly encourage the authors to further improve the quality of the study, as the conceptual idea is interesting.

---

> > ### Author Response · Authors · 2025-11-26
> > **Response to Reviewer WLGx**
> >
> > Thank you for your reply. We will address your concerns in detail below.
> >
> > > The authors appear to overstate the general applicability of the model. In the referenced works, the generalisability is clearly defined: „general purpose zero-shot forecaster“ in TimesFM (page 3), „general purpose time series forecasting“ in TOTO (abstract), while the term general-purpose is not even claimed in Sundial.
> >
> > We apologize for any confusion caused. **We have not intended to claim that Kairos is a "general-purpose time series foundation model."** Regarding the sentence you mentioned in line 039 of the original paper: "*More recently, inspired by the success of large-scale foundation models (FMs), researchers have turned to building general-purpose time series foundation models (TSFMs)*," we are referring to the general-purpose time series foundation model as a research direction that researchers are moving towards. This is the **only instance** in the paper where "general-purpose" is mentioned. **To avoid confusion, we have revised this sentence in the updated paper.** The term "Generalizable Time Series Foundation Models" that we emphasize in the title, similar to TimesFM [1], Sundial [2], and TOTO [3], refers to the ability to perform zero-shot forecasting tasks across different domains.
> >
> > > The roles and interpretations of key proposed metrics, specifically information density and spectral entropy, are still not clear.
> >
> > Thank you for pointing this out.
> >
> > 1. **To make our motivation clearer, we have clarified the definitions and roles of information density and spectral entropy in the revised paper.** Spectral entropy is a commonly used measure of uncertainty and complexity [4], and the information density defined from this metric reflects the signal complexity. Additionally, we have provided detailed calculation procedures in Appendix G.
> > 2. **Kairos does not use information density and spectral entropy as inputs, targets, or regularizers during training or inference**; instead, they are learned end-to-end from the **raw** time series. **Information density and spectral entropy, calculated using a sliding window, are used only in Figure 1 as tools to illustrate the motivation**, which demonstrates that even within a single dataset, local spectral complexity varies significantly. This observation motivates the modeling of heterogeneous information densities.
> >
> > We hope this explanation resolves your concerns. Should you have any further questions, we would be happy to discuss them.
> >
> > ---
> >
> > [1] A decoder-only foundation model for time-series forecasting (ICML 2024)
> >
> > [2] Sundial: A Family of Highly Capable Time Series Foundation Models (ICML 2025)
> >
> > [3] This Time is Different: An Observability Perspective on Time Series Foundation Models (NeurIPS 2025)
> >
> > [4] TimeMixer: Decomposable Multiscale Mixing for Time Series Forecasting (ICLR 2024)

---

### Official Review · Reviewer_6umn · 2025-10-31

**Soundness:** 3
**Presentation:** 2
**Contribution:** 2
**Rating:** 2
**Confidence:** 4

**Summary:**

The paper presents a time series foundation model (TSFM) that introduces an adaptive input embedding / tokenization scheme, called MoS-DP. Moreover, the authors introduce a modified positional embedding, called IARoPE, based on the well-known RoPE scheme. The authors then demonstrate their network performance on common time series leaderboards, such as GIFT-Eval and TSLib.

**Strengths:**

The paper addresses a timely problem of tokenization in the time series domain. Most architectures try to overtake input embedding schemes from the NLP domain, e.g., patching techniques, but it is well-known in this field that this formulation might be limited and not ideal. Thus the efforts to develop new approaches is important. Dynamically combining patches to form differently-sized patches sounds reasonable.

Also, the approach from the authors to enhance the RoPE is a good idea and especially important for their transformer backbone.

I appreciate the evaluation on standard leaderboard benchmarks in this field, where their model shows competitive performance.

**Weaknesses:**

The description of the MoS-DP feels **overly complicated and counter-intuitive**:\
The authors mention that the aim is to "yield the embedding for each finest patch by fusing
information from multiple granularities". However, if that is the goal, the more intuitive way would be to actually start from the fines resolution, i.e., the individual time steps and then fuse more and more information together to form larger patches eventually. It is unclear how splitting the initial coarse patch into finer patches would "fuse information from multiple granularities". Furthermore, the approach that the authors took also requires the user to choose the largest granularity p_S, which may not be easy as it already requires good knowledge about the time series at hand. The illustration in Figure 4 is not quite clear. It would help to overlay directly the selected size of the finest patch p_k for the regions. This way, the reader should see immediately that smaller patches are considered by the network in "interesting" areas, such as the ones with high fluctuation or rapid changes. Furthermore, the description of the patch fusion and the Figure 2 illustrating it feel overly complicated. In Figure 2, the information flow is from the bottom to the top. However, in the dynamic patch fusion phase, the coarse patch is broken up into smaller pieces, hence shouldn't the larger green patch be at the bottom and then the splits be towards the top? Also, the coloring scheme is confusing, both across the entire figure (inappropriate color reusing) and within the patch fusion as well (Expert 1 is not even selected for the patch on the right and Expert 1 in green is not chosen in both of the show patches, but highlighted on top).

The adjustment of the RoPE idea to be specific to the current instance sounds reasonable and the idea to use the core frequency components of the time series to do that also makes sense. However, the specific strength of the original RoPE approach was to be able to generalize to seamlessly generalize to various sequence lengths. It feels that with the IARoPE, this crucial benefit may be lost? Why is so much emphasis put on RoPE then, instead of using a custom position embedding based on the components that the authors extract from the time series?

The literature review part is very weak and not well written:\
Since the paper focuses so much on the input tokenization, there are several papers left unmentioned / undiscussed. For example, the PatchTST architecture, which introduces the central component of this paper, the patches, in the first place is not discussed. Other works, that also employ dynamic patching, such as the TTM, is just mentioned on passing and in a different context even. Finally, recent works that employ non-transformer-based architectures and thus don't necessarily require patching of positional encodings, such as TiRex or FlowState, are not discussed at all.

[1] Nie, Yuqi, et al. "A Time Series is Worth 64 Words: Long-term Forecasting with Transformers." arXiv, 27 Nov. 2022, doi:10.48550/arXiv.2211.14730.
[2] Ekambaram, Vijay, et al. "Tiny Time Mixers (TTMs): Fast Pre-trained Models for Enhanced Zero/Few-Shot Forecasting of Multivariate Time Series." arXiv, 8 Jan. 2024, doi:10.48550/arXiv.2401.03955.
[3] Auer, Andreas, et al. "TiRex: Zero-Shot Forecasting Across Long and Short Horizons with Enhanced In-Context Learning." arXiv, 29 May. 2025, doi:10.48550/arXiv.2505.23719.
[4] Graf, Lars, et al. "FlowState: Sampling Rate Invariant Time Series Forecasting." arXiv, 7 Aug. 2025, doi:10.48550/arXiv.2508.05287.

KAIROS does not appear to be a zero-shot model for the GIFT-Eval:\
Although the authors claim to "have excluded all test sets found within the GIFT-Eval benchmark", from the list of datasets used in Appendix F.1, this seems to be not true at all. For example, Loop Seattle, M_DENSE, Solar, KDD Cup 2018, M4 Hourly, and several other datasets are clearly marked as being part of the GIFT-Eval benchmark in Table 13 of [5] and are not within the "allowed" Pretraining datasets listed in Table 14 of said paper. This can make a huge difference.

[5] Aksu, Taha, et al. "GIFT-Eval: A Benchmark For General Time Series Forecasting Model Evaluation." arXiv, 14 Oct. 2024, doi:10.48550/arXiv.2410.10393.

**Questions:**

Regarding the MoS-DP:
Why was the approach chosen to start from the coarsest granularity and go to the finest and not from the finest to the coarsest?\
Does it work to choose the p_S to be the entire time series?\
Please revise Figure 2. The coloring scheme is confusing and the illustration of the MoS-DP may not be accurate.

Regarding the IARoPE:
Can the IARoPE still generalize to varying sequence lengths?

Regarding GIFT-Eval:
Please clarify whether indeed the datasets in Table 5 have been used for pretraining and if so, that the KAIROS model is NOT a zero-shot model.

---

> ### Author Response · Authors · 2025-11-21
> **Response to Reviewer 6umn (Part 1 / 6)**
>
> > W1: The description of the MoS-DP feels overly complicated and counter-intuitive: The authors mention that the aim is to "yield the embedding for each finest patch by fusing information from multiple granularities". However, if that is the goal, the more intuitive way would be to actually start from the fines resolution, i.e., the individual time steps and then fuse more and more information together to form larger patches eventually. It is unclear how splitting the initial coarse patch into finer patches would "fuse information from multiple granularities".
>
> >Q1: Regarding the MoS-DP: Why was the approach chosen to start from the coarsest granularity and go to the finest and not from the finest to the coarsest?
>
> Thank you for this thoughtful question. We clarify that, **although Figure 2 starts from a coarsest partition for implementation convenience, the tokens that enter the Transformer are always defined at the finest granularity and each token fuses information from multiple scales**. Concretely, for the $n$-th coarsest patch we first let the router select a set of active experts (patch sizes) and determine the finest patch size $p\_k = \min\\{p\_i \mid g\_{n,i}>0\\}$. This coarsest patch is then split into finest patches $\\{x^{p\_k}\_{n,m}\\}\_{m=1}^{M\_n}$, and each finest patch embedding$$e\_{n,m}=\sum\_{i=k}^S \alpha\_{n,i},\mathrm{MLP}\_i\big(\mathcal{A}(x^{p\_k}\_{n,m},p\_i)\big)$$is a weighted sum of the representations of its ancestors at larger patch sizes $p\_i$ (Eq. 3, Appendix D.3.2 Definition of the ancestor function). **In other words, the final token representation is obtained by fusing information from coarser granularities around each finest patch, which is conceptually equivalent to the “fine-to-coarse fusion” you suggested.** We choose to implement this fusion by first patchifying at the coarsest granularity and then going to the finest mainly for **efficiency and simplicity**:
> - **Efficiency**: Routing is performed once per coarsest patch, rather than for every time step or finest patch. For a sequence of length $T$, this reduces the routing cost from $O(T)$ to $O(\lceil T/p\_S\rceil)$ (where $p\_S$ is the largest patch size), and all splits into finest patches and ancestor lookups inside each coarsest patch can be computed in parallel with simple batched operations.
> - **Simplicity**: Starting from the coarsest level also avoids maintaining an explicit bottom-up hierarchical merging structure over all time steps. Once the router has chosen experts for a coarsest patch, its finest patches and their ancestors at other scales are determined deterministically and can be implemented with straightforward indexing.
>
> Thus, MoS-DP effectively realizes “finest-patch embeddings enriched by multi-granularity context”, **while using the coarsest partition only as a convenient unit for parallel routing and implementation**. A fully bottom-up variant that starts from point-level tokens and merges them into arbitrary-length patches is indeed an interesting, orthogonal alternative, which we view as promising future work.
>
> ---
>
> > W2: Furthermore, the approach that the authors took also requires the user to choose the largest granularity $p\_S$, which may not be easy as it already requires good knowledge about the time series at hand.
>
> > Q2: Does it work to choose the $p\_S$ to be the entire time series?
>
> Thank you for your thoughtful question. **We believe that the parameter $p\_S$ is necessary.**
>
> 1. Even in the alternative scheme you described, starting from a single time step and progressively aggregating larger patches, one still needs to define **an upper bound on the largest granularity**, which corresponds to our $p\_S$.
> 2. In our design, $p\_S$ is set as a multiple of the finest patch size. While prior works suggest that the patch size is ideally aligned with the underlying periodicity [1], a general-purpose foundation model trained on vast, heterogeneous data and tested on unseen data with unknown periodicities. In this setting, exhaustively searching for a dataset-specific $p\_S$ or any patch-size–related hyperparameter is both computationally expensive and likely to yield unstable performance across scenarios. **Therefore, we did not specially search for this parameter; instead, we treat it as a default hyperparameter that works robustly in practice, without requiring any domain expertise from the user.**
>
> **Regarding Q2, we believe this would not work well in our framework.** In MoS-DP, routing is performed at each $p\_S$ block. If $p\_S$ were set to the full sequence length, the entire sequence would go through the router only once and be segmented using a single patch size with shared fusion weights. **This would force all positions in the sequence to share the same granularity, thereby losing our Kairos' ability to adapt to different information densities across regions of the time series, which is precisely the capability MoS-DP is designed to provide.**

---

> ### Author Response · Authors · 2025-11-21
> **Response to Reviewer 6umn (Part 2 / 6)**
>
> > W3: The illustration in Figure 4 is not quite clear. It would help to overlay directly the selected size of the finest patch $p\_k$ for the regions. This way, the reader should see immediately that smaller patches are considered by the network in "interesting" areas, such as the ones with high fluctuation or rapid changes.
>
> Thank you for this valuable suggestion to improve the clarity of Figure 4. To better convey our intention in Figure 4, **we have moved the color bar directly below the plot so that readers can more easily notice and interpret what the colors represent.**
>
> We respectfully choose to **keep the current weighted patch size visualization** rather than overlaying only the finest patch size $p\_k$. Our goal is to illustrate how different granularities contribute to the information fusion process. **Even when two regions select the same finest patch size, their weights over different granularities can differ substantially.** Highlighting only the finest patch size would fail to reflect these differences, whereas the current visualization **more accurately represents the final modeling granularity after weighting by different information densities**.
>
> ---
>
> > W4: Furthermore, the description of the patch fusion and the Figure 2 illustrating it feel overly complicated. In Figure 2, the information flow is from the bottom to the top. However, in the dynamic patch fusion phase, the coarse patch is broken up into smaller pieces, hence shouldn't the larger green patch be at the bottom and then the splits be towards the top?
>
> Thank you for highlighting the confusion regarding Figure 2. **The bottom-to-top layout reflects the computational logic of feature aggregation rather than a simple top-down splitting.** As described in Section 3.2, MoS-DP first identifies the finest active patch size $p\_k$ for a segment and then computes its embedding by fusing features from this finest scale with its coarser "ancestor" scales (Eq. 3). Therefore, placing small patches at the bottom correctly visualizes this fine-to-coarse hierarchical fusion: **the model starts from the finest token (bottom) and integrates contextual information from larger ancestor patches (top).**
>
> We hope the above clarification sufficiently resolves your concern.
>
> ---
> > W5: Also, the coloring scheme is confusing, both across the entire figure (inappropriate color reusing) and within the patch fusion as well (Expert 1 is not even selected for the patch on the right and Expert 1 in green is not chosen in both of the show patches, but highlighted on top).
>
> > Q3: Please revise Figure 2. The coloring scheme is confusing and the illustration of the MoS-DP may not be accurate.
>
> We apologize that the original figure had two potential sources of confusion and have clarified both in the revision. **We have revised both the figure and caption to clarify the coloring scheme.**
>
> 1. Expert 3 was previously drawn in a color that was very similar to the FFN block, which could give the impression that they belong to the same module. **In the updated figure, FFN now uses a clearly distinct color.**
> 2. On the right-hand side the same colors are reused in two panels: the upper "aggregate embeddings by weight" panel and the lower examples (cases 1 and 2).
>     - The upper panel serves as a schematic legend that ties each patch size to its corresponding expert (e.g., $p\_1$ Expert 1, $p\_2$ Expert 2, $p\_3$ Expert 3),
>     - The lower panel illustrates, for specific coarsest patches, which experts are actually selected by the router.
>
>     **In both panels, colors only encode the granularity type (patch size / expert pair).** Whether an expert is active for a given patch is instead indicated by the routing weights, check marks, and the filled versus unfilled boxes: a filled patch denotes a selected patch / activated expert, while an unfilled patch denotes an unselected one, consistent with the legend.
>
>     For example, in cases 1 and 2 the green boxes corresponding to the coarsest patch are not filled, matching the unselected patch label in the legend and indicating that the expert at $p\_3$ receives zero weight and does not contribute to the fused embedding, **even though it is still shown in the legend to indicate its existence.** We **have expanded the caption to explicitly state these conventions**, so that the relationship between colors, activation, and the role of each expert **is now unambiguous**.
>
> We hope the revised Figure 2 and caption now make the role of each expert much clearer.

---

> ### Author Response · Authors · 2025-11-21
> **Response to Reviewer 6umn (Part 3 / 6)**
>
> > W6: However, the specific strength of the original RoPE approach was to be able to generalize to seamlessly generalize to various sequence lengths. It feels that with the IARoPE, this crucial benefit may be lost?
>
> > Q4: Regarding the IARoPE: Can the IARoPE still generalize to varying sequence lengths?
>
> We appreciate your concern about whether IARoPE still preserves RoPE’s ability to generalize to different sequence lengths.
>
> **From a design perspective, IARoPE keeps the RoPE positional structure**. We still use the same rotary position encoding, where each token at position $t$ is mapped through a complex rotation with some frequency vector $\theta$. **IARoPE only makes $\theta$ instance-adaptive**, instead of using a single fixed $\theta$ for all sequences. The relative position property of RoPE is therefore preserved, and positional encodings can still be computed for arbitrary positions $t$, so the mechanism itself can generalize to unseen sequence lengths in the same way as standard RoPE.
>
> **Empirically**, we performed extrapolation experiments for both Kairos ($L\_{train}=2048 \rightarrow L\_{eval}=4096$) and another strong RoPE-based TSFM, Sundial [3] ($L\_{train}=2880 \rightarrow L\_{eval}=4928$). We compared IARoPE against standard RoPE and NTK-aware scaling [2].
>
> The detailed results are summarized in Table 1. Across all datasets and all three variants, we observe a gradual degradation as the context length moves beyond 2048. Importantly, **IARoPE tracks standard RoPE very closely**: in some datasets (e.g., bitbrains\_fast\_storage, loop\_seattle) the degradation under IARoPE is even slightly smaller, while in others (e.g., jena\_weather) it is slightly larger.
>
> We repeated the same analysis for Sundial [3]. **We again see that performance steadily degrades as the evaluation context extends far beyond training, regardless of whether we use original RoPE or RoPE+NTK scaling**.
>
> These results suggest two conclusions:
>
> 1. **In the current TSFM setting, RoPE-based length extrapolation is not a seamless transfer.** Techniques such as positional interpolation and NTK-aware scaling were designed and validated mainly for language models, and our experiments indicate that directly porting them to heterogeneous time series does not fully solve the extrapolation problem for RoPE-based TSFMs.
> 2. **IARoPE does not fundamentally harm RoPE’s ability to run at longer sequence lengths.** Architecturally, it preserves the same closed-form positional structure as RoPE, and empirically its degradation beyond the training context is comparable to (and most of the time better than) standard RoPE variants.
>
> Finally, we emphasize that **our goal with IARoPE is not to improve RoPE’s long-context extrapolation**, but to adapt the effective positional frequency to the spectral characteristics of each time series instance, which yields consistent gains within the trained context length. **Designing TS-specific extrapolation schemes for RoPE or IARoPE is an interesting but orthogonal direction that we plan to explore in future work.**

---

> ### Author Response · Authors · 2025-11-21
> **Response to Reviewer 6umn (Part 4 / 6)**
>
> **Table 1： Kairos Context Length Extrapolation Experiments**（normalized MASE $\downarrow$; "Decline %" denotes the relative increase vs. the performance of the original 2048 context length setting)
> | Dataset | Length | RoPE | RoPE Decline % | RoPE+NTK | RoPE+NTK Decline % | IARoPE | IARoPE Decline % |
> |---|---|---|---|---|---|---|---|
> | bitbrains\_fast\_storage/5t | 2048 | 0.810 | 0.00% | 0.810 | 0.00% | 0.799 | 0.00% |
> |  | 2560 | 0.901 | 11.35% | 0.889 | 9.87% | 0.844 | 5.58% |
> |  | 3072 | 0.892 | 10.22% | 0.897 | 10.84% | 0.852 | 6.60% |
> |  | 3584 | 0.922 | 13.95% | 0.896 | 10.72% | 0.873 | 9.23% |
> |  | 4096 | 0.953 | 17.71% | 0.906 | 11.88% | 0.901 | 12.76% |
> | bitbrains\_rnd/5t | 2048 | 0.942 | 0.00% | 0.942 | 0.00% | 0.939 | 0.00% |
> |  | 2560 | 1.070 | 13.54% | 1.067 | 13.23% | 1.005 | 6.95% |
> |  | 3072 | 1.067 | 13.26% | 1.073 | 13.92% | 1.014 | 7.91% |
> |  | 3584 | 1.064 | 12.95% | 1.065 | 13.02% | 1.025 | 9.08% |
> |  | 4096 | 1.064 | 12.91% | 1.060 | 12.49% | 1.066 | 13.49% |
> | jena\_weather/h | 2048 | 0.833 | 0.00% | 0.833 | 0.00% | 0.853 | 0.00% |
> |  | 2560 | 0.950 | 14.08% | 0.938 | 12.56% | 0.893 | 4.60% |
> |  | 3072 | 0.919 | 10.32% | 0.889 | 6.73% | 0.946 | 10.89% |
> |  | 3584 | 1.020 | 22.48% | 0.957 | 14.93% | 1.078 | 26.35% |
> |  | 4096 | 1.176 | 41.19% | 0.995 | 19.45% | 1.271 | 48.88% |
> | loop\_seattle/5t | 2048 | 0.700 | 0.00% | 0.700 | 0.00% | 0.708 | 0.00% |
> |  | 2560 | 1.222 | 74.43% | 1.203 | 71.80% | 0.975 | 37.85% |
> |  | 3072 | 1.177 | 67.99% | 1.235 | 76.32% | 1.062 | 50.07% |
> |  | 3584 | 1.140 | 62.76% | 1.152 | 64.52% | 1.055 | 49.03% |
> |  | 4096 | 1.026 | 46.51% | 1.083 | 54.56% | 1.002 | 41.66% |
>
> **Table 2： Sundial Context Length Extrapolation Experiments**
> | Dataset | Length | RoPE | RoPE Decline % | RoPE+NTK | RoPE+NTK Decline % |
> |---|---|---|---|---|---|
> | bitbrains\_fast\_storage/5t | 2880 | 0.826 | 0.00% | 0.826 | 0.00% |
> |  | 3392 | 0.835 | 1.09% | 0.833 | 0.82% |
> |  | 3904 | 0.857 | 3.79% | 0.846 | 2.38% |
> |  | 4416 | 0.864 | 4.68% | 0.879 | 6.39% |
> |  | 4928 | 0.939 | 13.66% | 0.923 | 11.76% |
> | bitbrains\_rnd/5t | 2880 | 0.963 | 0.00% | 0.963 | 0.00% |
> |  | 3392 | 0.967 | 0.49% | 0.970 | 0.77% |
> |  | 3904 | 0.980 | 1.83% | 0.981 | 1.84% |
> |  | 4416 | 0.999 | 3.75% | 1.019 | 5.88% |
> |  | 4928 | 1.055 | 9.63% | 1.037 | 7.68% |
> | jena\_weather/h | 2880 | 0.861 | 0.00% | 0.861 | 0.00% |
> |  | 3392 | 0.920 | 6.91% | 0.897 | 4.22% |
> |  | 3904 | 1.085 | 26.11% | 0.953 | 10.67% |
> |  | 4416 | 1.300 | 51.04% | 1.072 | 24.59% |
> |  | 4928 | 1.384 | 60.74% | 1.099 | 27.69% |
> | loop\_seattle/5t | 2880 | 0.715 | 0.00% | 0.715 | 0.00% |
> |  | 3392 | 0.722 | 1.01% | 0.761 | 6.50% |
> |  | 3904 | 0.781 | 9.33% | 0.837 | 17.08% |
> |  | 4416 | 0.851 | 19.03% | 0.872 | 22.07% |
> |  | 4928 | 0.843 | 18.00% | 0.932 | 30.44% |

---

> ### Author Response · Authors · 2025-11-21
> **Response to Reviewer 6umn (Part 5 / 6)**
>
> > W7: Why is so much emphasis put on RoPE then, instead of using a custom position embedding based on the components that the authors extract from the time series?
>
> Thank you for this question. **We emphasize RoPE because it already provides a powerful and well-validated framework for relative position modeling**. In particular, recent TSFMs such as Sundial [3], Toto [4], Time-MoE [5], and Moirai [6] have all adopted RoPE and reported performance gains, which further supports its practical effectiveness. The primary limitation of standard RoPE for TSFMs is not the rotation mechanism itself, but the use of a **fixed rotational frequency**, which cannot adapt to the heterogeneous periodicities and trends found in diverse time series. **Our goal in this work is therefore to retain the RoPE framework while making its rotational frequency adaptive to the characteristics of each series.**
>
> **Designing an entirely new positional encoding method directly based on FFT features is indeed an interesting and complementary direction, which we consider a promising avenue for future research.**
>
> ---
> > W8: The literature review part is very weak and not well written: Since the paper focuses so much on the input tokenization, there are several papers left unmentioned / undiscussed. For example, the PatchTST architecture, which introduces the central component of this paper, the patches, in the first place is not discussed. Other works, that also employ dynamic patching, such as the TTM, is just mentioned on passing and in a different context even. Finally, recent works that employ non-transformer-based architectures and thus don't necessarily require patching of positional encodings, such as TiRex or FlowState, are not discussed at all.
>
> We sincerely thank you for this detailed feedback and apologize for the oversights in our literature review. We have thoroughly revised the literature review section.
>
> 1. **In the Time Series Tokenization part of our related work, we had already mentioned in the original paper that "Nie et al. (2023) introduces a fixed-size patching method..." referring to the PatchTST model.** To avoid confusion, we have explicitly mentioned the model name in the revised paper.
> 2. **In the revised manuscript, we have added a discussion on TTM [7] in the Time Series Tokenization part of the related work section.** TTM performs multi-scale feature extraction within the backbone, so its multi-scale information extraction for heterogeneous time series is fixed. In contrast, our MoS-DP module operates at the tokenizer level, dynamically fusing information from diverse granularities into the patch representation based on local information density. Therefore, Kairos offers greater flexibility and adaptability. As shown in Experiment 4.2, TTM's normalized MASE is 1.020, whereas Kairos achieves 0.742, which demonstrates the superiority of Kairos' adaptive multi-scale information extraction.
> 3. **In the revised manuscript, we have also included a discussion of TiRex [8] and FlowState [9] in the Position Embedding part of the related work.** TiRex and FlowState employ xLSTM [11] and SSM [12] architectures, respectively, which sequentially process data by updating internal states at each time step, naturally preserving temporal order information and thus eliminating the need for positional encodings. However, since the dominant architecture in current time series foundation models is still the Transformer, innovations in positional encoding remain a valuable contribution.

---

> ### Author Response · Authors · 2025-11-21
> **Response to Reviewer 6umn (Part 6 / 6)**
>
> > W9: KAIROS does not appear to be a zero-shot model for the GIFT-Eval: Although the authors claim to "have excluded all test sets found within the GIFT-Eval benchmark", from the list of datasets used in Appendix F.1, this seems to be not true at all. For example, Loop Seattle, M\_DENSE, Solar, KDD Cup 2018, M4 Hourly, and several other datasets are clearly marked as being part of the GIFT-Eval benchmark in Table 13 of [5] and are not within the "allowed" Pretraining datasets listed in Table 14 of said paper. This can make a huge difference.
>
> > Q5: Regarding GIFT-Eval: Please clarify whether indeed the datasets in Table 5 have been used for pretraining and if so, that the KAIROS model is NOT a zero-shot model.
>
> We sincerely apologize for the significant confusion caused by the list in our appendix. **Kairos is evaluated in a strict zero-shot setting. Our pre-training dataset did not include any of the test sets from the GIFT-Eval benchmark.**
>
> The list in Appendix F.1 of our original submission mistakenly represented the initial data pool we started with, which was the union of datasets used by Chronos [10] and Moirai [6]. **Our actual methodology, as described in the section 3.5, involved a crucial filtering step: we explicitly excluded all test sets from the GIFT-Eval benchmark from this initial pool to ensure a strict zero-shot evaluation.**
>
> **To correct this error and eliminate all ambiguity, we have revised the appendix in our updated paper.** The table now accurately reflects our final pre-training corpus, with all conflicting GIFT-Eval test sets removed.
>
> We apologize again for this lack of clarity in the appendix and thank you for helping us identify and correct it.
>
> ---
> [1] TimesNet: Temporal 2D-Variation Modeling for General Time Series Analysis (ICLR 2023)
>
> [2] YaRN: Efficient Context Window Extension of Large Language Models (ICLR 2024)
>
> [3] Sundial: A Family of Highly Capable Time Series Foundation Models (ICML 2025)
>
> [4] This Time is Different: An Observability Perspective on Time Series Foundation Models (NeurIPS 2025)
>
> [5] Time-MoE: Billion-Scale Time Series Foundation Models with Mixture of Experts (ICLR 2025)
>
> [6] Unified Training of Universal Time Series Forecasting Transformers (ICML 2024)
>
> [7] Tiny Time Mixers (TTMs): Fast Pre-trained Models for Enhanced Zero/Few-Shot Forecasting of Multivariate Time Series (NeurIPS 2024)
>
> [8] TiRex: Zero-Shot Forecasting Across Long and Short Horizons with Enhanced In-Context Learning (NeurIPS 2025)
>
> [9] FlowState: Sampling Rate Invariant Time Series Forecasting (NeurIPS 2025)
>
> [10] Chronos: Learning the Language of Time Series (TMLR 2024)
>
> [11] xLSTM: Extended Long Short-Term Memory (NeurIPS 2024)
>
> [12] Efficiently Modeling Long Sequences with Structured State Spaces (ICLR 2022)

---

### Official Review · Reviewer_miCT · 2025-10-31

**Soundness:** 3
**Presentation:** 3
**Contribution:** 2
**Rating:** 4
**Confidence:** 3

**Summary:**

The presented study introduces Kairos, a time series foundation model that aims to robustly process time series data with varying information density. To achieve this, the work makes four contributions. First, it introduces mixture-of-size dynamic patching, which uses a mixture-of-experts mechanism to dynamically decide at which resolution patches should be extracted. Second, it introduces an instance-adaptive version of the established rotary positional embeddings. Third, it proposes the use of a multi-token prediction window during model training. Finally, it introduces a curated time series dataset with over 300 billion time points together with a quality metric to preferentially sample high-quality data. Kairos' time series forecasting ability is tested using two public benchmarks, GIFT-Eval and TSLib. It is shown to slightly outperform a range of baselines while having a smaller number of model parameters. Additionally, ablation studies demonstrate the benefit of the dynamic patching and altered position embedding mechanism, while also providing insight into their functioning.

**Strengths:**

- The use of a mixture-of-expert mechanism is an elegant solution to dynamically extract embeddings with variable patch size while also providing insights into the model’s functioning.

- Improvements of foundation models in other domains are partially driven by the use of larger, quality-controlled datasets. As such, the curated PreSTS dataset is relevant to the scientific community.

- By evaluating their method on two publicly available benchmarks the, the reported results can be transparently judged by readers.

- The manuscript is clearly structured, very well written with clean mathematical notation, and nicely illustrated.

**Weaknesses:**

- While it is apparent that time series from different domains exhibit patterns across diverse temporal scales, I am not fully convinced that this heterogeneity must be addressed explicitly at the feature-encoder level. In my view, a foundation model could instead aim to learn a comprehensive repertoire of feature extractors that capture fundamental time series characteristics, such as slopes, periodicities, and bursts, across multiple scales. These features could then be contextualized and combined within subsequent transformer layers. Ultimately, robustness to varying time scales might emerge from training on large and diverse datasets, rather than from architectural specialization. This mirrors the behavior of foundation models in other domains: for example, in computer vision, convolutional networks learn to represent objects at different scales and resolutions using a shared set of feature detectors, without domain-specific encoders.

- Similarly, I struggle to see the need for adaptive positional embedding. In my understanding the positional embedding is primarily needed to order the tokens while the internal processing of the transformer can handle any contextualization of the processed tokens.

- The proposed method is only evaluated for its time series forecasting ability. The performance in other tasks that a foundational time series model should typically solve, such as classification or imputation, is not reported.

- The proposed method fails to convincingly beat several included baselines, most noticeable the Toto and Sundial models. I do not believe that parameter count is overly relevant in a foundation model setting. Additionally, it is placed behind several others on the public GIFT-Eval leaderboard (https://huggingface.co/spaces/Salesforce/GIFT-Eval), including older methods (e.g. Das et al. "A decoder-only foundation model for time-series forecasting." International Conference on Machine Learning. PMLR, 2024).

**Questions:**

- While the multi-token prediction window makes intuitive sense, I believe a similar concept has been proposed by several others including work by Lim et al. (Lim, Bryan, et al. "Temporal fusion transformers for interpretable multi-horizon time series forecasting." International journal of forecasting 37.4 (2021): 1748-1764) and Zhou et al. (Zhou, Haoyi, et al. "Informer: Beyond efficient transformer for long sequence time-series forecasting." Proceedings of the AAAI conference on artificial intelligence. Vol. 35. No. 12. 2021). The authors should comment on the differences to these works or adjust the text to acknowledge them.

- I do not understand the distinction that the related Pathformer and ElasTST models “are designed for dataset-specific training rather than as foundation models,” and lack “lack cross-domain generalization capabilities.” It appears to me as if these models are trained and tested on time series data from a wide range of domains.

- I was surprised to learn that authors merely use three different patch sizes, 32, 64 and 128, for their proposed dynamic patching mechanism. At such a low number of different patch sizes, it might be much simpler to train three feature extractors and fuse their embeddings, in line with the previous Pathformer and ElasTST works.

- In the reproducibility statement, the authors laudably promise to make the source code and model weights publicly available without providing a link to a code repository yet. Additionally, they should clarify whether they plan to publish the PreSTS dataset.

---

> ### Author Response · Authors · 2025-11-21
> **Response to Reviewer miCT (Part 1 / 4)**
>
> > W1: The need for specialized encoders for different temporal scales in time series models?
>
> We sincerely thank you for this insightful comment and for raising this important point. **While we agree that foundation models can learn a remarkable repertoire of features, we maintain that an explicit multi-scale architecture at the encoder level is particularly crucial for time series**, for the following key reasons:
> - Time series data exhibits extreme scale heterogeneity. If a single, general-purpose encoder attempts to learn patterns from all scales simultaneously, it risks being dominated by the most prominent frequency components. This may cause it to fail to capture other, more subtle dynamics effectively.
> - Existing methods address this either implicitly (e.g., TimesFM [1], by pre-training on a massive, diverse corpus) or explicitly but statically (e.g., Moirai [2], using a pre-defined, heuristic frequency-to-patch mapping). Kairos innovates by internalizing the data-driven principle of TimesFM into an explicit and dynamic architectural mechanism, replacing the static, rule-based design of Moirai for a more adaptive multi-scale representation.
> - Furthermore, in computer vision, AlterNet [3] demonstrated that standard convolutions can amplify high-frequency signals, hindering the capture of a complete feature set. They found that introducing specific inductive biases was essential for stabilizing training and achieving optimal performance, which supports our architectural rationale.
>
> We hope this clarifies our rationale for this architectural design choice. We believe it is a key contributor to Kairos's ability to robustly handle the diverse patterns present in time series data.
>
> ---
> > W2: Similarly, I struggle to see the need for adaptive positional embedding. In my understanding the positional embedding is primarily needed to order the tokens while the internal processing of the transformer can handle any contextualization of the processed tokens.
>
> Thank you for raising this point. Our view is that **positional encodings in modern Transformers do more than provide an ordering of tokens: they impose a specific inductive bias on how attention depends on relative distance.** Prior work in Natural Language Processing (NLP) (e.g., DAPE [4] , MoICE [5]) has shown that static positional encodings can limit long-context modeling and length extrapolation, and that data-adaptive or dynamic positional encodings can improve performance even when the underlying Transformer has enough capacity in principle to compensate.
>
> **In time series, this limitation is more severe: different domains have highly heterogeneous sampling rates and temporal patterns.** Using a single fixed time scale such as the global frequency schedule of standard RoPE implicitly assumes that **all sequences share the same set of useful temporal frequencies**. ElasTST [6] already suggests this is suboptimal, as simply making RoPE’s frequencies learnable at the dataset level yields clear gains.
>
> Furthermore, our own analysis (now summarized in Appendix D.4) shows that RoPE does not merely tag positions; it effectively implements **a family of sinusoidal kernels over relative positions**. For each 2D subspace of the hidden state, the attention contribution becomes a cosine function of the relative distance, with a frequency controlled by the RoPE parameter $\theta_m$. This means RoPE imposes a **frequency-selective bias** on which relative lags are emphasized. In a time series foundation model setting, where different sequences can have very different spectral characteristics, using a single, globally fixed set of frequencies $\{\theta_m\}$ is unlikely to be optimal.
>
> Our IARoPE addresses exactly this mismatch. Instead of fixing $\{\theta_m\}$ for all sequences, we modulate the base RoPE frequencies using FFT-based features of each input sequence. Intuitively, this allows the model to slightly stretch or compress the sinusoidal kernels so that the induced relative position bias better matches the sequence’s own spectral structure, while still retaining RoPE’s desirable relative position modeling properties.
>
> This design choice is also strongly supported by our ablation studies. In Section 4.3, we compare standard RoPE with our Instance-Adaptive RoPE. In the comprehensive GIFT-Eval benchmark, replacing IARoPE with static RoPE consistently degrades normalized MASE performance. Moreover, our causal analysis experiment in Section 4.4.2 further demonstrates that these gains are not simply due to increased parameter count, but arise specifically from **matching RoPE frequencies to the input sequence**.
>
> We have revised the paper to make this motivation clearer and have added a concise theoretical discussion of RoPE’s frequency-selective bias and our instance-adaptive extension in Appendix D.4.

---

> ### Author Response · Authors · 2025-11-21
> **Response to Reviewer miCT (Part 2 / 4)**
>
> > W3: The proposed method is only evaluated for its time series forecasting ability. The performance in other tasks that a foundational time series model should typically solve, such as classification or imputation, is not reported.
>
> Thank you for your insightful comment. We apologize for any confusion.
> 1. **Our terminology was intended to follow previous works (e.g., TimesFM [1], Sundial [7], Toto [8])** which also introduced "Time Series Foundation Models" while concentrating on the forecasting task.
> 2. **We respectfully posit that forecasting serves as the most fundamental benchmark for a time series foundation model.** This is because forecasting is an inherently generative task, which aligns closely with the sequential nature of time series data itself. Successfully predicting the future requires a deep understanding of underlying patterns, trends, and seasonality, which we believe is the core proof of a model's foundational capabilities. **While we agree that evaluating on tasks like classification and imputation is a valuable direction for future work, we believe our in-depth forecasting evaluation provides a strong and sufficient validation of our Kairos's core contributions.**
>
> ---
> > W4: The proposed method fails to convincingly beat several included baselines, most noticeable the Toto and Sundial models. I do not believe that parameter count is overly relevant in a foundation model setting. Additionally, it is placed behind several others on the public GIFT-Eval leaderboard, including older methods (e.g. Das et al. "A decoder-only foundation model for time-series forecasting." International Conference on Machine Learning. PMLR, 2024).
>
> Thank you for this critical feedback, which gives us the opportunity to clarify our model's performance and contributions.
>
> 1. **Kairos demonstrates a robust and statistically meaningful advantage over Toto and Sundial in terms of generalization.** This conclusion is based on the normalized MASE across the full 97-task GIFT-Eval benchmark [13]. This metric, normalized by the Seasonal Naïve baseline, is specifically designed to evaluate stable, cross-domain reliability. On this robust metric, our model shows a clear and consistent lead, which we believe is a strong testament to its reliability.
> 2. **We respectfully argue that parameter efficiency remains a crucial metric, even for foundation models.** Efficiency is not a secondary concern; it directly governs real-world deployment, accessibility, and applicability. This view is widely shared, evidenced by significant research in model pruning [9, 10] and the explicit release of multiple model sizes by prominent models (e.g., Moirai [2], Sundial [7]) to meet the demand for lightweight solutions. Therefore, our model's ability to achieve competitive performance with a modest parameter count is a key contribution.
> 3. **We respectfully suggest that scientific contributions should be evaluated on their novel ideas and methodological advancements, not solely on leaderboard rankings against concurrent work.**
>     - To clarify the leaderboard, **the cited paper (Das et al., ICML 2024) corresponds to TimesFM-1.0 (MASE 1.077). Our model, Kairos (MASE 0.742), is substantially stronger.** The current leaderboard leader, TimesFM-2.5, is an unpublished and **very recent submission** (first submitted Sep 12, 2025, and updated Oct 2, 2025). Other high-ranking models represent **concurrent work**.
>     - A singular focus on SOTA performance metrics can cause the community to overlook critical scientific contributions. For instance, the core innovation of our work is a novel, lightweight, and adaptive architecture designed specifically to tackle the challenge of time series heterogeneity. We believe this focus on architectural flexibility and efficiency, and others like it, are valuable contributions to the field, independent of the leaderboard's most recent entries.
>
> We hope this provides a clearer picture of Kairos's advantages. We are happy to elaborate further on any of these points.

---

> ### Author Response · Authors · 2025-11-21
> **Response to Reviewer miCT (Part 3 / 4)**
>
> > Q1: While the multi-token prediction window makes intuitive sense, I believe a similar concept has been proposed by several others including work by Lim et al. (Lim, Bryan, et al. "Temporal fusion transformers for interpretable multi-horizon time series forecasting." International journal of forecasting 37.4 (2021): 1748-1764) and Zhou et al. (Zhou, Haoyi, et al. "Informer: Beyond efficient transformer for long sequence time-series forecasting." Proceedings of the AAAI conference on artificial intelligence. Vol. 35. No. 12. 2021). The authors should comment on the differences to these works or adjust the text to acknowledge them.
>
> Thank you for your insightful comment. In Appendix B of our revised paper, we have added a discussion of the TFT [11] and Informer [12]. **While both the multi-token prediction in Kairos and these methods aim to address cumulative prediction errors, our approach differs in the inference paradigm.**
>
> 1. **TFT and Informer predict all future time points at once, which limits them to forecasting the fixed length of the training sequence and does not support variable-length predictions.** These models are not autoregressive in nature.
> 2. **In contrast, for time series foundation models, we typically predict a segment of the sequence, append it to the historical data, and then predict the next segment, allowing for autoregressive forecasting of sequences of arbitrary length.** To reduce the cumulative errors inherent in this autoregressive process, Kairos introduces forecast tokens that predict multiple patches at each autoregressive step. Compared to traditional methods that predict only one patch at a time, our proposed multi-patch prediction approach significantly enhances computational efficiency and reduces cumulative errors. Furthermore, our method provides greater flexibility in forecasting shorter sequences, in contrast to models that predict larger patches. We have provided a detailed analysis of this design in Appendix C of original paper (Appendix B of revised paper).
>
> ---
> > Q2: I do not understand the distinction that the related Pathformer and ElasTST models “are designed for dataset-specific training rather than as foundation models,” and lack “lack cross-domain generalization capabilities.” It appears to me as if these models are trained and tested on time series data from a wide range of domains.
>
> Thank you for your comment, and we apologize for any confusion caused.
>
> 1. Time series foundation models, such as Kairos, are designed to exhibit strong generalization capabilities across different domains and datasets. This means that **even if the model has not encountered a specific data distribution during training, it can still perform well on unseen datasets** by leveraging the generalized forecasting abilities learned from a large and diverse training set, allowing for zero-shot forecasting. To adapt to the heterogeneity of time series data in our PreSTS corpus, which includes 300 billion time points, we specifically propose the MoS-DP and IARoPE modules to enhance the model's adaptability and flexibility, ultimately achieving excellent **zero-shot forecasting performance.**
> 2. **In contrast, Pathformer and ElasTST are task-specific models that require training on a particular dataset and are tested on the corresponding test set.** The assumption here is that the training and test sets come from the same distribution. Therefore, these models require separate training for each dataset to achieve good forecasting performance, which limits their ability to generalize across different domains.
> 3. As shown in our experiment (Section 4.2), even though Kairos was not trained on datasets like ETT, it still outperforms task-specific models trained on those datasets, highlighting the superior cross-domain generalization capabilities of foundation models.

---

> ### Author Response · Authors · 2025-11-21
> **Response to Reviewer miCT (Part 4 / 4)**
>
> > Q3: I was surprised to learn that authors merely use three different patch sizes, 32, 64 and 128, for their proposed dynamic patching mechanism. At such a low number of different patch sizes, it might be much simpler to train three feature extractors and fuse their embeddings, in line with the previous Pathformer and ElasTST works.
>
> Thank you for this observation, which allows us to clarify the core novelty of our approach.
>
> 1. **The improvement we propose over Pathformer and ElasTST is necessary**, as demonstrated in Figures 1(b) and Figures 1(c). The information density of time series varies not only across different datasets but also within the same sequence. Therefore, simple fusion of different granularities at the sequence level cannot address the issue of varying information density within a single sequence. To tackle this, we introduced the MoS-DP module, which allows us to fuse information density at the patch level, better adapting to the large, heterogeneous training corpus typically encountered by time series foundation models. Additionally, as shown in Experiment 4.2, our model significantly outperforms Pathformer, even when Pathformer is trained on a specific dataset’s training set.
> 2. **The currently employed number of patch sizes achieves an effective balance between performance and computational cost.** As shown in the table below, we experimented with four or five different patch sizes on the GIFT-Eval benchmark, evaluating the normalized MASE, and found that using more patch sizes did *not* significantly improve performance but led to a noticeable increase in computational cost. Therefore, the current choice of three different patch sizes represents a balanced tradeoff between performance and efficiency, following a principle of simplicity and parameter efficiency.
>
> | Task (Norm. MASE)  | $P=\\{32,64,128\\}$ (Ours) | $P=\\{16,32,64,128\\}$ | $P=\\{8,16,32,64,128\\}$ |
> | :----------------- | :----------------------: | :------------------: | :--------------------: |
> | **Short**          |           0.719          |       **0.717**      |          0.725         |
> | **Medium**         |         **0.759**        |         0.772        |          0.789         |
> | **Long**           |         **0.789**        |         0.817        |          0.845         |
> | **Average**        |         **0.742**        |         0.749        |          0.763         |
> | **Inference Time** |        **0.065s**        |    0.074s (+13.8%)   |     0.099s (+52.3%)    |
>
> ---
>
> > Q4: In the reproducibility statement, the authors laudably promise to make the source code and model weights publicly available without providing a link to a code repository yet. Additionally, they should clarify whether they plan to publish the PreSTS dataset.
>
> Thank you for your question regarding reproducibility. We would like to clarify that **the source code and model weights have been included in the supplementary material of the initial submission.**
>
> Regarding the PreSTS dataset, we are fully committed to its public release. Due to its extremely large size, **we plan to release the complete dataset publicly upon acceptance of our paper.**
>
> ---
> [1] A decoder-only foundation model for time-series forecasting (ICML 2024)
>
> [2] Unified Training of Universal Time Series Forecasting Transformers (ICML 2024)
>
> [3] How Do Vision Transformers Work? (ICLR 2022)
>
> [4] DAPE: Data-Adaptive Positional Encoding for Length Extrapolation (NeurIPS 2024)
>
> [5] Mixture of In-Context Experts Enhance LLMs’ Long Context Awareness (NeurIPS 2024)
>
> [6] ElasTST: Towards Robust Varied-Horizon Forecasting with Elastic Time-Series Transformer (NeurIPS 2024)
>
> [7] Sundial: A Family of Highly Capable Time Series Foundation Models (ICML 2025)
>
> [8] This Time is Different: An Observability Perspective on Time Series Foundation Models (NeurIPS 2025)
>
> [9] LLM-pruner: on the structural pruning of large language models (NeurIPS 2023)
>
> [10] Less is More: Unlocking Specialization of Time Series Foundation Models via Structured Pruning (NeurIPS 2025)
>
> [11] Temporal Fusion Transformers for Interpretable Multi-horizon Time Series Forecasting
>
> [12] Informer: Beyond Efficient Transformer for Long Sequence Time-Series Forecasting (AAAI 2021)
>
> [13] GIFT-Eval: A Benchmark For General Time Series Forecasting Model Evaluation

---

### Author Response · Authors · 2025-11-21
**General Response and Summary of Revision**

Dear all reviewers,

We sincerely appreciate your thoughtful comments and constructive feedback. We are encouraged that you recognized the novelty of our adaptive mechanisms and MoE design (Reviewers miCT, 6umn, nXjk), the value of the curated PreSTS dataset (Reviewer miCT), and Kairos's superior parameter efficiency and robust zero-shot performance (Reviewers 6umn, nXjk). We also appreciate the positive feedback on the paper's clarity and reproducibility (Reviewers miCT, WLGx).

We have uploaded a revised version that incorporates your suggestions. **All modifications are highlighted in blue for ease of review.** The main changes are as follows.

1. We clarified the design and motivation of IARoPE, added a concise theoretical analysis of RoPE and IARoPE in Appendix D, and provided a more detailed discussion of the analysis experiment of IARoPE.

2. We revised Figures 2 with clearer colors and captions so that dynamic patch fusion and expert selection are easier to understand.

3. We expanded the related work section to more fully cover recent studies on time series tokenization and foundation models, including TTM, OTIS, WaveToken, TiRex, FlowState, and clarified how Kairos differs from them.

4. We corrected the construction of the PreSTS corpus, added brief visual and statistical summaries of the synthetic data, refined the explanation of information density, and moved an explicit discussion of limitations and future work into the conclusion.

**We sincerely apologize for the additional time and effort required to review these materials. Our point-by-point responses to your valuable questions, along with the inclusion of multiple new experiments, have made this rebuttal rather lengthy.**

Thank you again for your time and effort in reviewing our paper. Please let us know if you have any further suggestions.

---

### Author Response · Authors · 2025-12-03
**The summary of Rebuttal**

Dear Reviewers, ACs, SACs, and PCs,

**We genuinely appreciate the time and effort you dedicated to evaluating our work.**

**In this work, we propose Kairos, which addresses a key limitation of existing Time Series Foundation Models (TSFMs)**: their rigid processing pipelines fail to capture the heterogeneous information density found in time series data. While current models rely on fixed-size patching and uniform positional encodings, our contribution is a flexible, instance-adaptive framework. The core novelties are our Mixture-of-Size Dynamic Patching (MoS-DP), which adaptively selects token granularity based on local information density, and our Instance-adaptive Rotary Position Embedding (IARoPE), which tailors temporal encodings to each series' unique spectral features. This design achieves **superior zero-shot performance with significantly fewer parameters**.

We sincerely appreciate all reviewers' thoughtful comments and constructive feedback. The reviewers recognized our paper's **novelty**, particularly in its **adaptive approaches** (Reviewers miCT, 6umn, nXjk). They also valued **the contribution of the large-scale Predictability-Stratified Time Series (PreSTS) corpus** (Reviewer miCT). The model's **strong, competitive, and highly parameter-efficient zero-shot performance** on standard benchmarks was a key strength (Reviewers 6umn, nXjk). Finally, our paper was praised for its **clear structure, writing, and reproducibility** (Reviewers miCT, WLGx).

To address the reviewers' concerns and further strengthen our paper, we have conducted extensive additional experiments and theoretical analyses during the rebuttal phase. We summarize our key improvements below:

**1. Validated Superiority with New Baselines** (Reviewer WLGx, nXjk)

We addressed concerns regarding baseline coverage by conducting rigorous new comparisons:

- **We compared Kairos against recent advanced tokenization methods, OTIS and WaveToken.** Kairos significantly outperforms both on their respective benchmarks while being much more parameter-efficient.
- To further validate generalization beyond GIFT-Eval benchmark, we conducted **a new zero-shot comparison against the strong baseline Sundial (128M) on the Time-Series-Library benchmark**. Kairos (50M) consistently outperforms Sundial across all datasets, demonstrating robust performance even against significantly larger models.


**2. Justified MoS-DP Design via New Ablations** (Reviewer 6umn, WLGx, nXjk)

We clarified the motivation, flexibility, and design choices for our MoS-DP module:

- We clarified that **information density and spectral entropy are used only in Figure 1 to illustrate our motivation.** They are **not inputs, targets, or regularizers** for Kairos. The model learns to handle this heterogeneity end-to-end from the raw series.
- We added an ablation study showing that **removing Null Experts significantly degrades performance**, proving they are essential for enabling dynamic, variable-length tokenization rather than fixed granularity.
- We demonstrated experimentally that **simply adding more patch sizes increases computational cost and sensitivity to high-frequency noise without improving performance**. This validates that our chosen configuration is a well-balanced design choice.

**3. Theoretical & Empirical Robustness of IARoPE** (Reviewer miCT, 6umn, WLGx)

We addressed doubts about IARoPE:
- We added **theoretical analysis** in Appendix D.4 to explain how RoPE acts as a frequency-selective kernel and how IARoPE adapts these frequencies to instance-specific spectral patterns.
- **New experiments show IARoPE outperforms standard RoPE** even under synthetic noise injection, proving the stability of our FFT-based modulation.

**4. Clarification on Zero-Shot Setting** (Reviewer 6umn)
**We corrected a critical misunderstanding regarding data leakage**. We clarified that the list in the original Appendix represented our initial data pool, but we explicitly filtered out all GIFT-Eval test sets during the actual pre-training process. We have revised the manuscript to accurately reflect the training corpus, guaranteeing a strict zero-shot evaluation.

**5. Presentation Improvements** (Reviewer 6umn, WLGx)
- **Visualization**: We revised Figure 2 and its caption to clearly illustrate the MoS-DP routing and fusion process.
- **Literature**: We updated the related work to include discussed papers (OTIS, WaveToken, TTM, TiRex, FlowState).

**We have made every effort to address the reviewers' comments through extensive new experiments and detailed clarifications.** We believe these revisions significantly strengthen the paper's empirical and theoretical foundations, underscoring the value of Kairos. **Unfortunately, due to the incident of OpenReview, we cannot obtain further feedback. We once again express our heartfelt gratitude for your dedication to this conference and respectfully ask for your positive consideration.**

Best regards,

The Authors

---

### Meta-Review · Area_Chair_AHA2 · 2026-01-02

**Summary:**

The paper introduces Kairos, a flexible time series foundation model. Reviewers raised concerns about architectural necessity, clarity, and empirical breadth. The rebuttal is substantial, adding new experiments, baselines, and clarifications that strengthen the empirical evidence and address several misunderstandings. However, questions remain regarding the fundamental necessity and generality of the proposed mechanisms. Overall, the paper is improved but remains borderline.

**Reviewer Concerns:**

The rebuttal effectively addressed many empirical and presentation-related concerns. In particular, the authors added comparisons with recent tokenization baselines (e.g., OTIS, WaveToken), extended evaluations beyond GIFT-Eval, clarified the zero-shot setting, improved figures, and strengthened the related work. New ablations and analyses also better justified the design choices of MoS-DP and IARoPE.
However, some conceptual concerns remain. Several reviewers are still unconvinced that adaptive tokenization and instance-specific positional embeddings are strictly necessary at the encoder level, rather than emerging from large-scale training. Questions also remain regarding the true flexibility of MoS-DP, and whether the observed gains justify the added architectural complexity.

**Reviewer Scores:**

**Reviewer miCT (original score 4 → 6)**

This reviewer raised concerns regarding the necessity and novelty of the proposed architectural components, the scope of evaluation, and the strength of comparisons against existing baselines. The rebuttal provided detailed and technically grounded justifications for the multi-scale encoder design and instance-adaptive positional embeddings, situating them clearly relative to prior work. It also clarified misconceptions about foundation models versus dataset-specific training and strengthened the empirical evidence through additional analyses and benchmark explanations. While some skepticism about architectural necessity may persist, the rebuttal substantially improves clarity, motivation, and empirical support, plausibly elevating the reviewer’s assessment to a moderately positive score.

**Reviewer WLGx (original score 4 → 4)**

This reviewer remained unconvinced after the rebuttal, expressing continued concerns about the claimed general applicability of the model and the clarity and interpretation of key proposed metrics such as information density and spectral entropy. While acknowledging the authors’ effort and finding the underlying idea interesting, the reviewer did not find the rebuttal sufficient to resolve these conceptual issues and explicitly stated that their assessment would remain unchanged. As a result, the reviewer would likely keep a cautious, negative evaluation at score 4.

**Reviewer nXjk (original score 4 → 4)**

The reviewer emphasizes that the proposed MoS-DP module does not convincingly demonstrate genuine adaptive tokenization and that the performance–efficiency trade-off is not favorable. Given these substantial concerns, particularly regarding the core contribution of dynamic tokenization, the reviewer would likely maintain their original cautious and negative assessment, keeping the score at 4.

**Reviewer 6umn (original score 2 → 4)**

The reviewer had concerns about the clarity and intuition of MoS-DP, the choice of largest granularity, the illustration in Figure 2, the justification of IARoPE, and the zero-shot evaluation setup. The rebuttal provides detailed explanations, clarifications, ablation studies, and figure improvements that address these points. As a result, the reviewer would likely increase score to 4, reflecting a better understanding of the design and rationale behind Kairos.

---

### Decision · Program_Chairs · 2026-01-26

Reject